# The transcriptional regulatory network modulating human trophoblast stem cells to extravillous trophoblast differentiation

Mijeong Kim [1], Yu Jin Jang [1], Muyoung Lee [1], Qingqing Guo [1], Albert J. Son [1], Nikita A. Kakkad[1], Abigail B. Roland[1], Bum-Kyu Lee [2] & Jonghwan Kim [1]✉

During human pregnancy, extravillous trophoblasts play crucial roles in placental invasion into the maternal decidua and spiral artery remodeling. However, regulatory factors and their action mechanisms modulating human extravillous trophoblast specification have been unknown. By analyzing dynamic changes in transcriptome and enhancer profile during human trophoblast stem cell to extravillous trophoblast differentiation, we define stage-specific regulators, including an early-stage transcription factor, TFAP2C, and multiple late-stage transcription factors. Loss-of-function studies confirm the requirement of all transcription factors identified for adequate differentiation, and we reveal that the dynamic changes in the levels of TFAP2C are essential. Notably, TFAP2C pre-occupies the regulatory elements of the inactive extravillous trophoblast-active genes during the early stage of differentiation, and the late-stage transcription factors directly activate extravillous trophoblast-active genes, including themselves as differentiation further progresses, suggesting sequential actions of transcription factors assuring differentiation. Our results reveal stage-specific transcription factors and their inter-connected regulatory mechanisms modulating extravillous trophoblast differentiation, providing a framework for understanding early human placentation and placenta-related complications.

During early human pregnancy, embryonic trophoblast cells give rise to several differentiated cell types that support placental development. Among them are the extravillous trophoblasts (EVTs), which invade the maternal uterine decidua and remodel maternal spiral arteries to ensure adequate blood flow from the mother to the fetus for nutrient supply and gas exchange[1,2]. Shallow placental implantation and abnormal spiral artery remodeling due to defective EVT differentiation have been implicated in placenta-associated diseases, such as preeclampsia (PE)[3,4] and intrauterine growth restriction (IUGR)[4]. Such pregnancy complications not only cause maternal and perinatal morbidity and mortality but also affect the lifelong health of both mother[5,6]

and child[7,8]. While proper EVT differentiation is crucial for a successful pregnancy, regulatory factors required for EVT differentiation and their modes of action are not fully understood.

Given the limited access to early-stage developing human placentas, one key obstacle to studying human placentation has been a lack of appropriate in vitro models; immortalized human trophoblast cells, rodent trophoblast stem cells (TSCs), and trophoblast stem-like cells (TSLCs) established from pluripotent stem cells (PSCs) by bone morphogenetic protein 4 (BMP4) treatment have been used, but have considerable limitations, such as differences in mouse and human placentas and difficulties in maintaining self-renewing TSLCs[9,10].

[1]Department of Molecular Biosciences, The University of Texas at Austin, Austin, TX 78712, USA. [2]Department of Biomedical Sciences, Cancer Research Center, University at Albany, State University of New York, Rensselaer, NY 12144, USA. ✉e-mail: jonghwankim@mail.utexas.edu

Recently, bona fide human TSC lines have been established from human blastocysts and first-trimester chorionic villi. These self-renewing TSCs show bipotency to differentiate into EVTs and syncytiotrophoblasts (STs)[11]. TSCs therefore provide unprecedented opportunities to understand early human placentation, including EVT and ST specification, at molecular levels. Indeed, recent studies have used human TSC models to identify factors controlling trophoblast progenitors[12–14]. Moreover, a recent CRISPR-Cas9 screen has identified multiple essential factors controlling human TSC identity[15]. Another study has discovered the significance of ASCL2 as a pivotal factor in EVT differentiation using TSCs, and its role has been validated in vivo placenta development using a rat model[16], suggesting the feasibility of the use of TSC to EVT differentiation model to study in vivo EVT differentiation. However, despite the availability of human TSCs, regulation of proper EVT differentiation has yet to be systemically investigated.

Here, using human TSC to EVT differentiation as a model, we define previously unreported stage-specific transcription factors (TFs), including an early-stage TF, TFAP2C, and multiple late-stage TFs. With various experimental tools, we first reveal the requirement of these TFs controlling EVT differentiation and define how these stage-specific TFs regulate each other, forming transcriptional regulatory circuitry responsible for EVT-specific gene expression programs and functions. In sum, our results identify the factors critical for EVT differentiation and human placentation, providing insights into the regulatory programs controlling EVT differentiation.

## Results

### Transcriptome and enhancer dynamics during EVT differentiation

As little is known about EVT differentiation, we first carried out RNA-sequencing (RNA-seq) during time course differentiation of human TSCs to EVTs (Fig. 1a). Upon differentiation, epithelial to mesenchyme-like morphological changes and a corresponding increase in protein levels of HLA-G, an EVT marker gene, and a decrease in the expression of TP63, a TSC marker gene, were observed, indicating proper EVT differentiation (Fig. 1a and Supplementary Fig. 1a). The enhanced surface expression of HLA-G (>94% of HLA-G positive cell) further reinforced this observation (Supplementary Fig. 1b). The activation of EVT marker genes (HLA-G and MMP2) and downregulation of TSC markers (TP63 and TEAD4) were also observed in RT-qPCR and RNA-seq results (Fig. 1b and Supplementary Data 1). In addition, Gene Set Enrichment Analysis (GSEA) using the EVT and cytotrophoblast (CT, in vivo counterpart of TSCs) marker genes defined from the Molecular Signatures Database[17,18] and a scRNA-seq study of the human first-trimester placenta[19] further supported the proper upregulation of EVT-active genes along with the downregulation of CT genes, substantiating the proper EVT differentiation (Supplementary Fig. 1c). Principle component analysis (PCA) confirmed the gradual alteration of the transcriptome during EVT differentiation, which was distinct from the transcriptome of fully differentiated STs (Fig. 1c). The number of differentially expressed genes (DEGs; absolute log2-fold change >1 and $p < 0.05$) increased as differentiation progressed (Supplementary Fig. 1d), and we obtained a total of 5679 DEGs (corresponding to at least one of the time-points).

To functionally characterize the DEGs, we performed a Dirichlet Process Gaussian process (DPGP) mixture model clustering[20], and identified four distinct expression patterns (Fig. 1d and Supplementary Data 2). The genes belonging to class 1 showed high expression in TSCs and a gradual decrease in expression throughout differentiation (TSC-active). In contrast, class 2 genes were inactive in TSCs but continuously activated upon differentiation (EVT-active). Class 3 genes showed dynamic expression with strongest activity at the early-stage of EVT differentiation (at days 2 and 3) followed by rapid repression in mature EVTs (early-stage-active). Finally, class 4 genes were active in both TSCs and mature EVTs, but downregulated in between these two states.

To map corresponding changes in enhancer landscape, we performed chromatin immunoprecipitation coupled with next-generation sequencing (ChIP-seq) of H3K27ac[21] during EVT differentiation at day 0 (TSC), early-stage differentiation at day 3 (EVT D3), and mature EVT at day 8 (EVT D8) (Fig. 1a). We identified 46,591, 44,056, and 42,666 H3K27ac-positive sites in TSC, EVT D3, and EVT D8, respectively (Supplementary Fig. 1e), confirming substantial and dynamic changes in enhancer usage during EVT differentiation (Fig. 1f). A large portion of these sites localizes within intergenic or introns as expected (Supplementary Fig. 1e). We further validated the enhancers identified by H3K27ac signals using other well-established enhancer markers, including P300, MED1, and H3K4me1[22]. Supplementary Fig. 1f illustrates the overall similarities in the enhancer landscape. Specifically, we observed co-occupancy of P300 and MED1 at the enhancer loci, along with the enriched signals of H3K27ac and H3K4me1 close to these binding sites.

We confirmed that changes in enhancer landscape and gene expression are overall positively correlated, as observed in other contexts (Fig. 1d and Supplementary Fig. 1g)[23–26]. These results align with the notion that cell-type-specific enhancers determine cell-type-specific gene expression programs[27–29]. Validating our results, previously known TSC markers such as TP63, ELF5, and TEAD4 belonged to class 1, while the EVT markers, such as HLA-G and MMP2, and a known EVT regulator, ASCL2[16], were included in class 2 (Supplementary Fig. 1h and Supplementary Data 2). Furthermore, gene ontology (GO) analysis identified terms consistent with the known functions of each cell type. Class 1 genes were enriched in terms including cell division, cell cycle, and replication, consistent with rapidly proliferative TSCs[11,12]. Classes 2 and 4 genes were enriched in terms associated with EVT functions, such as invasive characteristics[30] and interaction with immune cells, which is known to occur during spiral artery remodeling[31]. Class 3 genes were enriched in cellular response to stress and intrinsic apoptotic signaling pathways, consistent with the prior reports describing that differentiation occurs along with cellular stress and apoptosis[32–34] (Fig. 1e).

### Two classes of EVT regulators

As master TFs are often associated with clusters of enhancers known as super-enhancers (SEs)[35], we determined SEs to identify candidate key regulators of EVT differentiation. We applied the ROSE algorithm[35,36] to the H3K27ac ChIP-seq data and mapped SE-associated genes in TSC, EVT D3, and EVT D8 cells (Supplementary Fig. 2a and Supplementary Data 3). As anticipated, the previously known TSC-associated TFs TEAD4[12], YAP1[37], and MSX2[14] were associated with SEs in TSCs. A recently reported EVT regulator ASCL2[16] was associated with the SEs defined in EVT D8 cells (Supplementary Fig. 2a and Supplementary Data 3), confirming the validity of our approach.

We first focused on the genes belonging to the classes 2 and 4, which are highly active in fully differentiated EVTs (Fig. 1d), as these genes were also enriched in EVT-related GO terms (Fig. 1e). Among 394 class 2 genes, 18 were TFs[38], and 7 TFs (DLX5, DLX6, ASCL2, ZNF439, IRF7, SNAI1, and VAX2) were associated with the SEs defined in EVT D8 (Fig. 2a–c). Among them, we selected 4 TFs (DLX5, DLX6, ASCL2, and ZNF439, hereafter, late-stage TFs) for further validation. We also included NRIP1 from class 4 as a late-stage TF candidate due to its association with multiple invasive cancers[39,40] and SEs defined in EVT D8. NRIP1 also highly expressed in mature EVTs, similar to other late-stage TFs (Fig. 2b, c).

We surmised that TFs belonging to class 3 may also play roles in EVT differentiation, perhaps acting as early-stage regulators as they showed a unique expression pattern, peaking at 2–3 days of differentiation, followed by rapid downregulation (Fig. 1d). Since there were

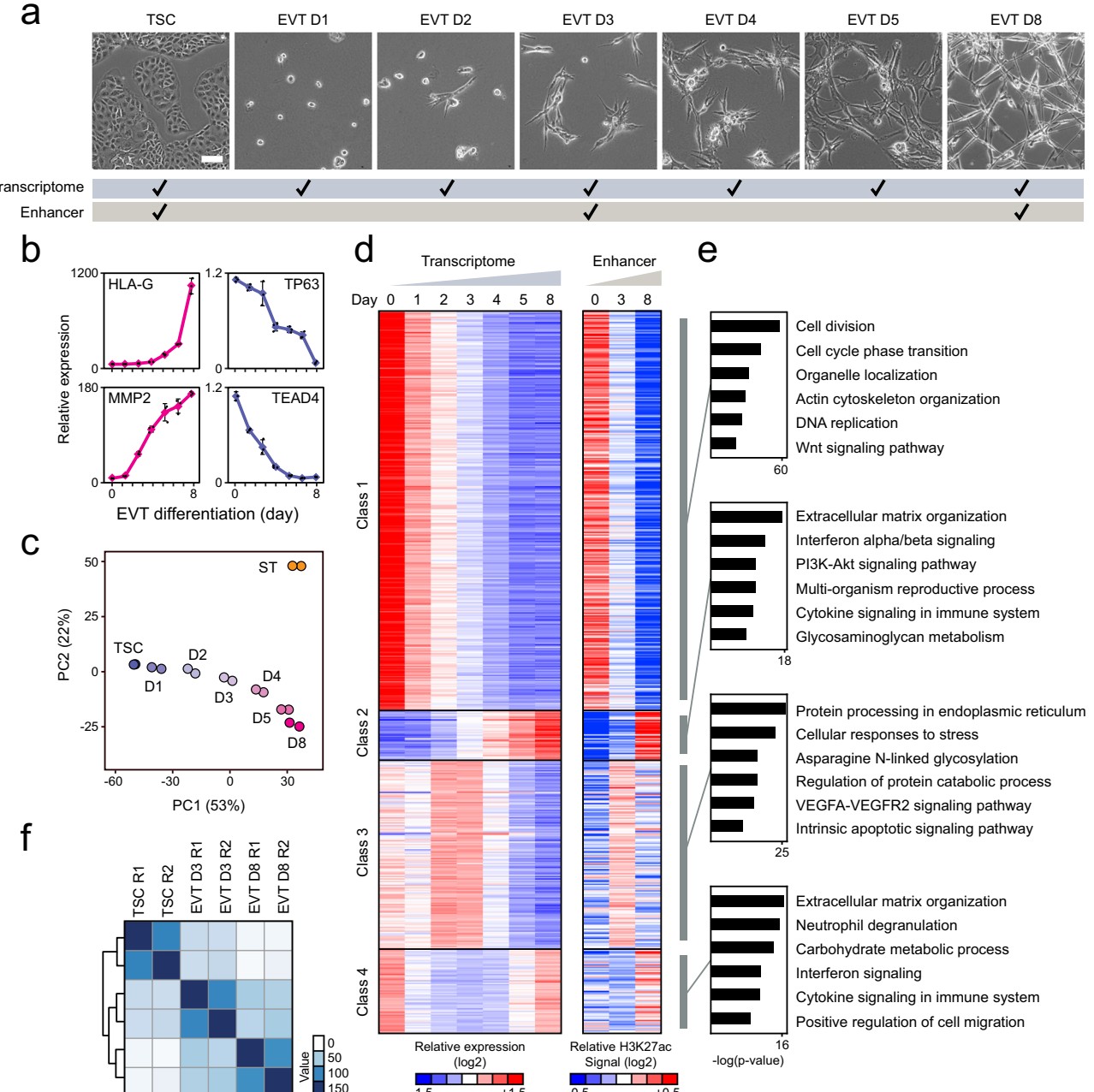

**Fig. 1 | Dynamic changes in transcriptome and enhancer landscape during EVT differentiation. a** Representative brightfield images of TSCs and differentiating cells to EVTs on days 1 (EVT D1), 2 (EVT D2), 3 (EVT D3), 4 (EVT D4), 5 (EVT D5), and 8 (EVT D8). Scale bar: 100 μm. Two independent repeats were carried out, resulting in similar results. Cells used for transcriptome profiling (top) and enhancer mapping (bottom) are labeled. **b** Relative mRNA expression levels of EVT marker genes (HLA-G and MMP2) and TSC marker genes (TP63 and TEAD4) in TSCs and EVT differentiating cells. The fold change relative to TSCs is plotted. Error bars: mean ± SD ($n = 3$, independent repeats). **c** Principle component analysis illustrating the

transcriptome profiles of TSCs, EVT differentiating cells, and STs. **d** Heatmaps showing Dirichlet Process Gaussian process (DPGP) mixture model clustering[20] of genes that exhibited differential expression during at least one time point compared to TSCs (left), and H3K27ac signals assigned to classes 1–4 genes (right). The values represent the log₂-transformed counts and H3K27ac signals, respectively, compared to the average. **e** GO analysis of classes 1–4 genes was conducted and p-values were calculated in Metascape[89]. **f** Sample-to-sample distances heatmap showing an overview of similarities and dissimilarities in H3K27ac signals among TSCs, EVT D3, and EVT D8.

many TFs-associated with SEs in this group (39 TFs, Fig. 2a), we additionally conducted a motif analysis of the top 30% enhancer peaks defined in EVT D3 cells. Interestingly, TFAP2C, whose motif is similar to that of TFAP2A, was the only SE-associated TF that overlapped with the top 5 motifs identified (Fig. 2d). In mice, *Tfap2c* activates TSC self-renewal genes[41] and has also been implicated in the trans-differentiation of mouse PSCs and fibroblasts into TSLCs[42,43]. Similarly, recent studies using human TSCs reported that *TFAP2C* is

required for self-renewal of TSCs[14,44]. Intriguingly, *TFAP2C* has also been implicated in human epidermal lineage commitment as an initiation-stage regulator of keratinocyte specification[45]. Based on our data and the known roles of *TFAP2C* during cell fate commitment, we selected TFAP2C as an early-stage regulator candidate controlling EVT differentiation.

To validate the expression patterns of the identified candidates for EVT regulators, we conducted RT-qPCR and Western blot analyses

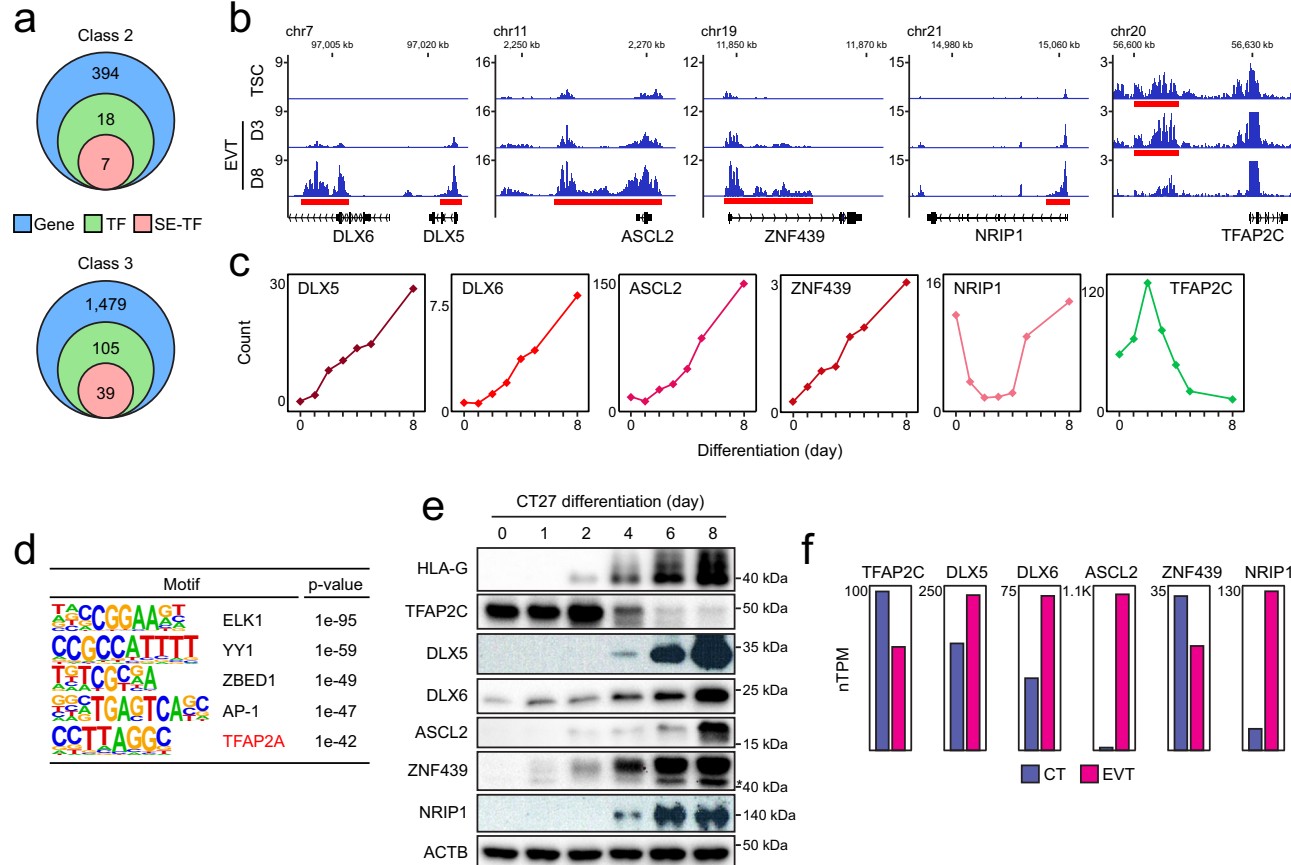

**Fig. 2 | Expression and enhancer association define two classes of EVT regulators. a** Number of genes (blue), TFs (green), and SE-associated TFs (red) in class 2 and 3 genes. **b** Gene track view of H3K27ac signals near the candidate regulators of EVT differentiation, with red bars indicating SEs. **c** Expression patterns of early- and late-stage TF candidates during EVT differentiation. Values represent $\log_2$-transformed counts compared to the average. **d** Enriched motifs within the top 30% of H3K27ac peaks in EVT on differentiation day 3 were identified using the find-MotifsGenome.pl module under HOMER (v4.11). *p*-values were calculated based on

the default settings of the module, with a region size of 500 bp. **e** Western blot analysis demonstrating the expression patterns of EVT regulator candidates in CT27 TSC line. HLA-G expression was measured to validate EVT differentiation. ACTB was used as a loading control. *ZNF439 (lower band). Independent repeats, utilizing other TSC lines, were conducted as presented in Supplementary Fig. 2d. **f** Normalized transcripts per million value (nTPM) of EVT regulator candidates in scRNA-seq data of human first-trimester placenta (retrieved from the Human Protein Atlas[19,102]).

in TSCs and differentiating cells towards EVT (CT27 TSC line). These experiments confirmed that the candidate EVT regulators displayed similar expression patterns to those observed in the RNA-seq results (Fig. 2e and Supplementary Fig. 2b). Notably, *NRIP1* exhibited comparable expression levels in TSCs and EVTs on differentiation day 8 in the RNA-seq data. However, the expression pattern of *NRIP1* during EVT differentiation, as observed in the RT-qPCR and Western blot results, resembled that of other late-stage TFs (Fig. 2e and Supplementary Fig. 2b, d). We further confirmed the consistent morphological changes and expression patterns of the candidate EVT regulators during EVT differentiation in other TSC lines (CT29 and CT30; Supplementary Fig. 2b–d).

To further determine the in vivo relevance of the candidate TFs, we verified the expression of selected TFs in human placentas. Data from single-cell RNA-seq (scRNA-seq) studies of the human first-trimester placentas[19,46] confirmed that the late-stage TFs, DLX5, DLX6, ASCL2, and NRIP1 are also highly expressed in EVTs (Fig. 2f and Supplementary Fig. 2e, f). As expected, TFAP2C expression was strong in CT and lower in EVT populations (Fig. 2f and Supplementary Fig. 2e, f). Overall, existing data validate our approach and classification scheme. Our transcriptome and enhancer analysis therefore identified a set of regulators expressed both early (TFAP2C) and late (DLX5, DLX6, ASCL2, ZNF439, and NRIP1) stages that were candidates for regulating the EVT differentiation process.

## Early and late-stage TFs are essential for EVT differentiation

To determine whether these candidate TFs are required for proper EVT differentiation, we conducted short hairpin RNA (shRNA)-mediated knockdown (KD) in TSCs and subsequently differentiated them to EVTs (Fig. 3a and Supplementary Fig. 3a). While control TSCs showed elongated and spindle-like cell morphology upon differentiation, KD of both early- and late-stage TFs led to an abnormal cellular morphology (Supplementary Fig. 3b) and the failure to induce EVT markers, such as HLA-G and MMP2 (Fig. 3a and Supplementary Fig. 3c). Notably, the impaired induction of the EVT markers was also observed in CT29 and CT30 TSC lines upon KD of the candidate TFs (Supplementary Fig. 3a, c). Since one critical characteristic of EVTs is their invasion capacity, we examined the invasiveness of the KD cells using invasion chambers (Supplementary Fig. 3d)[47]. Compared to control EVTs, all KD cells showed significantly reduced invasion ability, indicating that both early- and late-stage TFs are essential for normal EVT differentiation (Fig. 3b).

To further explore the importance of the early- and late-stage TFs in controlling EVT-specific gene expression programs, we profiled the transcriptome of each TF KD cell line upon EVT differentiation by bulk RNA-seq. We identified 1534-2072 DEGs in KD cells compared to the control cells (Supplementary Fig. 3e, f and Supplementary Data 4). Firstly, we monitored the expression of the class 1–4 genes we defined earlier (Fig. 1d). As shown in Fig. 3c, KD of these TFs led to defective

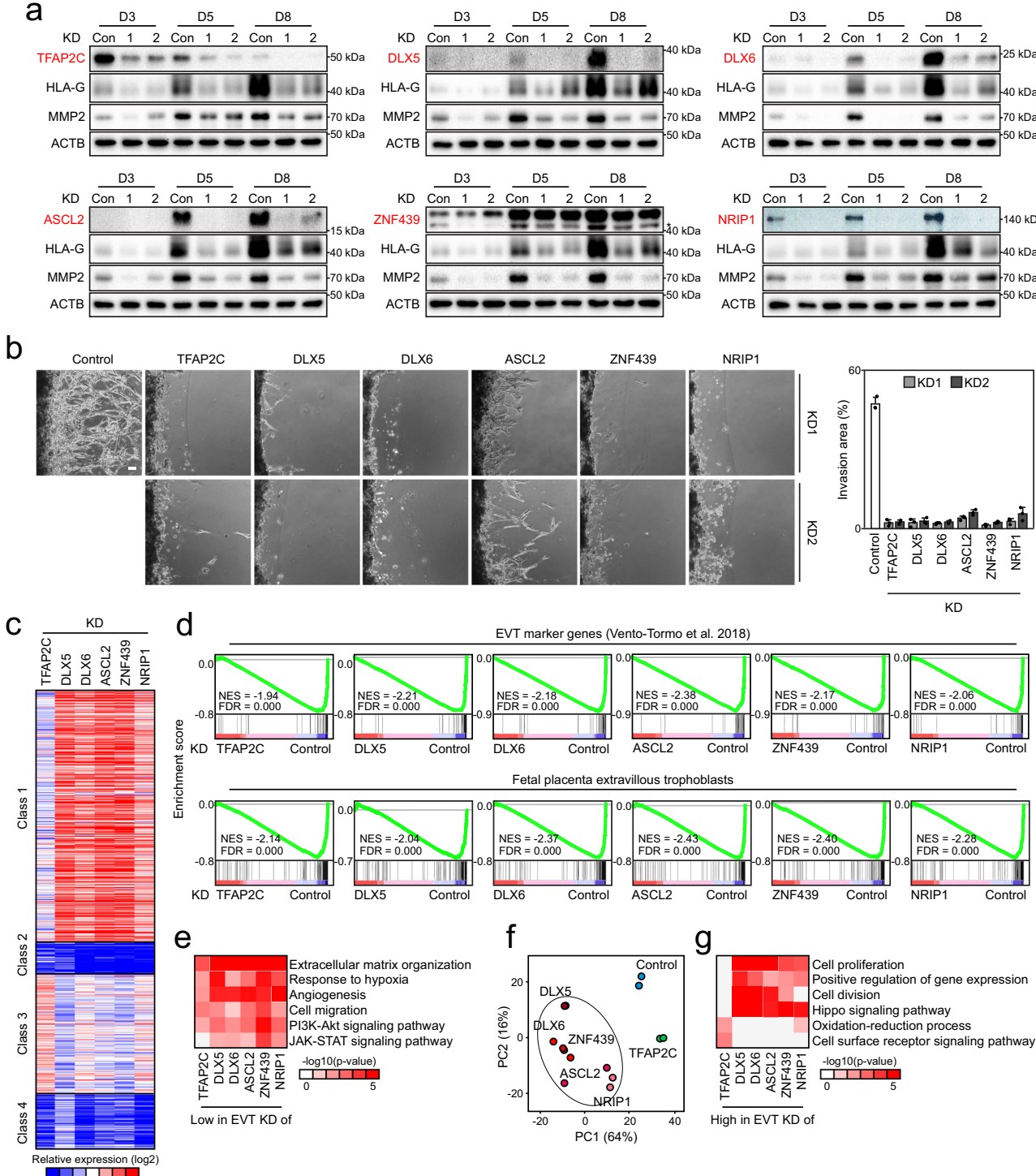

**Fig. 3 | Both early and late-stage TFs are required for EVT differentiation.**
**a** Western blot analysis showing KD efficiency and expression level of EVT marker genes in EVTs following KD of each candidate EVT regulator. EVTs differentiated from TSCs infected with lentivirus expressing non-targeting sequence were used as control. Two independent repeats were carried out, resulting in similar results.
**b** Invasion ability of EVTs after KD of each candidate EVT regulator, compared to control cells. Scale bar: 100 μm. The invasion area was quantified using Image J software. Measurements were obtained from images taken in three independent biological replicates. Error bars: mean ± SD (*n* = 3, independent repeats). **c** Heatmap showing relative expression of class 1–4 genes in EVT factor KD cells compared to non-targeting control. Color represents log$_2$ mRNA expression relative to control KD. **d** GSEA using EVT marker genes defined in scRNA-seq data of the human first-

trimester placenta[19,102] and fetal placenta EVT gene set obtained from the Molecular Signatures Database, comparing EVT factor KD cells with control. NES, normalized enrichment score. FDR, false discovery rate. **e** GO analysis was performed on genes exhibiting lower expression in EVTs after KD of each EVT factor compared to the control. The analysis was conducted using DAVID[88], and *p*-values were calculated. Color representation indicates significance, with -log(p-value). **f** Principal component analysis of EVT factor KD and control cell transcriptome. **g** GO analysis was performed on genes exhibiting higher expression in EVTs after KD of each EVT factor compared to the control. The analysis was conducted using DAVID[88], and *p*-values were calculated. Color representation indicates significance, with -log(*p*-value).

induction of genes active in EVTs (classes 2 and 4) compared to control. To further validate the results, we performed GSEA with the EVT-active gene set defined from the scRNA-seq of the human placenta by another group[19], confirming the lower expression of the EVT signature genes in each TF KD line. The GSEA using another EVT signature gene set obtained from the Molecular Signatures Database[17,48] also generated similar results (Fig. 3d). Consistently, metallopeptidases, such as *MMP2*[49,50] and *ADAM19*[51,52], as well as previously known regulators of invasion, such as *CXCR4*[53,54] and *EGFR-AS1*[55], were downregulated in the EVT factor KD cells. In addition, *NOTUM*, an extracellular Wnt deacylase[56,57], was commonly downregulated in the EVT factor KD cells, suggesting impaired EVT differentiation, as Wnt inhibition is necessary for EVT lineage formation[11,58]. We also observed a decrease in the expression of genes associated with interstitial EVT (iEVT) differentiation, such as *ITGA1*, *PLAC8*, and *SREPINE2*[59], in the EVT factor KD cells. To gain further insight into the function of each TF, we conducted GO analysis with the genes showing lower expression in the KD cells compared to the control cells (Supplementary Data 4). We found that the genes showing lower expression in both early- and late-stage TF KD cells are enriched in terms previously implicated in EVT characteristics and functions, such as extracellular matrix organization, cell migration, extracellular matrix disassembly, Jak-STAT and PI3K-Akt signaling pathways[60], and response to hypoxia[61] (Fig. 3e). We concluded that both TFAP2C and the late-stage TFs are essential for normal EVT function and induction of EVT-active genes.

## TFAP2C KD leads to a unique expression pattern

Although the KD of all individual TFs triggers similar differentiation defects, we surmised that there might be functional differences between early- and late-stage TFs. As shown in Fig. 3c, we found that TFAP2C KD cells showed distinct gene expression patterns among the tested TFs. While KD of late-stage TFs showed improper downregulation of class 1 (TSC-active) genes, overall expression of the class 1 genes in TFAP2C KD cells was even lower than that in the control cells. This implies that early- and late-stage TFs may have unique roles in regulating class 1 genes. Since TFAP2C is significantly expressed in self-renewing TSCs, TFAP2C may act as an activator of class 1 genes, and it may play important roles in not only EVT differentiation but also TSC self-renewal as recently suggested[14,44]. On the other hand, KD of the late-stage TFs resulted in failed downregulation of class 1 genes, implying that the repression of TSC-active genes does not occur efficiently when these factors are knocked down. This observation was further supported by the results of GSEA using the CT marker genes defined from the placenta scRNA-seq study[19] (Supplementary Fig. 3g). Similarly, several genes associated with cell cycle regulation, such as *CDK1* and *CCND1*, along with the factors involved in cell proliferation, including *MYBL2*[62] and *TOP2A* (DNA topoisomerase 2-alpha, which plays a role in DNA replication and cell division), were significantly upregulated in the late-stage TF KD cells. Moreover, the late-stage TF KD cells exhibited sustained expression of the genes associated with epithelial cells, including *EPCAM*[63], *CDH1*[64], and *TJP1* (tight junction protein 1). Additionally, the key genes involved in Wnt signaling, such as *AXIN2*, *FZD5*, *LRP5*, and *TCF7L1*[65–67], displayed increased expression in the late-stage TF KD cells, implying that the KD cells still retained a TSC-like gene expression program (Supplementary Data 4). Thus, late-stage TFs might be involved in the repression of TSC-active genes directly or indirectly during differentiation. We also examined the changes in ST marker genes[19] in the KD cells. Consistent with the previous report[16], we observed increased expression of ST lineage genes in ASCL2 KD cells compared to control cells (Supplementary Fig. 3h). Likewise, TFAP2C KD, DLX6 KD, and NRIP1 KD cells exhibited higher expression of ST genes compared to the individual controls. Despite the increased expression of the ST gene set in TFAP2C KD and some late-stage TF-KD cells, the overall differences in the expression between TFAP2C KD and late-stage TF KD cells were confirmed by PCA

(Fig. 3f). The late-stage TF KD cells were clustered together but separated from the TFAP2C KD or control cell clusters.

To further understand the TF-specific differences in gene regulation, we conducted GO analysis with the genes showing higher expression in the KD cells than in control cells (Supplementary Data 4). Consistent with Fig. 3c, the genes that failed to be repressed upon KD of late-stage TFs were enriched in cell proliferation and Hippo signaling pathway, the terms associated with proliferation of TSC and inhibition of differentiation[12,37]. However, this was not the case in TFAP2C KD cells (Fig. 3g). We concluded that while both TFAP2C and late-stage TFs are commonly required for proper EVT differentiation and the activation of EVT-active genes, they also play unique roles during EVT differentiation, likely in a differentiation stage-specific manner.

## Critical role of early TFAP2C expression in EVT differentiation

TFAP2C shows a unique expression pattern during EVT differentiation, peaking early and then decreasing (Fig. 2c). To gain further insight into the role of this TF, and to test the significance of its expression dynamics, we employed both loss- and gain-of-function studies (Fig. 4a). First, to test if TFAP2C plays distinct roles depending on the stage of differentiation, we performed KD of TFAP2C at different time points during differentiation. Similar to the previous results, KD of TFAP2C at the early stage (KD at day −1, KD-Early) resulted in impaired EVT differentiation evidenced by defects in morphological changes, EVT marker gene activation, and invasion ability. Intriguingly, we did not detect such defects upon KD of TFAP2C beginning at day 3 (KD-Mid) or 5 (KD-Late) (Fig. 4b–d and Supplementary Fig. 4a), which was verified with multiple EVT marker genes (Supplementary Fig. 4b).

As TFAP2C is essential for TSC maintenance and the KD of TFAP2C in TSCs induces apoptosis after four days[68], it is possible that TFAP2C KD in TSCs could impact downstream cellular physiology, thereby impeding EVT formation. To differentiate between the effects of TFAP2C KD in self-renewing TSCs and during differentiation, we performed additional KD experiments at multiple time points during the early stage of EVT differentiation. As shown in Supplementary Fig. 4c, the cells treated with lentivirus targeting TFAP2C on days −1, 0, and 1 exhibited similar impairments in the induction of EVT marker genes. However, TFAP2C KD on days 2 and 3 did not lead to defects in EVT marker gene induction. Additionally, we observed that apoptosis was not significantly induced in TFAP2C KD cells after 18 h of lentivirus infection, which aligns with the timing of EVT differentiation initiation (Supplementary Fig. 4d). These findings indicate that the depletion of TFAP2C in TSCs may not be the primary cause of defects in EVT differentiation. Instead, TFAP2C plays a crucial role during the early stage of EVT differentiation. In contrast, the depletion of late-stage TFs on days 3 and 5 hindered the induction of EVT marker genes, suggesting that these late-stage TFs are necessary for activating EVT-active genes during the late stage of differentiation (Supplementary Fig. 4e).

Next, we tested whether the downregulation of TFAP2C during the late stage of differentiation is also important for EVT differentiation. To do so, we maintained TFAP2C expression beyond the early stage of differentiation. We over-expressed TFAP2C via a doxycycline (Dox)-inducible system (see methods section and Supplementary Fig. 4f) on day 2 of differentiation (TFAP2C OE), when the endogenous TFAP2C level peaks and starts decreasing (Figs. 2c and 4a). Although the ectopic expression of TFAP2C was not sustained till day 8, we observed that ectopic TFAP2C induces morphological abnormalities and defects in TSC marker repression as well as EVT marker activation (Fig. 4e–g, and Supplementary Fig. 4g–j). Accordingly, transcriptomic analysis revealed that EVT-active genes (classes 2 and 4) are not adequately upregulated, while the expression of TSC-active genes (class 1) is not downregulated normally in the TFAP2C OE cells (Fig. 4h). This

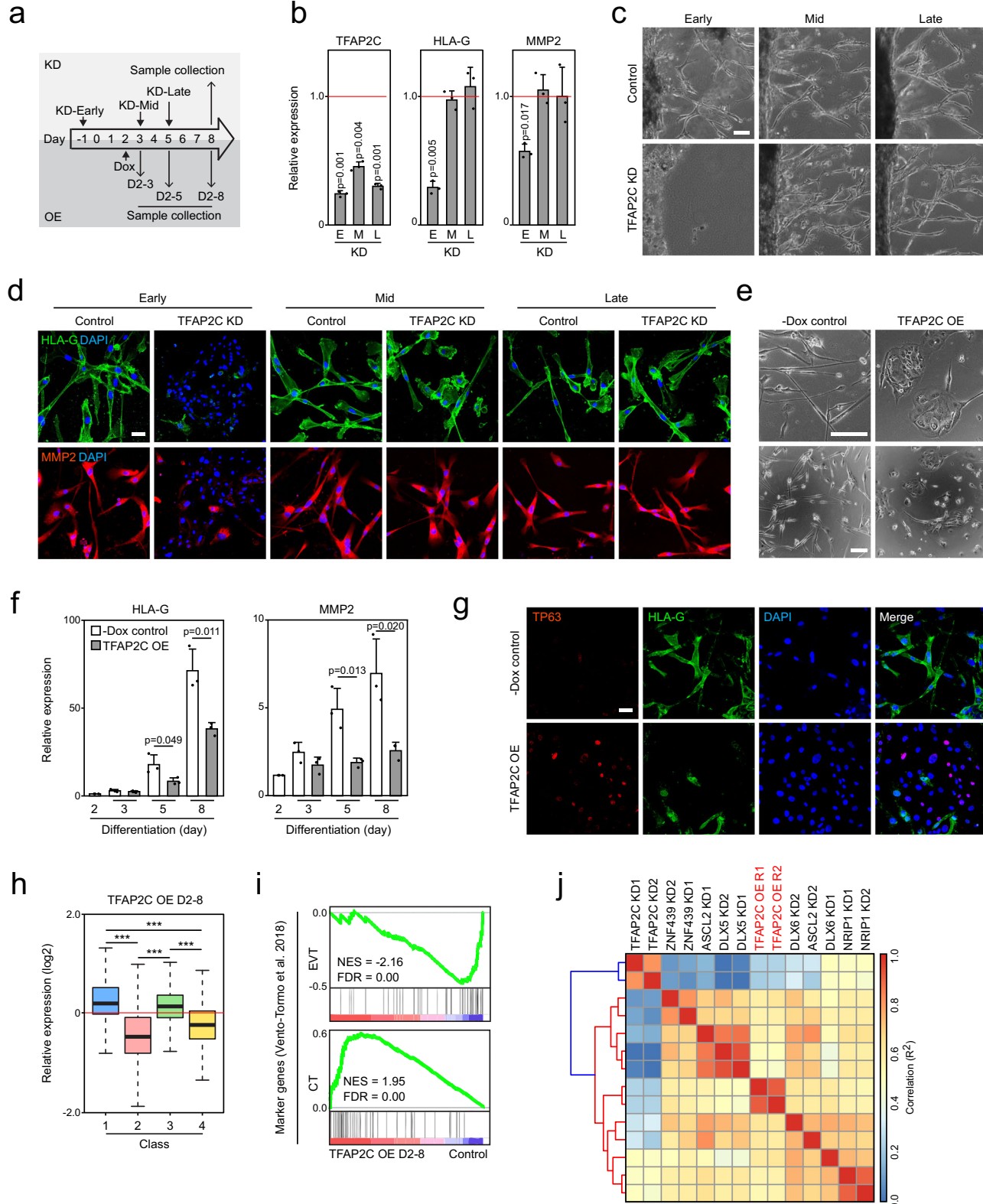

observation is further substantiated by GSEA results employing previously defined EVT and CT gene sets[19] (Fig. 4i). Interestingly, the pattern of expression of classes 1–4 genes in the TFAP2C OE cells was most similar to the pattern in cells lacking late-stage TFs cells (Fig. 4j), suggesting that prolonged expression of TFAP2C has a similar effect to loss of late-stage TFs. In conclusion, our data reveal that an optimal level and timing of TFAP2C expression are critical for proper EVT differentiation.

## TFAP2C primes late-stage TFs and EVT-active genes

Our data thus far show that, despite its requirement for the activation of EVT-active genes, TFAP2C is primarily essential during the early stage of differentiation when EVT-specific genes are not yet transcriptionally active. Moreover, TFAP2C needs to be downregulated during the late stage of differentiation for proper induction of EVT-active genes. To investigate the mechanistic basis of these effects, we examined the relationship of TFAP2C to DLX6, one of the late-stage

**Fig. 4 | Stage-specific regulation of TFAP2C during EVT differentiation.**
**a** Experimental design for time-course TFAP2C KD (top) and OE (bottom).
**b** Relative expression of TFAP2C and EVT marker genes in EVTs after KD of TFAP2C at different time points. E: KD day −1–8, M: KD day 3–8, L: KD day 5–8. Error bars: mean ± SD ($n = 3$, independent repeats). Significance by two-sided Student's $t$-test.
**c** Invasion ability of EVTs after KD of TFAP2C at different time points and individual control cells. Early: KD day −1–8, Mid: KD day 3–8, Late: KD day 5–8. Scale bar: 100 μm. **d** Immunofluorescence of EVT marker genes in EVTs after TFAP2C KD at different time points. Scale bar: 100 μm. Two independent repeats were carried out, resulting in similar results. **e** Brightfield images of EVTs overexpressing TFAP2C (Dox-treated day 2–8) and control. Scale bars: 100 μm. Two independent repeats were carried out, resulting in similar results. **f** Relative expression of EVT markers in TFAP2C OE compared to each control. Dox was treated on day 2, and cells were collected on days 3, 5, and 8. Error bars: mean ± SD ($n = 3$, independent repeats). Significance by two-sided Student's $t$-test. **g** Immunofluorescence of HLA-G and TP63 in TFAP2C OE (day 2–8) and control. Scale bar: 100 μm. Two independent repeats were carried out, resulting in similar results. **h** Box plot illustrates class 1–4 gene expression in TFAP2C OE (day 2–8) compared to control (***$p < 2.2e$-16, two-tailed Wilcoxon rank-sum test). Each group has two independent RNA-seq results. The median (black bar) and box (25th to 75th percentiles) represent data distribution, while whiskers denote the minimum to maximum values. **i**, GSEA of TFAP2C OE and control using the EVT and CT marker gene sets defined in scRNA-seq data of human first-trimester placenta[19,102]. NES normalized enrichment score, FDR false discovery rate. **j** Correlation heatmap showing comparisons of TFAP2C OE with EVTs after KD of individual EVT regulators. Spearman correlations were calculated using $\log_2$-fold changes of the class 1–4 genes.

TFs, using time-course ChIP-seq (Fig. 5a). Similar to its expression pattern during EVT differentiation (Fig. 2c and Supplementary Fig. 5a, b), TFAP2C bound to its targets in self-renewing TSCs and cells in early-stage day 2, followed by decreased occupancy as differentiation progressed. On the other hand, we detected DLX6 target occupancy starting at day 5 of differentiation (Fig. 5b and Supplementary Fig. 5c, d). As expected, the TFAP2C motif was the most enriched motif in the TFAP2C binding loci (Supplementary Fig. 5e), and the DLX3 motif, which is similar to the DLX5 or DLX6 motif (Supplementary Fig. 5f), was enriched in the DLX6 target loci (Fig. 5c). Interestingly, the TFAP2C motif was also significantly enriched in DLX6 target loci, suggesting common targets for TFAPC2 and the late-stage TF DLX6 (Fig. 5c).

The overlapping targets between TFAP2C and DLX6 (Fig. 5d and Supplementary Fig. 5g) suggested a potential regulatory relationship between these factors, and perhaps between TFAP2C and other late-acting genes. To further explore this relationship, we conducted a differential binding analysis comparing target loci of TFAP2C in TSCs and DLX6 in EVTs. This analysis allowed us to identify three distinct groups of loci: unique target loci of TFAP2C in TSCs (Group 1, G1), TFAP2C-DLX6 common target loci (Group 2, G2), and unique target loci of DLX6 in EVTs (Group 3, G3; Supplementary Fig. 5h). We then plotted the ChIP-seq peaks of TFAP2C and DLX6 according to the differentiation stage and target groups (Fig. 5e, f). Each row in the plot represents the same genomic locus, allowing for observing the changes in TFAP2C and DLX6 binding patterns at different time points. As anticipated, G1 and G3 peaks are associated with TSC- and EVT-active genes, respectively (Supplementary Data 5). Intriguingly, many EVT-active genes were also found in G2 loci (Supplementary Data 5), suggesting that, at different differentiation stages, both TFAP2C and DLX6 occupy regulatory elements of the common EVT-active genes. We observed that matrix metalloproteinases, specifically *MMP2* and *MMP9*, which are essential for invasion into the extracellular matrix, are associated with G2 loci. Additionally, we identified master regulators of epithelial-to-mesenchymal transition (EMT), namely *SNAI1* and *TWIST1*[69], as targets of both TFAP2C and DLX6. This suggests that TFAP2C and DLX6 play crucial roles in controlling the genes responsible for the invasive function of EVTs. Notably, we also noticed that the angiogenic factor *VEGF* and its receptors *KDR* and *FLT1*[70] are occupied by both TFAP2C and DLX6, suggesting that both early- and late-stage TFs are involved in the activation of genes associated with spiral artery remodeling, a critical function of EVT. Focusing on genes in classes 2 and 4, as defined in Fig. 1d, we also confirmed TFAP2C occupancy at regulatory loci near the EVT-active genes in both TSC and EVT D2. These loci are later occupied by DLX6 in EVT D8 (Fig. 5g, h). Interestingly, we also observed that the binding of TFAP2C at G3 loci increases during early differentiation (from day 0 to 2) (Fig. 5e, i). While some EVT-active genes are pre-occupied by TFAP2C in TSCs (G2), others show increased occupancy of TFAP2C in EVT D2 (G3). TFAP2C's occupancy on these EVT-active genes during the late stage of differentiation was not obviously detectable as the level of TFAP2C greatly diminished. Since EVT-active genes are not significantly transcribed in TSCs and EVT D2 cells, our results suggest that TFAP2C primes EVT-active genes during the early stage of differentiation.

Given that the EVT factor KD cells exhibited similar impairments in the induction of EVT-active genes, we further investigated the relationships between TFAP2C and the late-stage TFs by conducting an integrative analysis of TFAP2C/DLX6 targets and the RNA-seq data from the individual EVT TFs KD cells. As shown in Supplementary Fig. 5i, approximately 75–84% of the DEGs upon each KD were directly bound by TFAP2C, DLX6, or both. Interestingly, around 49% and 60% of the upregulated and downregulated genes in each KD were common targets of TFAP2C and DLX6 (G2), suggesting functional similarities among the late-stage TFs and potential collaborative works between other the late-stage TFs and TFAP2C.

### Potential pioneering activity of TFAP2C on EVT-active genes

The pre-occupancy of TFAP2C for the future activation of EVT-active genes is reminiscent of pioneer factor activity[71,72] (Fig. 5d, g, h). Indeed, the patterns of TFAP2C and DLX6 binding to target loci near late-stage TFs and EVT markers (*HLA-G* and *MMP2*) are consistent with this role (Fig. 6a and Supplementary Fig. 6a, b). Our data show that TFAP2C occupies cis-regulatory elements of the late-stage TFs and EVT markers during the early stage of differentiation, and DLX6 binds to the loci initially primed by TFAP2C during the late stage (Fig. 6a and Supplementary Fig. 6a, b). To test whether TFAP2C functions as a pioneer factor, we assessed TFAP2C target loci in the context of the genome-wide chromatin accessibility landscape monitored by ATAC-seq. We also mapped H3K4me3 signatures, a hallmark of active promoters, in TSCs and EVTs, to monitor the activity of the TFAP2C target genes. We found that TFAP2C occupies both open and closed chromatin regions near late-stage TFs and EVT markers (Fig. 6a and Supplementary Fig. 6a, b). Notably, several genomic loci bound by TFAP2C were near-closed chromatin in TSCs but showed significantly increased accessibility in EVTs. These TFAP2C-primed regions also exhibited increased H3K27ac signals and increased transcriptional activity of the associated genes in mature EVTs.

To further investigate the potential pioneer factor activity of TFAP2C during EVT specification, we compared TFAP2C binding loci and ATAC-seq signals in TSCs using the MAnorm[73]. We observed that TFAP2C binding loci with low ATAC-seq signals (Fig. 6b, indicated by a black bar), which are primarily found in enhancer regions (Supplementary Fig. 6c). The genes associated with these loci are enriched in the terms, such as JAK-STAT and PI3K-Akt, previously implicated in EVT differentiation (Fig. 6c). The vasculature development-related terms were also consistent with the spiral artery remodeling function of EVTs (Fig. 6c). These results indicate that TFAP2C binding loci with low accessibility are associated with EVT-active genes. To validate the binding of TFAP2C on genomic loci with low ATAC-seq signals near EVT-active genes in TSCs, we conducted an ATAC-qPCR analysis. We examined the loci with varying degrees of openness in TSCs, including those with very weak ATAC-seq signals near olfactory receptor genes (inactive), weak ATAC-seq signals near EVT-active genes not active in

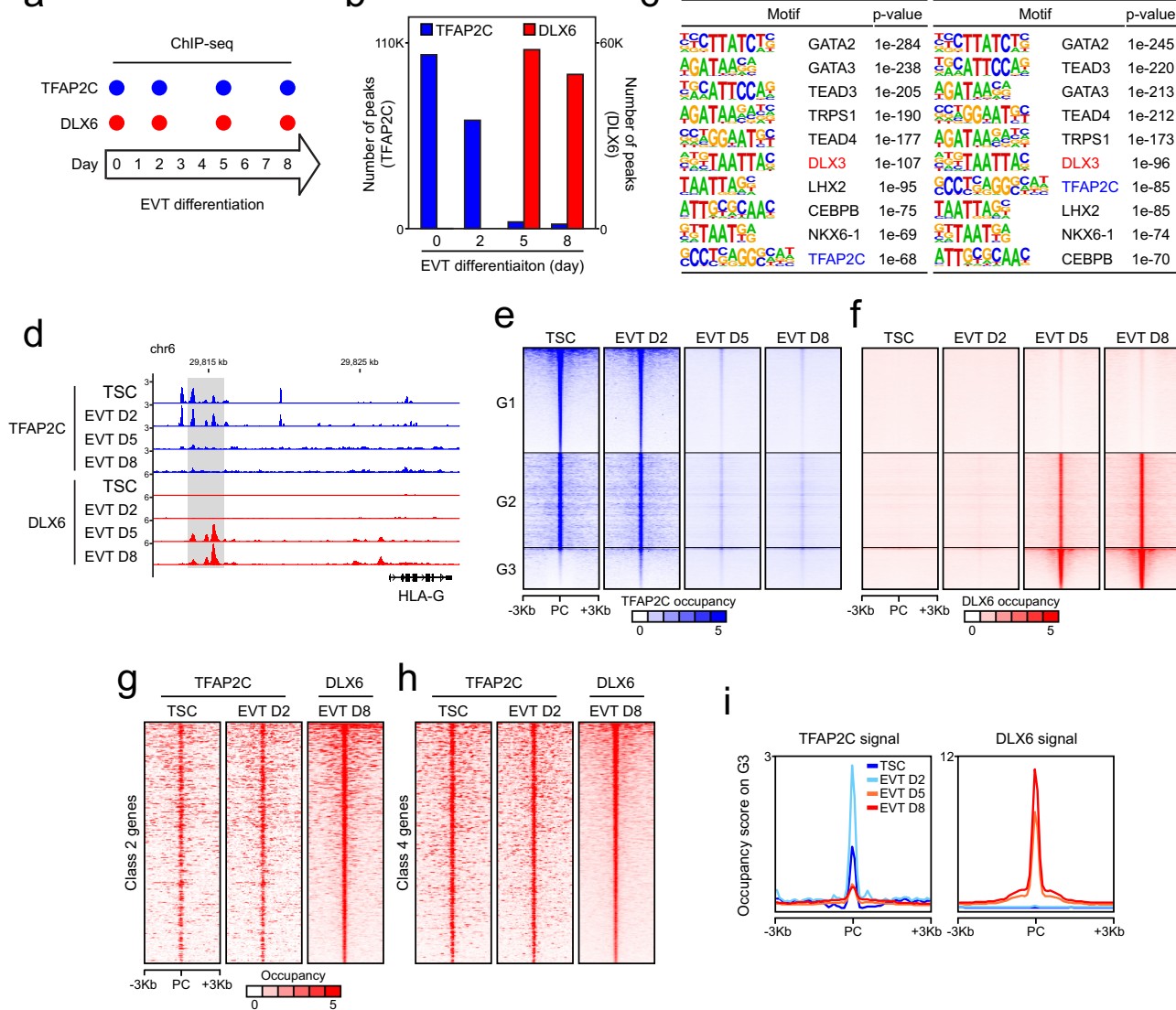

**Fig. 5 | TFAP2C primes late-stage TFs and EVT-active genes at the early-stage of differentiation. a** Experimental design for time-course ChIP-seq of TFAP2C and DLX6 during EVT differentiation. **b** Number of TFAP2C and DLX6 ChIP-seq peaks in TSCs and differentiating cells to EVT on days 2, 5, and 8. **c** Enriched motifs in DLX6 ChIP-seq peaks in EVT day 5 (left) and 8 (right). Motifs were identified using the findMotifsGenome.pl module under HOMER (v4.11), and *p*-values were calculated using the module's default settings. **d** Gene track view of TFAP2C (blue) and DLX6 (red) signals in TSCs, EVT D2, EVT D5, and EVT D8 at the locus near HLA-G. Heatmaps of (**e**) TFAP2C and (**f**) DLX6 signals on TFAP2C in TSCs unique target loci (G1), TFAP2C/DLX6-common target loci (G2), and DLX6 in EVT D8 unique target loci (G3) in TSCs, EVT D2, EVT D5, and EVT D8. PC, peak center. Heatmaps of TFAP2C in TSCs and EVT D2, and DLX6 in EVT D8 near the genes in (**g**) class 2 and (**h**) class 4. **i** Occupancy score of TFAP2C (left) and DLX6 (right) on DLX6 in EVT D8 unique target loci (G3) in TSCs, EVT D2, EVT D5, and EVT D8.

TSCs (EVT-active), strong ATAC-seq signals near TSC marker genes (TSC-active), and ribosomal protein-coding genes (TSC/EVT-active) (Supplementary Fig. 6d). The ATAC-qPCR results in Fig. 6d confirmed that the genomic loci near EVT-active genes bound by TFAP2C in TSCs (red bars) exhibit overall lower openness compared to the TSC-active loci (blue bars) and TSC/EVT-active loci (green bars), while still being more open than closed loci near olfactory receptor genes (inactive, black bars).

Consistently, the ATAC-seq signals at the TFAP2C unique loci were significantly increased in EVTs, along with increased enhancer signals (Fig. 6e, f). Thus, TFAP2C target loci with low accessibility during the early EVT differentiation become more accessible as differentiation proceeds, further supporting the idea of pioneering activity for TFAP2C. As KD of TFAP2C at the early differentiation stage inhibits EVT differentiation (Fig. 3), TFAP2C-dependent priming of EVT-active genes appears essential for EVT differentiation. Together, these

results support the model that TFAP2C exhibits potential pioneer factor activity for specific targets during EVT differentiation.

## Late-stage TFs form a transcriptional regulatory network

We then sought to determine the mechanisms by which late-stage TFs regulate EVT differentiation. DLX6 occupancy on the regulatory elements of EVT-active genes suggests direct transcriptional regulation. To further investigate the function of late-stage TFs, we mapped their genomic targets by ChIP-seq (DLX5 and ASCL2) and bioChIP-seq (NRIP1)[74] in mature EVT D8 cells (Supplementary Fig. 4f). We could not detect significant ZNF439 binding peaks although we detected both nuclear and cytoplasmic localization of ZNF439 in EVT D8 cells (Supplementary Fig. 7a). While the number of binding loci varied (3413–86,735) depending on the TF (Supplementary Fig. 7b), motif analysis revealed that DLX and ASCL2 motifs are significantly enriched in DLX5 and ASCL2 target loci, respectively (Supplementary Fig. 7c).

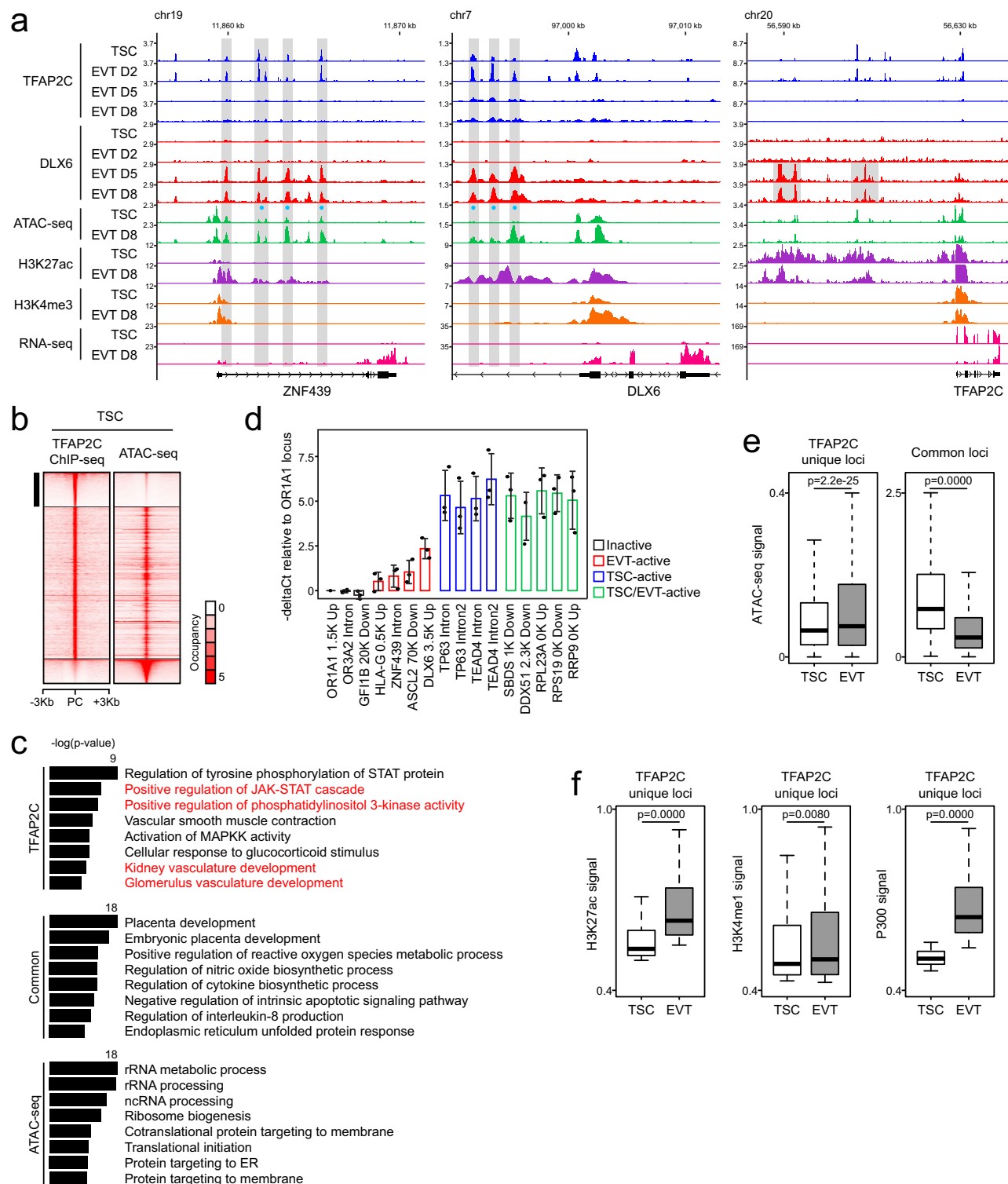

Similar to the motif analysis results obtained for DLX6 (Fig. 5c), we observed significant enrichment of the TFAP2C motif in binding loci of both DLX5 and ASCL2, implying a similar target priming mechanism by TFAP2C. An NRIP1 binding motif has not been reported, but we found enriched motifs for SP and KLF family proteins and TSC-related factors, such as TEAD3 and FOSL2 within the NRIP1 target loci (Supplementary Fig. 7c). Thus, we were able to map target occupancy patterns of most of the late TFs and confirm the validity of our results by motif analysis.

As cell-type specific key TFs often auto-regulate their own gene expression[29,75], we asked whether the late-stage TFs also bind to their own regulatory regions. As shown in Fig. 7a and Supplementary Fig. 7d, DLX5, DLX6, and ASCL2 (except for NRIP1) occupied regulatory regions of late-stage TFs and EVT markers, forming auto-regulatory loops. Notably, DLX6, DLX5, and ASCL2 bound the regulatory elements of TFAP2C, suggesting potential roles of late-stage TFs in the regulation of TFAP2C expression during late stage of differentiation (Figs. 6a and 7a).

**Fig. 6 | TFAP2C acts as a pioneer factor for late-stage TFs and EVT-active genes.** **a** Gene track view displaying TFAP2C and DLX6 signals in TSCs and differentiating cells to EVT on days 2, 5, and 8, along with ATAC-seq, H3K27ac, H3K4me3 signals in TSCs and EVTs. mRNA expression levels are shown near EVT factors. **b** Heatmaps showing TFAP2C ChIP-seq unique, common, and ATAC-seq unique loci in TSCs. TFAP2C ChIP-seq unique loci are indicated by a black bar. PC peak center. **c** Enriched GO terms of the top 3000 TFAP2C unique, common, or ATAC-seq unique peaks were determined using GREAT software[100]. *p*-values were calculated within the software. **d** ATAC-qPCR results validating TFAP2C binding to loci near EVT-active genes with low ATAC-seq signal. To account for primer variability, the Ct value was first normalized by subtracting the Ct value of sheared genomic DNA, resulting in a normalized Ct value. The deltaCt value was then calculated by subtracting the normalized Ct value of a genomic locus near an olfactory receptor gene (OR1A1), which exhibits very low ATAC-seq signal and low transcription levels in

TSCs. The *y*-axis represents negative deltaCt values, indicating that an increase in value corresponds to increased openness. Error bars: mean ± SD (*n* = 3, independent repeats). **e** ATAC-seq signals at TFAP2C unique and common loci identified by comparing ChIP-seq results of TFAP2C in TSCs and DLX6 in EVTs. Box plots represent the median (black bar) and interquartile range (box boundaries), with whiskers from minimum to maximum values. Significance by two-tailed Wilcoxon rank-sum test. Analysis involved ATAC-seq signals on all TFAP2C unique or common loci in TSCs and EVTs. **f** Enhancer signals at TFAP2C unique loci in TSCs and EVTs. Box plots represent the median (black bar) and interquartile range (box boundaries), with whiskers from minimum to maximum values. Significance by two-tailed Wilcoxon rank-sum test. Statistical analysis involved signals from enhancer ChIP-seq results on all specified TFAP2C unique loci in both TSCs and EVTs.

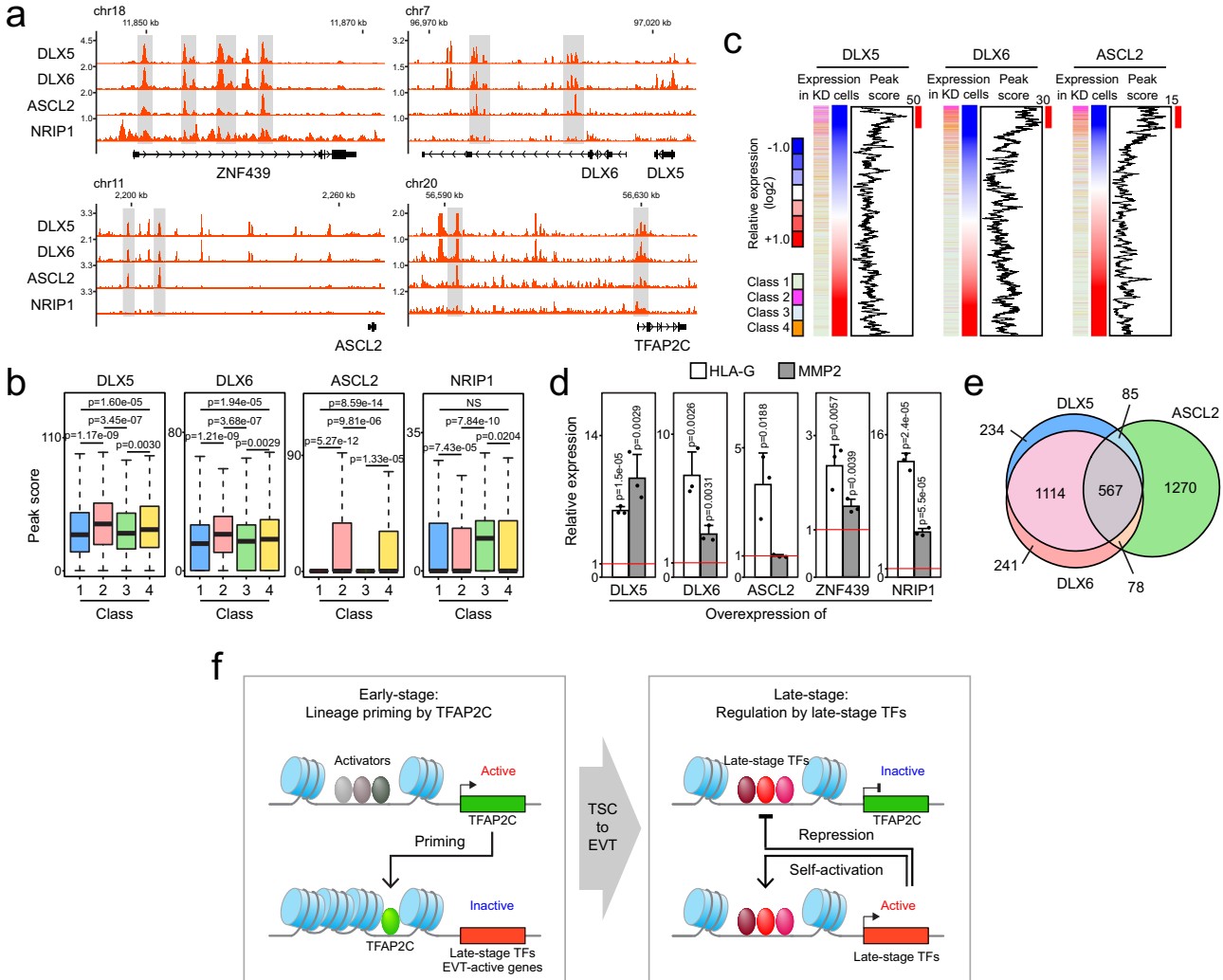

**Fig. 7 | Late-stage TFs collaborate in regulating the EVT-specific gene expression program.** **a** Gene track view of late-stage TF binding loci in mature EVTs (EVT D8), near EVT factor loci. **b** Peak scores of late-stage TFs on class 1–4 genes. Box plots represent the median (black bar) and interquartile range (box boundaries), with whiskers indicating minimum to maximum values. Statistical significance by two-tailed Wilcoxon rank-sum test (NS not significant). All peak scores identified from ChIP-seq results were employed for statistical analysis. **c** Heatmaps of class 1–4 gene expression in EVTs after KD of individual late-stage TFs. Genes were ordered based on relative gene expression levels in the KD cells compared to the

control (from lowest to highest). Occupancy signals plotted as a moving average (window size: 50). Red bars indicate enriched signals. **d** Relative mRNA expression of EVT marker genes in TSCs overexpressing individual late-stage TFs, induced by administrating Dox for 6 days. Error bars: mean ± SD (*n* = 3, independent repeats). Statistical significance was determined by two-sided Student's *t*-test. **e** Venn diagram illustrating overlap of top 2000 target genes of DLX5, DLX6, and ASCL2. **f** Model: TFAP2C primes late-stage TFs during early stage of differentiation, while late-stage TFs activate themselves and repress TFAP2C, forming a transcriptional regulatory circuit to ensure proper EVT differentiation.

To obtain insights into the function of the late-stage TFs on their target genes, we examined TF occupancy on the four gene classes we defined earlier (Fig. 1d). As NRIP1 showed unique global occupancy and target gene regulation patterns (Fig. 7a and Supplementary Fig. 7d, e), we focused on DLX5, DLX6, and ASCL2. As shown in Fig. 7b, DLX5, DLX6, and ASCL2 showed significantly higher occupancy signals on the EVT-active genes (classes 2 and 4) compared to genes in classes 1 and 3, consistent with a role in activation of class 2 and 4 genes. To test late-stage TFs' role as a direct activator of EVT-active genes, we performed combined analysis of occupancy and changes in expression upon KD of the late-stage TFs on class 1–4 genes (Fig. 7c). We observed increased occupancy patterns for the late-stage TFs at genes down-regulated upon KD of each TF (activated genes by the TFs). Moreover, a majority of affected genes were from classes 2 and 4 (Fig. 7c). To further confirm this, we generated individual inducible OE cell lines of the late-stage TFs (Supplementary Fig. 7f) and performed OE of the late-stage TFs in self-renewing TSCs (Supplementary Fig. 7g). We observed the activation of EVT markers *HLA-G* and *MMP2* (Fig. 7d).

As key TFs co-regulate cell-type specific genes by forming transcriptional regulatory networks[29,74,76–78], we monitored the top 2000 target genes of DLX5, DLX6, and ASCL2 and confirmed that they indeed share many common targets (Fig. 7e). GO analysis of the common target genes revealed their enrichment in EVT function-related terms, such as Rho GTPase cycle, regulation of cell/focal adhesion, and tube morphogenesis (Supplementary Fig. 7h). Similar to our analysis shown in Fig. 7c, the common target genes were also the most significantly affected genes (classes 2 and 4 genes) upon individual KD of all late-stage TFs including ZNF439 and NRIP1 (Supplementary Fig. 7i). Altogether, we reveal that the late-stage TFs we identified form a transcriptional regulatory network and positively control EVT-active genes during EVT specification.

## Discussion

Despite the critical roles of EVTs during human pregnancy, molecular-level understanding of normal and defective EVT differentiation has been elusive. In the present study, using in vitro model systems, we identified key early and late regulators sequentially functioning during EVT differentiation. The early factor TFAP2C exhibits potential pioneer factor activities, priming multiple EVT-active genes including late-stage TFs. Late-stage TFs act together to from a regulatory network to activate EVT-specific gene expression programs, while potentially suppressing an early-stage regulator, TFAP2C to ensure proper EVT differentiation (Fig. 7f). To our knowledge, this is the first report revealing how multiple stage-specific TFs are intertwined to control EVT differentiation. Our findings will serve as important frameworks to understand the EVT differentiation process during normal and abnormal placentation.

In this study, we employed a well-established protocol to differentiate TSCs into EVTs, resulting in a transcriptome profile similar to that of EVTs derived from the human placenta[11]. However, it remains unclear whether this in vitro differentiation could precisely recapitulate the differentiation process of CTs into EVTs within the human placenta. Recently, Arutyunyan et al. conducted spatial transcriptomics and single-nucleus RNA sequencing (snRNA-seq) of human placenta at 8–9 post-conceptional weeks, identifying marker genes for various trophoblast populations[59]. Considering the findings in vivo, we performed additional analysis of our time-course EVT differentiation data. We found that the expression levels of markers associated with villous cytotrophoblasts (VCTs), the progenitor populations, was observed to be active in undifferentiated TSCs and gradually decreased as in vitro EVT differentiation progressed. The marker genes for early-stage placental EVT differentiation, such as VCT proliferative cells (VTC-p) and CT cell column VCT (VCT-CCC), exhibited activation during the early stage of in vitro EVT differentiation. Additionally, the markers for intermediate-stage placental EVT

populations, EVT-1 and EVT-2, were upregulated during days 2–4 and 5–8 of EVT differentiation, mirroring the in vivo process. Similar to the previous findings[59], our analysis revealed the activation of iEVT markers during in vitro differentiated EVTs, whereas the markers specific to endovascular EVTs (eEVTs) and deep invasive placental bed giant cells (GCs) were not activated. The observed expression patterns of the marker genes imply that in vitro EVTs mirror the cells found in the human placenta, with a predominant bias towards the iEVT population. While the iEVTs differentiated from the human TSC model may not fully replicate their in vivo counterparts in terms of maturity and encompassing all cell types[59], it can serve as a valuable tool for investigating early iEVT differentiation and exploring the molecular mechanisms underlying this process.

Regarding the expression patterns of the identified EVT TFs in vitro, the scRNA-seq and snRNA-seq results obtained from human placenta confirmed a lower expression level of TFAP2C in EVTs compared to CTs, with some variation in the extent of reduction. However, the expression levels of DLX5/6 and ZNF439 are somewhat higher in CTs and EVTs, as reported in previous studies[11,19,46,59], despite their observed inactivity in TSC models, aligning with findings from this study and ref. 11. Although certain late-stage TFs in CTs exhibited expression levels comparable to those in EVTs, we observed high expression of all late-stage TFs in the placental EVTs. We also confirmed high expression patterns of all late-stage TFs across multiple established TSC lines (Fig. 2e and Supplementary Fig. 2b–d). While in vitro systems may not perfectly replicate in vivo expression patterns, they offer valuable insights into the key regulators and regulatory mechanisms involved in biological processes. For instance, studying somatic cell reprogramming in vitro enhanced our understanding of drivers, essential factors, signaling pathways, and molecular mechanisms governing cell fate control, despite the process not occurring naturally in vivo. Although the expression patterns of some EVT regulators in TSC models may not entirely mirror those observed in human placentas, investigating these EVT regulators in TSC models contributes to a better understanding of their critical roles and action mechanisms, particularly considering their high expression in EVTs within human placentas.

Our recent findings and those of others defined critical roles for TFAP2C in both self-renewal[15,44,68]. While our current study focuses on TFAP2C's role in EVT differentiation as an early-stage regulator, recent reports using CRISPR-Cas9 knockout or siRNA-based KD studies suggested that TFAP2C also controls human TSC self-renewal[15,44]. Interestingly, a prior study in mouse TSCs suggested that Tfap2c makes self-renewal vs. differentiation decision through its expression levels and interaction partner Elf5[41]. As we revealed that dynamically changed but precisely controlled levels of TFAP2C are crucial for human EVT differentiation, it will be of great interest which partner proteins TFAP2C interacts with during the progression of EVT differentiation. We assume that TFAP2C have unique sets of interaction partners depending on cellular status. Given that the level of TFAP2C also moderately increases during the early stages of ST differentiation followed by rapid downregulation, it will be important to test if TFAP2C is similarly required for ST differentiation as a general regulator of human trophoblasts lineage specification.

In our recent study, we have demonstrated that TFAP2C exhibits dual roles, functioning both as an activator and potentially as a repressor on its targets in self-renewing TSCs[68]. However, in the specific context of EVT-active genes, we have ruled out the possibility of TFAP2C acting as a repressor. This conclusion is supported by our findings that KD of TFAP2C leads to insufficient upregulation of EVT-active genes during EVT differentiation, as depicted in Fig. 3a, c, d and Supplementary Fig. 3a, c. The observed discrepancy may stem from context-specific or target-specific roles of TFAP2C. Interestingly, we have also noted that TFAP2C KD results in increased expression of marker genes associated with STs (Supplementary Figs. 3h and 4b).

Although the direct or indirect nature of this regulation remains unclear, these results suggest potential repressor functions of TFAP2C on other classes of target genes during EVT differentiation. Consequently, our findings indicate that TFAP2C does not act as a repressor, at least on EVT-active genes, during EVT differentiation.

Furthermore, we have discovered that TFAP2C binds to cis-regulatory elements characterized by low openness, which are associated with EVT-active genes in TSCs. As EVT differentiation progresses, we observed increased enhancer signals in proximity to TFAP2C target loci, accompanied by the upregulation of associated genes (Fig. 6a, e and Supplementary Fig. 6a, b). These findings suggest potential pioneer factor activities of TFAP2C on EVT-active genes. Supporting this notion, we have identified the enrichment of the TFAP2C motif in the target loci of DLX6 (Fig. 5c), further bolstering the potential pioneer factor activities of TFAP2C. However, to further support whether TFAP2C functions as a pioneer factor during EVT differentiation, it is crucial to investigate the physical association between TFAP2C and chromatin remodelers or modifiers. While we did not directly assess these interactions, gaining insights into the dynamics of TFAP2C binding, changes in chromatin accessibility, and the presence of specific motifs provide valuable information.

Understanding the varied roles of TFAP2C also has implications for in vitro cell reprogramming as TFAP2C has been used to generate induced TSCs (iTSCs) lines in both human and mouse[43,79,80]. Notably, multiple studies in non-trophoblast contexts have reported that TFAP2C also promotes cell fate transitions. For example, combining TFAP2C with other TFs promoted the conversion of human induced PSCs (iPSCs)[81] or mouse PSCs[82] to primordial germ cell-like cells and TFAP2C also promoted keratinocyte differentiation from human embryonic stem cells (ESCs) via induction of changes in the chromatin landscape[45]. In addition, Tfap2c also facilitated mouse iPSC generation[83]. Although the exact mechanistic bases of TFAP2C function have not been fully elucidated in these contexts, together with our studies, these results suggest that TFAP2C may play a universal role during cell fate conversions.

## Methods

### Cell culture
TSCs derived from human CTs of the first-trimester human placenta (gift of Dr. Takahiro Arima, Tohoku University) were cultured as previously described[11] with modification by communicating with Drs. Takahiro Arima and Hiroaki Okae. TSC were cultures in DMEM/F12 medium (Thermo Fisher Scientific, 11320082) supplemented with 1% ITS-X supplement (Thermo Fisher Scientific, 5150056), 0.5% Penicillin-Streptomycin (Thermo Fisher Scientific, 15140-122), 0.3% BSA (MilliporeSigma, A7906), 0.2% FBS (Gemini Bio-Products, 100106), 0.1 mM β-mercaptoethanol (MilliporeSigma, M3148), 0.5 μM A83-01 (MilliporeSigma, SML0788), 0.5 μM CHIR99021 (Selleck Chemicals, S2924), 0.5 μM SB431542 (STEMCELL Technologies, 72232), 5 μM Y27632 (ROCK inhibitor, Selleck Chemicals, S1049), 0.8 mM VPA (Wako Pure Chemical Corporation, 227-01071), 50 ng/ml EGF (PeproTech, AF-100-15), and 1.5 μg/ml L-ascorbic acid (MilliporeSigma, A8960) on dishes coated with 0.05% iMatrix-511 (Nippi, N-892012). Media were changed every 2 days, and cells were passaged using TrypLE™ Select Enzyme (Thermo Fisher Scientific, A1217701) when the cells were 70–80% confluent at a 1:8–1:10 ratio. Differentiation of TSC into EVT was performed as previously described[11]. For EVT differentiation, cell culture plates were coated with 1 μg/mL Collagen IV (Corning, 354233) at 37 °C for 1 h 30 min. TSC were grown to 70–80% confluency and then single-cell dissociated using TrypLE™ Select Enzyme. The dissociated cells ($1–1.5 \times 10^4$ cells/cm$^2$) were seeded with 2 mL of EVT differentiation medium (DMEM/F12 medium supplemented with 1% ITS-X supplement, 0.5% Penicillin-Streptomycin, 0.3% BSA, 0.1 mM β-mercaptoethanol, 100 ng/ml human neuregulin-1 (NRG1, Cell Signaling Technology, 5218SC), 7.5 μM A83-01, 2.5 μM Y27632, 4% KnockOut™

Serum Replacement (KSR, Thermo Fisher Scientific, 10828028), and 2% Matrigel (Corning, 354234)). At day 3, media were replaced with 2 mL EVT medium with 0.5% Matrigel in the absence of NRG1. At day 6, the media were replaced with 2 mL EVT medium with 0.5% Matrigel in the absence of both NRG1 and KSR. HEK293T cells (ATCC, CRL-3216) and MCF7 cells (ATCC, HTB-22™) were cultured in DMEM (Thermo Fisher Scientific, 11965118) with 10% FBS (Gemini Bio-Products, 100106) and 1x Penicillin-Streptomycin-Glutamine (Thermo Fisher Scientific, 10378016). All cells were incubated at 37 °C and 5% $CO_2$.

### RNA sequencing
Total RNAs were harvested using the RNeasy Plus Mini Kit (Qiagen, 74136) following manufacturer's instructions. Poly-A RNA was isolated from 300 ng of total RNA using NEBNext® Poly(A) mRNA Magnetic Isolation Module (NEB, E7490L), and RNA-seq libraries were prepared using NEBNext® Ultra™ II RNA Library Prep Kit for Illumina® (NEB, E7770L). NEXTflex® ChIP-Seq Barcodes (Bioo Scientific, NOVA-514122) or NEBNext® Multiplex Oligos for Illumina® (NEB, E7600S, E7780S) were used for indexing. RNA-seq libraries were sequenced on the Illumina NovaSeq 6000 platform.

### RNA sequencing analysis
RNA-seq reads were aligned to all known human transcripts (hg38) using salmon (v1.4.0)[84]. Transcripts per million (TPM) value was calculated using R library tximport[85]. The read count of each gene was calculated by R package DESeq2 (v1.30.1)[86] using the median of ratio method, and genes showing $P$-adjusted value < 0.01 and fold change >2 were considered differentially expressed genes (DEGs). Dirichlet Process Gaussian process (DPGP) mixture model[20] was used to cluster DEGs during EVT differentiation. Heatmaps were generated using Java TreeView[87]. GO and pathway enrichment analysis by using DAVID[88] and Metascape[89] and Gene-set enrichment analysis (GSEA)[17] were performed on DEGs and classes 1–4 genes. Spearman correlation[90] of classes 1–4 genes among EVT factor KD and TFAP2C OE cells were calculated using the R package Corrr (Kuhn, M., Jackson, S., & Cimentada, J. (2020). corrr: Correlations in R. R Package version 0.4, 2.), and pheatmap (v1.0.12, Kolde, R. (2019). pheatmap: Pretty Heatmaps. R package version 1.0. 12. CRAN. R-project. org/package = pheatmap.) was used to visualize the results.

### Chromatin immunoprecipitation coupled with next-generation sequencing (ChIP-seq)
Chromatin immunoprecipitation (ChIP) and biotin-mediated ChIP (bioChIP) reactions were performed as previously described[91,92] with minor modifications. Cells were cross-linked with 1% formaldehyde for 7 min at room temperature and then quenched by adding glycine (final concentration 125 mM) for 5 min. The fixed cells were washed three times with PBS and re-suspended in ChIP dilution buffer (0.1% SDS, 1% Triton X-100, 2 mM EDTA, 20 mM Tris-Cl pH 8.1, 150 mM NaCl, and protease inhibitors) and sonicated using a Bioruptor (Diagenode, UCD-200) with a setting of 30 s on and 30 s off for 30 min. The average size of the sheared genomic DNA was 200 bp. The sample was centrifuged at $18,894 \times g$ at 4 °C for 15 min, and the supernatant was collected. The supernatant was pre-cleared with protein A or G agarose bead (Roche, 11134515001 or 11243233001) at 4 °C for 3 h and incubated with native antibody against each target for ChIP or streptavidin beads (Dynabeads MyOne Streptavidin T1, Invitrogen, 65602) for bioChIP at 4 °C overnight. Immunoprecipitated complexes from ChIP with a native antibody were washed with buffer I (0.1% Deoxycholate, 1% Triton X-100, 1 mM EDTA, 50 mM HEPES pH 7.5, 150 mM NaCl) twice, buffer II (0.1% Deoxycholate, 1% Triton X-100, 1 mM EDTA, 50 mM HEPES pH 7.5, 500 mM NaCl), buffer III (250 mM LiCl, 0.5% NP-40, 0.5% Deoxycholate, 1 mM EDTA, 10 mM Tris-Cl pH 8.1), and TE buffer (10 mM Tris-Cl pH 8.1, 1 mM EDTA) twice. Immunoprecipitated complexes from bioChIP were

washed with 2% SDS twice, buffer II, Buffer III, and TE buffer twice. To reverse crosslink protein-DNA complexes, elution buffer (1% SDS, 10 mM EDTA, 50 mM Tris-Cl 8) was added to the washed beads and incubated at 65 °C overnight. After treatment with RNase A (Thermo Scientific, EN0531) for 30 min and proteinase K (NEB, P8107S) for 2 h, DNA was extracted with phenol:choloroform:isoamyl alcohol (Invitrogen, 15593031) and precipitated. The precipitated DNA was resuspended in TE. ChIP- and bioChIP-seq libraries were generated using NEBNext Ultra II DNA Library Prep kit (NEB, E7103L) following the manufacturer's instruction. NEXTflex® ChIP-Seq Barcodes (Bioo Scientific, NOVA-514122) or NEBNext® Multiplex Oligos for Illumina® (NEB, E7600S, E7780S) was used for indexing. The libraries were sequenced on Illumina NovaSeq 6000 platform. The following primary antibodies were used for ChIP-seq: H3K27ac (Active Motif, 39133, 10 µg), H3K4me3 (Active Motif, 39159, 10 µg), DLX5 (Novus Biologicals, NBP1-85793, 20 µL), DLX6 (Proteintech, 23216-1-AP, 20 µL), ASCL2 (Millipore, MAB4418, 20 µL), TFAP2C (Santa Cruz Biotechnology, sc-8977, 10 µg), P300 (abcam, ab10485, 4 µL), MED1 (Bethyl Laboratories, A300-793A, 5 µL), and H3K4me1 (abcam, ab8895, 4 µL).

## ATAC-seq and ATAC-qPCR

ATAC-seq libraries were prepared as previously described[93] using the ATAC-seq kit (Diagenode, C01080002) and 24 UDI for Tagmented libraries set II (Diagenode, C01011036). Briefly, 50,000 nuclei were pelleted and resuspended with tagmentase for 30 min at 37 °C. The tagmented DNA was then purified and used for library generation through PCR amplification. Size selection was conducted using AMPure XP beads (Beckman coulter, A63881), and the purified ATAC-seq libraries were sequenced on the Illumina NovaSeq 6000 platform.

For ATAC-qPCR experiments, the final elution product of the ATAC protocol was diluted 1:80. qPCR was conducted using PerfeCTa SYBR Green FastMix Reaction Mixes (QuantaBio, 95072-012) and a StepOnePlus™ Real-Time PCR System (Thermo Fischer Scientific, brand Applied Biosystems). Genomic DNA was utilized as a control for primer variability. All primer sequences used for qPCR are in Supplementary Data 6. The ATAC-qPCR results were first normalized by subtracting the Ct value of genomic DNA, yielding a normalized Ct value. The deltaCt value was then calculated by subtracting the normalized Ct value of a genomic locus near olfactory receptor gene (OR1A1), which exhibits very low ATAC-seq signal and low transcription levels in TSCs. The data was presented as negative deltaCt values, where an increase in value corresponds to increased openness.

## ChIP-seq, bioChIP-seq, and ATAC-seq data processing and analysis

In addition to the sequencing data we generated in this study, the published TFAP2C ChIP-seq in TSC data was used (GSE208539). The raw data were processed according to the following pipeline. The sequencing reads were aligned to the human reference genome hg38 using Bowtie2 (v2.4.5)[94] with the default setting. The aligned reads were filtered based on mapping quality (quality value > 10) using samtools[95]. The data was visualized on Integrative genomics viewer (IGV)[96] with normalized wig files generated by deepTools (bamCoverage -bs 10 --normalizeUsing CPM --extendReads 67)[97]. Individual peaks were called by MACS3[98] with default settings. Further score filtering was applied to remove weak peaks, and peaks found in simple redundant genome regions were also filtered out. Enriched motifs of binding loci of each TFs were identified using findMotifsGenome.pl module with the default setting under HOMER (v4.11)[99]. For motif analysis of histone ChIP-seq data (H3K27ac ChIP-seq peaks in EVT D3), a region size of 500 bp was used as recommended. Quantitative comparison of ChIP-seq and ATAC-seq peaks was performed with MAnorm (v1.3.0)[73] with the default setting. The results were visualized using Java TreeView[87], and GO enrichment analysis was performed using the GREAT software[100].

## Identification of super-enhancers

H3K27ac peaks were used to identify super-enhancers using the ROSE program[35,36]. Briefly, H3K27ac peaks called by MACS3 were transferred to gff files and used as input for the ROSE program. The program was run with the default setting along with the -s 12500 and -t 2500 options, which exclude peaks within 2,500 pb from the TSS to avoid capturing promoter peaks.

## Mapping peaks to gene

The associated genes of ChIP-seq or bioChIP-seq peaks were mapped by navigating surrounding regions (± 20 Kb). All RefSeq genes in the refFlat file (hg38) downloaded from the UCSC genome browser were used for gene mapping. The genomic distribution of the ChIP-seq or bioChIP-seq peaks was monitored based on the following criteria: intron, exon, promoter region (within ± 2 Kb from the TSS), and intergenic region (binding sites except for intron, exon, and promoter).

## Virus preparation and infections

Bacterial stocks containing lentiviral shRNA for a target gene were purchased from MilliporeSigma (MISSION shRNA library, Supplementary Data 7). Lentiviral vector carrying shRNA for KD or non-target sequence (MilliporeSigma, SHC202) was transfected to HEK293T cells at 70% confluency using GenJet™ In Vitro DNA Transfection Reagent (SignaGen Laboratories, SL100488), according to the manufacturer's instructions. After 18 h, HEK293T medium was replaced with TSC medium. Viral supernatants were harvested 24 h and 48 h following medium replacement. To KD a target gene, TSC were infected in viral supernatant containing polybrene (Santa Cruz Biotechnology, sc-134220) at a final concentration of 10 µg/mL. After 18 h, the infected cells were induced to differentiate into EVT using EVT differentiation media supplemented with puromycin at a final concentration of 1 µg/mL. For the time-course KD of TFAP2C, the viral media was applied to TSC for early knockdown (Early-KD), and to differentiating cells at day 3 (Mid-KD) and day 5 (Late-KD). The differentiation media supplemented with puromycin was used according to the specific differentiation time, replacing the media after 18 h of infection.

## Stable cell line generation for inducible overexpression of tagged proteins

To establish stable cell lines with inducible OE of a target gene, we initially modified the pSBtet-GP vector (Addgene, plasmid #60495), a sleeping beauty vector with inducible promoter. The luciferase sequence in the vector was replaced with NotI and NheI restriction enzyme sites, and a flag tag and a biotinylation sequence were inserted at the N-terminal of the enzyme sites. This modified vector was referred to as the SBFB vector. To establish stable cell lines with inducible overexpression of a target gene, we initially modified the pSBtet-GP vector (Addgene, plasmid #60495), a sleeping beauty vector with an inducible promoter. The luciferase sequence in the vector was replaced with NotI and NheI restriction enzyme sites, and a flag tag along with a biotinylation sequence was inserted at the N-terminal of the enzyme sites. This modified vector was referred to as the SBFB vector. Next, the coding sequence of individual target genes was inserted into the SBFB vector. This construct, along with a vector containing the sequence for the BirA enzyme and a transposase-containing pCMV(CAT)T7-SB100 vector (Addgene, plasmid #34879), was transfected into TSC. After 24 h, the cells were subjected to selection using puromycin at a final concentration of 1 µg/mL and G418 at a final concentration of 250 µg/mL. To induce protein expression, doxycycline (Fisher Scientific) was administered at a final concentration of 1 µg/mL.

## Quantitative gene expression analysis

cDNA synthesis was performed from 500 ng of total RNA using qScript™ cDNA SuperMix (QuantaBio, 101414-108). The synthesized cDNA was diluted (x20), and 2 μL of the diluted cDNA was used for each reaction. qPCR was performed using PerfeCTa SYBR Green FastMix Reaction Mixes (QuantaBio, 95072-012) and a StepOnePlus™ Real-Time PCR System (Thermo Fischer Scientific, brand Applied Biosystems). Primers were designed to amplify the junction between two exons using a web-based primer design program, Primer3 (https://primer3.ut.ee/). All primer sequences used for qPCR are shown in Supplementary Data 6. Relative transcript abundance was normalized to GAPDH as a loading control. Data were calculated as relative to control data using the $2^{-\Delta\Delta CT}$ method.

## Immunofluorescence

Cells were fixed with 4% paraformaldehyde for 20 min at room temperature and washed three times with PBS (containing $Ca^{2+}$ and $Mg^{2+}$, PBS-CM). The fixed cells were permeabilized and blocked in 10% normal goat serum in PBS-CM with 0.3% Triton X-100 (Sigma-Aldrich, T8787) for 45 min at room temperature. For staining of HLA-G (surface protein), a blocking solution without Triton X-100 was used. For staining of HLA-G with nuclear proteins, HLA-G was stained without permeabilization then the membranes were permeabilized with a blocking solution with 0.1% Triton X-100. The cells were incubated with primary antibody at 4 °C overnight, followed by washing three times in PBS-CM. Secondary antibodies with Alexa Fluor 488- or 594-conjugated (Thermo Fisher Scientific) were incubated 1 h 30 min at room temperature. The nuclei were counterstained with 4′,6-diamidino-2-phenylindole (DAPI, Sigma-Aldrich). Images were obtained with a confocal microscope (Zeiss, LSM 710) with a combination of 2–4 laser lines. The best signal option which is a sequential imaging of all fluorophores in different tracks, was used to minimize crosstalk. A 20x objective was employed for image acquisition. To maintain consistency, samples within the same experiment underwent identical immunofluorescence microscopy parameters. Subsequently, the images were processed using the ZEISS ZEN lite program and labeled using illustrator program for presentation purposes. The following primary antibodies were used for IF: DLX5 (Novus Biologicals, NBP1-85793, 1:400), DLX6 (Proteintech, 23216-1-AP, 1:200), ASCL2 (R and D Systems, AF653, 1:20), ZNF439 (Novus Biologicals, NBP2-13574, 1:100), NRIP1 (abcam, ab42126, 1:100), TFAP2C (Cell Signaling Technology, 2320, 1:400), HLA-G (Santa Cruz Biotechnology, sc-21799, 1:100), MMP2 (Cell Signaling Technology, 40994, 1:200), and TP63 (BioLogo, PP040-0.5, 1:200).

## Western blotting

Cells were lysed using Laemmli Sample Buffer (Bio-Rad, 1610747) supplemented with 5% β-mercaptoethanol (MilliporeSigma, M3148). Lysates were heated to 95 °C for 10 min then loaded on 7.5%, 10%, or 12% SDS gel, depending on the size of target proteins, followed by transfer to PVDF membranes (MilliporeSigma, IPVH00010). Membranes were blocked in 5% skim milk (Bio-Rad, 1706404) or 5% BSA (Sigma-Aldrich, A7906) in TBS-T (Tris-buffered saline with 0.1% Tween 20), and incubated with primary antibody overnight at 4 °C, followed by washing three times in TBS-T. The membranes were incubated in secondary antibodies for 1 h at room temperature, followed by washing three times in TBS-T. Membranes were incubated with ECL Western blotting detection reagent (Fisher Scientific, 45-002-401) and imaged using ChemiDoc XRS+ (Bio-Rad). The following primary antibodies were used for Western blot: DLX5 (Novus Biologicals, NBP1-85793, 0.2 mg/mL), DLX6 (abcam, ab137079, 1:2,500), ASCL2 (R and D Systems, AF653, 1 ug/mL), ZNF439 (GeneTex, GTX119735, 1:1,000), NRIP1 (Millipore, MABS1917, 1:1,000), TFAP2C (Santa Cruz Biotechnology, sc-12762, 1:500), HLA-G (Santa Cruz Biotechnology, sc-21799, 1:500), MMP2 (Cell Signaling Technology, 40994, 1:1,000), and ACTB (Abgent, AM1829B, 1:20,000).

## Invasion assay

Invasion assay was performed as previously described[47] with modification. Briefly, Molecular Probes Secure Seal Hybridization Chambers (Thermo Scientific, S24732) were treated with ultraviolet (UV) light for 1 h in a laminar flow hood. The chambers then adhered to the bottom of 12-well plate. Then the mixture of Matrigel and EVT differentiation medium (2:1, v/v) were added to the chambers. After equilibration in $CO_2$ incubator (5% $CO_2$, 37 °C) for 1 h, $5 \times 10^4$ cells in 30 μL EVT medium were seeded, and the cells were allowed to adhere to the Matrigel surface for 1 h in a $CO_2$ incubator while tilting the plates perpendicular to the surface of the incubator. Then the chambers were covered with 1 mL of EVT medium. The culture medium was changed on day 3 and 6 as EVT differentiation protocol described above. Images were captured using an EVOS Fl microscope (Thermo Fisher Scientific). The invasion area was calculated using Image J software[101]. Measurements were obtained from images taken in three independent biological replicates.

## Flow cytometry

Cells were dissociated into single cells using TrypLE and then washed with FACS buffer (PBS without Mg2+, Ca2+, 5% FBS, 2 mM EDTA, and 0.1% sodium azide). For measuring surface HLA-G expression, the cells were incubated with phycoerythrin-conjugated HLA-G (abcam, ab24384) at a 1:50 dilution in FACS buffer for 30 min on ice. For the apoptosis assay, the single cells were resuspended in 200 μl of 1× Annexin V binding buffer with 5 μl of Sulforhodamine 101–Annexin V (Texas Red) stock solution (Biotium, 30067) and incubated for 30 min at room temperature, while being protected from light. After incubation, the cells were washed three times with FACS buffer and filtered through a 70 μm cell strainer (Falcon, 352235). Flow cytometry analysis was performed using a BD Accuri flow cytometer (BD Biosciences), and the data were analyzed using FlowJo software (v9, Treestar).

## Statistics and reproducibility

Data in graphs are expressed as mean, and error bars represent the standard deviation of the mean (SD). The number of biological replicates per experiment is indicated in the figure legends. All experiments reporting values, except for the time-course study, were performed at least two biological repeats. No statistical method was used to predetermined sample size. No data were excluded from the analysis. Statistical significance between groups were determined by two-tailed Student's t-test. A $p < 0.05$ was considered statistically significant. For all the presented boxplots, the center represents the median value, and the lower and upper lines represent the 5% and 95% quartiles, respectively. Significant difference between different groups in box plots and violin plots was determined using the two-tailed Wilcoxon rank-sum test, and a $p < 0.05$ was considered statistically significant.

## Reporting summary

Further information on research design is available in the Nature Portfolio Reporting Summary linked to this article.

# Data availability

The sequencing data generated in this study have been deposited in the GEO database under accession code GSE212267. The published TFAP2C ChIP-seq in TSC data was utilized from GSE208539. Source data are provided with this paper.

# Code availability

This paper does not report original code. Any additional information required to reanalyze the data reported in this paper is available from the corresponding author upon reasonable request.

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

## Acknowledgements
We thank Drs. Takahiro Arima and Hiroaki Okae (Tohoku University) for sharing human TSC lines and technical advice as well as Dr. Lucy LeBlanc for critical reading of the manuscript. We also thank the UT Austin Genome Sequencing and Analysis Facility (GSAF) and Texas Advanced Computing Center (TACC) for sequencing data analysis as well as the Center for Biomedical Research Support (CBRS) for confocal microscopy. This work was supported by R01HD101512 (NIH/NICHD) and Preterm Birth Research Grant (1017294, Burroughs Wellcome Fund) to J.K. and the Graduate School Continuing Fellowship to M.K.

## Author contributions
Conceptualization: J.K.; Investigation: M.K., B.K.L., Y.J.J., Q.G., A.J.S., N.A.K. and A.B.R.; Formal Analysis: M.K., B.K.L. and M.L.; Resources: J.K.; Data Curation: M.K.; Writing - Original Draft: J.K., M.K. and B.K.L.; Visualization: M.K., B.K.L. and J.K.; Supervision: J.K.; Funding Acquisition: J.K.

## Competing interests
The authors declare no competing interests.
