## [Peer Review File · Nature Communications]

The transcriptional regulatory network modulating human trophoblast stem cells to extravillous trophoblast differentiationREVIEWER COMMENTS

Reviewer #1 (Remarks to the Author):

Summary:

In this manuscript, Kim et al. use the human trophoblast stem cell (TSC) model to explore the function of transcription factors (TFs) in modulating the extravillous trophoblast EVT differentiation. Based on RNA-seq expression dynamics and enhancer profiling, the authors selected putative early (TFAP2C) and late stage (DLX5/6) regulators of EVT differentiation and performed KD, OE, ATAC-seq, and ChIP-seq analysis to further characterize their function. They propose that TFAP2C functions as a pioneer factor, controlling the expression of late stage TFs, which in turn repress TFAP2C and activate EMT in both EVTs and cancer cells.

Overall, this is a relevant and comprehensive study requiring revisions in terms of data integration and biological relevance.

Major comments:

- 1) The second sentence in the abstract claims that regulatory factors of EVT differentiation are unknown, while the statements at the end of the first and second Introduction paragraphs are more tempered and appropriate as the role of some factors like ASCL2 in EVT differentiation has been recently described (PMID:33649217). This should be mentioned in the Introduction and not just in the Results section.
- 2) The efficiency of EVT differentiation has not been sufficiently characterized in Fig.1, as FACS quantification, immunostainings and comparison to real EVT transcriptome is missing. Since the paper is based on the bulk RNA-seq analysis coming up with potential EVT differentiation regulators, the efficiency and quality of the differentiation are essential to demonstrate. RNA expression dynamics of two markers is not sufficient to characterize any cell type.
- 3) Throughout the manuscript, the authors focus on the 4 gene clusters identified in Fig.1 at the expense of EVT-related genes. As a result, the manuscript lacks insights into EVT biology and does not sufficiently discuss the results in this context. This point must be addressed in the revised version.
- 4) In Fig.2, where two kinds of EVT regulators are being postulated, a basic immunostaining of these in the 1st. trimester placental tissue is necessary. In particular, in the context of DLX5 and DLX6 expression in CT (Fig.2e).
- 5) The EVT migration assays used in Fig.3b is an important measure of functionality and should therefore include a quantitative readout.
- 6) In general, the KDs in Fig.3 should be characterized first in an unbiased way (i.e., not only in the context of the 4 clusters) and more comprehensively. How are other lineage markers affected by these KDs? Are the cells stuck in a progenitor state or do they deviate into other lineages (as reported for the ASCL2-KD, PMID:33649217)?
- 7) As in Fig.3, the characterization of the TFAP2C-KD time course in Fig.4 does not clearly show what happens to other key lineage genes except HLA-G, MMP2 and TP63 during the differentiation. A heatmap containing a set of key trophoblast TFs and comparing their expression changes during the time course would be useful. What genes are impacted most by the KDs/OE and how does this relate back to trophoblast biology.
- 8) The integration of the ChIP-seq data from Fig.5 with the KD expression data from Fig.3/4 is surprisingly missing. This is essential to show which genes are directly bound and regulated/co-regulated by a given TF. What is the overlap of bound vs. misregulated genes? When are these bound genes expressed? To what extent TFAP2C and DLX6 share target genes overall and not just in specific clusters?
- 9) Along the same lines, while comprehensive ChIP-seq/RNA-seq analysis have been

performed, they've not been comprehensively analyzed in an integrated manner. For instance, the authors make important conclusions (e.g., lines 263-276) and reference only genome browser snapshots without the global perspective (Fig.5d, Fig.6a and Extended Fig.6a,b).

10) Line235-237: Conclusion that "TFAP2C bound to its targets in self-renewing TSCs and cells in early-stage day 2, followed by decreased occupancy as differentiation progressed. On the other hand, we detected DLX6 target occupancy starting at day 5 of differentiation (Fig.5b)" is based on the number of peaks. Instead, it should be based on a quantitative analysis of TFAP2C and DLX6 binding (i.e., differential binding analysis).

11) The display G1-G3 loci in Fig.3e,f is misleading. The authors should present data obtained for each time-point.

12) A part of the paper focuses on the putative TFAP2C pioneer activity without discussing/excluding other possibilities. What criteria does a pioneer factor need to fulfill? Is it possible that TFAP2C acts as a repressor of late EVT genes, instead as a pioneer-activator? Pioneer factors do not just prime genes for later expression but bring in immediate change to the local chromatin environment by recruiting other chromatin modifying/remodeling activities, which has not been demonstrated.

13) A lot (in my view too much) prominence is given to the general GO-term analysis without mentioning specific gene examples. Instead, the authors should focus on what genes are affected and how this impacts EVT biology. Are there new interesting target genes that are bound by TFAP2C/DLX6? What critical stage in EVT development/function may they regulate?

14) The ability of DLX5/6 to activate EMT genes should also be demonstrated in EVTs and not just cancer cells. The statement that SNAI1 might be an EVT-specific EMT TF is misleading as SNAI1 is a well-known general EMT regulator in different contexts. The general connection between cancer EMT and EVT differentiation could be explored a bit more by data mining. For instance, are there other relevant expression signatures shared by cancers and the EVT lineage? What about TFAP2C in this context? Is it also upstream of DLX genes in cancers? The general idea of faulty re-activation of extra-embryonic genes could be discussed as a wider context.

15) The Discussion needs a substantial revision in terms of structure. More detailed discussion of pioneer vs. possible repressor functions might be beneficial. It is requested that the authors discuss the results in the context of EVT biology beyond just general aspects of EMT.

Minor comments:

1) Fig.2c is unreadable.

2) The original scans for the Western blots are missing.

3) Fig.4j and Fig.1f: x-axis labeling would be beneficial.

4) The EVT KD labeling in Fig.7f is unreadable.

5) All genome browser snap-shots lack scale.

6) The authors should specify settings of their bioinformatic analyses, also when standard settings were used.

Significance

Taken together, this study represents a significant advance in exploring the role of TFs during EVT differentiation using a novel human TSC model. After revisions, the paper would be relevant not just to the trophoblast development and reproductive medicine community, but also in more general terms related to TF biology.

Reviewer #2 (Remarks to the Author):

The Manuscript by Kim et al. used human trophoblast stem cells (hTSC) as a model system to understand the transcriptional control of human extravillous trophoblast (EVT) stem cells. Authors concluded a multi-stage transcriptional mechanism of EVT differentiation process, in which transcription factor TFAP2C instigates the EVT differentiation process and later other transcription factors, such as DLX5/6, mediates the late stages of the EVT differentiating process. The strength of the manuscript is complementary genomics approaches that provide an understanding of longitudinal transcriptional mechanisms during EVT differentiation. However, the manuscript has several flaws in designing experiments, includes contradictory data compared to published reports and is extremely premature.

My concerns are mentioned below:

1. The overwhelming issue with the manuscript is the significance of the findings. In Figure 1, authors presented transcriptomics data in early vs. late stages of EVT differentiation. However, there is no effort to correlate early stage EVTs and late stage EVTs with respect to the EVT populations at human placentation sites. It is true that the present concept accepts multiple EVT populations in vivo, such as proximal and distal placental EVTs as well as Interstitial and endovascular EVTs in the maternal uterine compartment. However, the manuscript lacks any data (Such as gene expression verification of class 1-4 genes) with human tissues to correlate early vs. late EVTs with respect to the actual EVT populations in a placentation site.

The major issue is variable gene expression patterns in human TSCs reported here with existing data. The available gene expression data by Okae et al. in initial Cell Stem Cell manuscript as well as multiple available single-cell RNA seq data clearly shows that TFAP2C is abundantly expressed in primary cytotrophoblasts (CTBs) and EVTs. More importantly, unlike the findings here, TFAP2C expression is maintained in fully differentiated EVTs. Similarly, DLX5/6 are as highly expressed in CTBs as they are in EVTs and ZNF439 expression is suppressed in primary EVTs. Thus, given contradictory observations regarding expressions of key factors in this study vs. primary trophoblasts, without any in vivo validation, the whole manuscript seems an intellectual exercise.

In relation to this, it is important to include FPKM values and fold changes of the genes in supplementary table (which only contain gene names). Otherwise, the information seems incomplete.

2. The manuscript includes only Histone H3K27 modification to define enhancer signature. This experiments need verification with other marks such as H3K4Me1 and H3K4Me3. In addition, at least one non-histone signature needs to be included to identify super-enhancers. Otherwise, the enhancer dynamics prediction is incomplete. The experimental procedure needs to include more detail about how enhancer mapping was done. The supplementary table needs to indicate genomic locations of enhancers and super enhancers with respect to nearby genes. It is not clear what is the cut-off distance of super-enhancers (The 12.5kb spanning and a minimal distance of 2.5 kb from TSS is not complete information). It seems several of the genes may have multiple nearby SEs. The authors should provide more information for better understanding of the manuscript.

3. The loss-of-function analyses of TFAP2C and Other genes are also of major concern. The

efficiency of the time-course approach of TFAP2C-depletion is not clear (Did not notice any time-course of expression data and the methods section lacks the detail description of the RNAi approach). The approach is flawed as it creates window of extreme variability in day 0 vs. day 3 vs. day 5 cells. A better experimental approach, in which TFAP2C is conditionally depleted (using inducible shRNA) over different time-points need to be used.

4. As TFAP2C is important to maintain the self-renewal in human TSCs, the lack of EVT differentiation in day 0 depleted cells may have other conclusions, such as extreme cell stress, induction of apoptosis, promotion of syncytiotrophoblast differentiation etc. These aspects are overlooked.

5. If DLX5/6 and NRIP1 (which is induced upon EVT differentiation) are late acting factors, why they are depleted at day 0? Also, the data needs western-blot expression analyses of target genes at different days (Fig. 3A includes western blot but is not in a time-dependent manner).

6. The ATAC-seq data showing TFAP2C occupied regions have low accessibility needs better validation. Again the expression variability is a concern. TFAP2C is abundantly expressed in undifferentiated TSCs and is suppressed in late EVTs. Also, as mentioned above, DLX5/6 are highly expressed in undifferentiated cells. Thus, the data seems contradictory.

7. The EMT analyses and relationship to cancer (Presented in Fig. 7) is very preliminary, incomplete and out of the focus of the study. Authors should remove that section.

Minor Comments:

1. As mentioned above, experimental procedure section needs more details.
2. Supplementary tables should have more information (Mentioned above)
3. Line 114 mentions "genes belonging to class ---which are highly active in fully differentiated EVTs". What is meant by "fully active"?

Reviewer #3 (Remarks to the Author):

This manuscript by Kim et al. has attempted to track the trajectory of human trophoblast stem cells as they differentiate into extra-villous trophoblast (EVT). They focus on changes in the transcriptome and what the authors refer to as the "enhancer landscape" This is a valuable exercise because, surprisingly, the steps to EVT formation have not been pursued in any detail despite the availability of the TSC lines of Okae et al. for more than four years and the importance of EVT in relation to placental disease. Of particular interest is the role of TFAP2C in directing both the early and later steps of differentiation. Although much of the work is descriptive (though still of value) intervention in the process by shRNA silencing confirms many of the authors' inferences about the roles of particular groups of transcription factors. Moreover the description of a defined medium for TSC differentiation will be helpful to others.

The manuscript is clearly written and easy to follow. The conclusions are reasonable and not overblown. It provides data that will be useful to the ever growing number of groups interested in TB emergence and differentiation. I do, however, have a few comments that the authors should address.

1. TSC lines, and particularly those originating from induced pluripotent stem, do not all behave uniformly. I was unsure whether only a single or multiple lines had been tested. If just a single line, how do we know whether the conclusions have general validity? Also the origin of the line(s), i.e., from blastocyst versus villous, and their sex should be made clear.
2. EVT is likely not a single functional lineage, despite having overall similarities. For example is it agreed that the EVT that invades blood vessels is of the same genetic identity as those that form an EVT shell around the implanting conceptus or even give rise to the so-called trophoblast giant cells? This topic deserves some discussion in relation to differentiation trajectories.

Overall Response

We appreciate the reviewers' invaluable comments and suggestions, which were helpful for us in improving the quality of our manuscript. The main concerns raised by the reviewers are:

1. The biological relevance of the EVT TFs.
2. The general validity of the results obtained from a single TSC line.
3. The need for data integration and considering other lineages.

To address these concerns, we performed additional experiments and analyses.

First, we investigated the biological relevance of the EVT TFs identified in TSCs *in vitro* by utilizing publicly available bulk RNA-seq data¹, scRNA-seq data^{2,3}, and single-nucleus RNA sequencing (snRNA-seq) data⁴ obtained from first-trimester human placenta. Our analysis revealed that all late-stage TFs we investigated in our manuscript are also highly expressed in primary EVT, although some factors (DLX5/6 and ZNF439) exhibit comparable expression levels in primary CT. Importantly, we compared the expression levels of TFAP2C between EVT and CT and observed lower expression of TFAP2C in EVT, supporting our findings.

Second, to confirm the general validity of our results, we conducted new experiments using additional TSC lines with both sexes. Our results indicated that the expression pattern of EVT TFs, initially observed in the CT27 line, was indeed conserved across all the TSC lines examined. To further strengthen our findings, we performed additional KD experiments using the same experimental setup described in our original manuscript to assess the essential roles of the defined EVT TFs in EVT differentiation across multiple TSC lines. The results demonstrated the consistent requirement of the EVT TFs for proper EVT differentiation, regardless of the specific TSC line utilized. This consistency supports the robustness of our findings and highlights the significance of these TFs in EVT differentiation.

Third, with additional experiments, we performed an in-depth analysis to integrate our ChIP-seq and RNA-seq data, facilitating a comprehensive understanding of the regulatory landscape during EVT differentiation. Furthermore, we investigated the impact of KD of EVT TFs on the expression of marker genes associated with other trophoblast lineages. We observed that the cells with KD of the late-stage TF maintained the expression of CT marker genes, and the KD of TFAP2C, DLX6, ASCL2, and NRIP1 resulted in increased expression levels of ST genes. These integrative analyses and monitoring of CT and ST marker genes contribute to a more comprehensive view of the roles of EVT TFs and their relationship with the broader trophoblast biology.

Lastly, we added new data and thoroughly revised our manuscript. We also provide point-by-point answers to the questions raised by the reviewers. We hope our responses sufficiently address the reviewers' concerns.

Summary of responses to Reviewers' comments:

*Data reanalyzed; **Additional experiment was conducted

M1-6 and M1-3: minor comments

Reviewers' comment	Response
1	The text was edited (page 2).
2	* ** IF and flow cytometry were newly conducted and RNA-seq data were reanalyzed (Extended Data Fig. 1a-c).
3	Addressed Reviewer's comment.
4	* ** Expression levels of EVT TFs in the human placenta were monitored by using publicly available bulk RNA-seq, scRNA-seq, and snRNA-seq data. Expression patterns of EVT TFs were confirmed in the five established TSC lines (New Fig. 2e and Extended Data Fig. 2b-d).
5	Invasion assay results were quantified, and Fig. 3b was updated accordingly.
6a	We addressed Reviewer's comment.
6b	* Results of clustering analysis and GSEA with ST gene set were newly added (Extended Data Fig. 3h).
7	** Heatmaps showing changes in EVT, TSC, and ST marker genes in TFAP2C KD/OE cells were newly added as Extended Data Fig. 4b,j.
8	* A comprehensive analysis of ChIP-seq and RNA-seq data was conducted, and Extended Data Fig. 5g,i were newly added (page 8).
9	We addressed Reviewer's comment.
10	* Heatmaps showing TFAP2C and DLX6 peaks at each time point were newly added (Extended Data Fig. 5c,d).
11	We addressed Reviewer's comment, and the main text was edited (page 8).
12	* ** Changes in enhancer signals on TFAP2C ChIP-seq unique loci, as compared to ATAC-seq data, were newly added as Fig. 6f, and the Discussion section was revised (page 12).
13	Specific gene examples were added to the Results section (pages 5-6 and 8).
14	EMT part was excluded as suggested by Reviewer 2 (#2-7).
15	The Discussion section was revised.
M1	Fig. 2c was edited (Extended Response Data Fig. 1).
M2	Original scans of Western blot images were added (Extended Response Data Fig. 2).
M3	X-axis labels were edited in Fig. 4j and Fig. 1f (Extended Response Data Fig. 3).
M4	Fig. 7f was excluded (please refer to the response to the Reviewer's comment #1-14).
M5	Scale and locus information were added to all genome browser snap-shots (Extended Response Data Fig. 4).
M6	Specific settings of bioinformatic analysis were added to the Methods section (Extended Response Data Text 1).

2	1a	*	The expression levels of EVT subtype markers, defined by spatial sequencing and snRNA-seq data of human placenta, in EVT differentiating cells in vitro were presented. This topic was discussed in the Discussion section (page 11).
	1b	*	Please refer to our response to Reviewer 1's comment #1-2 for the expression levels of EVT TFs in the human placenta. Count information of RNA-seq data was newly included as Supplementary Table 1. Log2 fold changes of class 1-4 genes were added to Supplementary Table 2.
	2a	* **	Additional enhancer mapping with other enhancer marks, including P300, Med1, and H3K4me1 was conducted and Extended Data Fig. 1f was newly added. Updated Supplementary Table 3 contains SE locus and associated gene information.
	2b	*	Detailed information about SE mapping was provided in the revised Methods section (page 27).
	3	**	Data showing KD timing after lentivirus infection were added.
	4	**	Data showing the effects of TFAP2C KD from multiple time points during the early stage of differentiation and apoptosis levels of TSCs infected with lentivirus were newly added as Extended Data Fig. 4c,d.
	5a	**	Results showing the requirement of the late-stage factor during the late stage of differentiation were newly added as Extended Data Fig. 4e.
	5b	**	Fig. 3a was replaced with Western blot results of cells collected at different time points upon KD of individual EVT TFs.
	6	**	ATAC-qPCR results confirming the binding of TFAP2C on cis-regulatory elements of EVT-active genes with low ATAC-seq signal in TSCs were newly added (Fig. 6d and Extended Data Fig. 6d).
	7		The EMT part was excluded.
	M1		Details in experimental processes were added to the Methods section (pages 26-30, Extended Response Data Text 1).
	M2		Supplementary Tables were updated (Extended Response Data Table 1).
	M3		Text was edited (page 7).
3	1	**	The importance of EVT TFs was confirmed in other TSC lines (New Extended Data Fig. 3a,c)
	2		In vivo EVT lineages and EVTs differentiated in vitro were discussed in the Discussion section (similar to the comment #2-1a; page 11).

REVIEWER COMMENTS AND POINT-BY-POINT RESPONSES

Reviewer #1

Summary:

In this manuscript, Kim et al. use the human trophoblast stem cell (TSC) model to explore the function of transcription factors (TFs) in modulating the extravillous trophoblast EVT differentiation. Based on RNA-seq expression dynamics and enhancer profiling, the authors selected putative early (TFAP2C) and late stage (DLX5/6) regulators of EVT differentiation and performed KD, OE, ATAC-seq, and ChIP-seq analysis to further characterize their function. They propose that TFAP2C functions as a pioneer factor, controlling the expression of late stage TFs, which in turn repress TFAP2C and activate EMT in both EVTs and cancer cells. Overall, this is a relevant and comprehensive study requiring revisions in terms of data integration and biological relevance.

Significance

Taken together, this study represents a significant advance in exploring the role of TFs during EVT differentiation using a novel human TSC model. After revisions, the paper would be relevant not just to the trophoblast development and reproductive medicine community, but also in more general terms related to TF biology.

We sincerely appreciate the reviewer's insightful and constructive comments on our manuscript. We have carefully considered each suggestion and implemented significant revisions to address them. We want to assure the reviewer that we have taken the reviewer's comments seriously and dedicated considerable effort to incorporating them into our revised manuscript.

Major comments:

1-1. The second sentence in the abstract claims that regulatory factors of EVT differentiation are unknown, while the statements at the end of the first and second Introduction paragraphs are more tempered and appropriate as the role of some factors like ASCL2 in EVT differentiation has been recently described (PMID:33649217). This should be mentioned in the Introduction and not just in the Results section.

We appreciate the Reviewer for bringing this to our attention. In the revised Introduction section, we have incorporated a mention of ASCL2, a known regulator of EVT. Now it reads, "Another study has discovered the significance of ASCL2 as a pivotal factor in EVT differentiation using TSCs, and its role has been validated *in vivo* placenta development using a rat model⁵, suggesting the feasibility of the use of TSC to EVT differentiation model to study *in vivo* EVT differentiation." (page 2)

1-2. The efficiency of EVT differentiation has not been sufficiently characterized in Fig.1, as FACS quantification, immunostainings and comparison to real EVT transcriptome is missing. Since the paper is based on the bulk RNA-seq analysis coming up with potential EVT differentiation regulators, the efficiency and quality of the differentiation are essential to demonstrate. RNA expression dynamics of two markers is not sufficient to characterize any cell type.

We appreciate the critical comment provided by the reviewer. In our original study, we utilized a well-established protocol for EVT differentiation¹ and employed RT-qPCR and RNA-seq to assess the induction of EVT marker genes and the reduction of CT genes along with the morphological change (**Fig. 1a,b,d** and **Extended Data Fig. 1h**), demonstrating successful EVT differentiation.

To confirm the robustness of EVT differentiation in the revised manuscript, we conducted additional immunofluorescence analysis on samples at different time points during EVT differentiation. The results validated the increased protein expression of HLA-G, an EVT marker gene, and demonstrated a reduction of TP63, a CT marker gene (**Response Fig. 1a**). Moreover, we further observed the increase in surface HLA-G expression during EVT differentiation via additional flow cytometry (>94% of HLA-G positive cell; **Response Fig. 1b**). All these strongly support the successful differentiation of TSC to EVT.

Since the differentiation of TSC to EVT alters core TSC and EVT genes, we reanalyzed our RNA-seq data on TSCs and EVTs (EVT D8) to assess the levels of TSC and EVT marker genes by GSEA. The results revealed an increase in the expression of fetal EVT genes, while a decrease in the expression of fetal CT/ST genes was observed after 8 days of EVT differentiation^{6,7} (**Response Fig. 1c**). Similarly, when comparing our EVTs to TSCs using gene sets defined in a scRNA-seq study of the human first-trimester placenta², we observed the significant upregulation of EVT genes, along with the downregulation of CT genes (**Response Fig. 1c**). These findings provide additional evidence supporting the successful EVT differentiation in our study. We have included these results in the revised manuscript as **Extended Data Fig. 1a-c** and have made corresponding edits to the main text (pages 2-3).

Response Fig. 1: Conformation of efficient EVT differentiation.

a, Immunofluorescence analysis of TP63 and HLA-G in TSCs and differentiating cells to EVT at day 1 (EVT D1), 2 (EVT D2), 4 (EVT D4), 6 (EVT D6), and 8 (EVT D8). Scale bar: 100 μ m. **b**, Flow cytometry analysis of surface HLA-G expression of TSCs and EVT differentiating cells. HLA-G expression was measured using a PE-conjugated HLA-G antibody and compared with the PE-conjugated isotype control. **c**, GSEA using Descartes fetal placenta extravillous trophoblasts (Fetal EVT) and Descartes fetal placenta syncytiotrophoblasts and villous cytotrophoblasts (Fetal CT/ST) gene sets^{6, 7}, along with EVT and CT marker genes². The transcriptomes of EVT D8 and TSCs were compared to confirm the efficiency of EVT differentiation. NOM and FDR indicate nominal p-value and false discovery rate, respectively.

1-3. Throughout the manuscript, the authors focus on the 4 gene clusters identified in Fig.1 at the expense of EVT-related genes. As a result, the manuscript lacks insights into EVT biology and does not sufficiently discuss the results in this context. This point must be addressed in the revised version.

We apologize for any lack of clarity. First, we would like to clarify that our analysis did not solely depend on the four gene classes we defined. Our original manuscript includes unbiased analyses, such as PCA (**Fig. 1c** and **3f**), distance heatmap (**Fig. 1f**), GSEA (**Fig. 3d, 4i** and **Extended Data Fig. 3g,h**), and GO analysis (**Fig. 3e,g**), which utilize the entire transcriptome rather than focusing specifically on the four gene clusters.

The classification of DEGs from the time-course transcriptome data was crucial for identifying interesting genes that might have been overlooked by solely examining the transcriptome of the endpoint TSCs and EVTs. In fact, this classification enabled us to identify class 3 genes whose expression was upregulated during the early stage of EVT differentiation and decreased in the late stage. This finding significantly contributed to our understanding of the EVT differentiation mechanism.

Furthermore, the defined four gene classes accurately reflected known TSC and EVT biology. As mentioned in our original manuscript, well-established TSC marker genes and regulators, such as TP63, ELF5, and TEAD4 were classified as class 1 (TSC-active) genes, while EVT markers, such as HLA-G and MMP2, along with the known regulator ASCL2⁵, were categorized as class 2 (EVT-active) genes. The GO term analysis results presented in our original manuscript further supported the notion that the defined gene classes represent the functions and characteristics of TSCs and EVTs (**Fig. 1e**).

Moreover, in our original manuscript, we incorporated gene sets defined by other groups/sources, including CT and EVT gene sets from a scRNA-seq study of human placenta² (**Fig. 3d, 4i** and **Extended Data Fig. 3g**) and a fetal placenta EVT gene set from the Molecular Signatures Database^{6, 7} (**Fig. 3d**). Notably, all the data derived from these gene sets yielded similar results to those obtained from the analysis using our defined gene classes, further validating the usability of our classification as an alternative approach.

In summary, our analysis incorporated various unbiased Methods that considered the entire transcriptome to investigate the dynamics of gene expression profiles during differentiation and the roles of EVT TFs. The gene classes we defined accurately represented known TSC and EVT biology, and the inclusion of gene sets from external sources further validated our findings, collectively supporting the feasibility of our analysis.

1-4. In Fig.2, where two kinds of EVT regulators are being postulated, a basic immunostaining of these in the 1st. trimester placental tissue is necessary. In particular, in the context of DLX5 and DLX6 expression in CT (Fig.2e).

Due to the limitation of accessing first-trimester human placenta specimens, an ongoing challenge in the field of human placenta research, we attempted to address the comment by using publicly available bulk RNA-seq data¹, scRNA-seq data^{2, 3}, and recently released snRNA-seq data⁴ of first-trimester human placentas.

In our original manuscript, we demonstrated through RT-qPCR and RNA-seq analyses that TFAP2C expression is prominently high in TSCs and at the early stages of EVT differentiation, gradually decreasing as TSCs differentiate into EVTs (**Fig. 2c**). Consistent with these findings, scRNA-seq data from

human placenta exhibited lower levels of TFAP2C expression in EVT_s compared to CT_s, showing approximately 35% and 87% reductions in Vento-Tormo et al.'s² and Suryawanshi et al.'s³ data sets, respectively (**Response Fig. 2a,b**). These observations were further supported by the more recent snRNA-seq data from 8-9 post-conceptual weeks trophoblasts⁴. As shown in **Response Fig. 2c**, the expression level of TFAP2C was lower in EVT subtypes (EVT-1, EVT-2, iEVT, and eEVT) compared to CT subtypes, including villous cytotrophoblasts (VCT), proliferating VCT (VCT-p), and CT cell columns (VCT-CCC, precursors of EVT-1 and EVT-2). Although the reduction degree varies, the bulk RNA-seq analysis of trophoblasts sorted from the human placenta indicated a lower expression of TFAP2C in EVT_s than in CT_s, with approximately 20% reduction (**Response Table 1**). Collectively, these data support the conclusion that TFAP2C expression levels are lower in EVT_s than in CT_s in the human placenta, albeit with some variation in the degree of reduction.

Among the late-stage TF_s, ASCL2 and NRIP1 exhibited higher expression in EVT_s compared to CT_s, while DLX5/6 and ZNF439 showed similar expression levels between CT_s and EVT_s in human first-trimester placenta (**Response Fig. 2a-c** and **Response Table 1**). This difference observed between *in vitro* and *in vivo* samples is also found in Okae et al.'s bulk RNA-seq data¹ (**Response Table 1**). While we accept some discrepancy between the human placenta and *in vitro* models, to confirm the consistent expression patterns of EVT regulators *in vitro* (from CT27), we differentiated additional TSC lines established (CT29, CT30, bTS5, and bTS11) into EVT. We first observed similar morphological changes upon differentiation (**Response Fig. 3a**) and found that the expression patterns of EVT TF_s during EVT differentiation are consistent across all tested TSC lines (**Response Fig. 3b**). Furthermore, we confirmed the similarity in protein expression patterns of EVT TF_s in TSC lines derived from CT (**Response Fig. 3c**). Notably, while we observed that the transcript levels of TFAP2C in EVT_s are 40%-50% of its levels in TSC_s, the levels of TFAP2C proteins in EVT_s are barely detectable (**Response Fig. 3c**).

Although *in vitro* systems do not perfectly mimic *in vivo* expression patterns, they provide valuable insights into the key regulators and regulatory mechanisms of biological processes. For example, studying somatic cell reprogramming *in vitro* enhances the understanding of drivers, essential factors, signaling pathways, and molecular mechanisms of cell fate control, despite the process not even occurring naturally *in vivo*. Although the expression patterns of some EVT regulators in TSC models may not fully resemble those in human placentas, investigating these EVT regulators in TSC models improves the understanding of their crucial roles and underlying mechanisms, considering their high expression in EVT_s in human placentas.

In summary, our additional analysis confirmed reduced TFAP2C expression levels in EVT_s compared to CT_s in the human placenta and TSC_s. Additionally, we observed strong expression of all the late-stage TF_s in EVT_s in the placenta. While there are some discrepancies in expression patterns between *in vitro* models and *in vivo* settings, we verified consistent expression patterns of EVT TF_s during the differentiation of multiple TSC lines. We have addressed this aspect in the Discussion section of the revised manuscript (page 11). In addition, we have included the morphological changes and expression patterns of EVT TF_s during the EVT differentiation of CT29 and CT30 in our revised manuscript as **Extended Data Fig. 2b-d** and the patterns of protein expression level of EVT TF_s in CT27 during EVT differentiation as **Fig. 2e**, with corresponding edits to the main text to reflect these additions (page 4).

Response Fig. 2: Expression of EVT TFs in various cell types in human first-trimester placenta.
a, Bar graphs showing expression levels of EVT TFs in different cell types of human first-trimester placenta, as determined by scRNA-seq data from (a) Vento-Tormo et al.² (normalized by human protein atlas) and (b) Suryawanshi et al.³ (analyzed by PlacentaCellEnrich tool). **c**, UMAP plot of snRNA-seq data from donor P13 trophoblast nuclei (approximately 8-9 post-conceptual weeks) in the maternal-fetal interface (n=37,675 nuclei)⁴, colored based on cell state information. The plots were generated using the Reproductive Cell Atlas provided by VenTo Lab (<https://www.reproductivecellatlas.org/>). The UMAP plots display the expression levels of TSC and EVT markers, TP63 and HLA-G, respectively, as well as EVT TFs in trophoblasts.

Response Table 1. Expression of EVT TFs in placenta, TSCs and their derivatives¹.

Gene symbol	Isolated from placentas		TSCs derived from CT		TSCs from blastocyst	
	CT	EVT	TS ^{CT}	EVT-TSCT	TS ^{blast}	EVT ^{TSblast}
TFAP2C	6.2	4.9	6.2	5.7	6.0	5.6
DLX5	6.4	5.7	1.3	6.1	1.6	6.4
DLX6	4.4	4.3	0.6	4.9	0.6	5.5
ASCL2	3.8	6.6	3.5	7.2	3.2	7.3
ZNF439	4.2	3.5	0.6	3.3	0.0	3.1
NRIP1	4.9	6.5	3.6	6.4	3.2	6.6

Response Fig. 3: Expression patterns of EVT TFs during EVT differentiation in different TSC lines.
a, Brightfield images of TSCs and differentiating cells to EVT at day 1 (EVT D1), 2 (EVT D2), 3 (EVT D3), 4 (EVT D4), 5 (EVT D5), 6 (EVT D6), 7 (EVT D7), and 8 (EVT D8) in five TSC lines. The TSC lines include cells derived from CT of first-trimester placenta (CT27, CT29, and CT30) and blastocyst (bTS5 and bTS11). Scale bar: 100 μ m. **b**, Relative mRNA expression (fold change) of TFAP2C and late-stage TFs during EVT differentiation in the five human TSC lines. The fold change was calculated relative to the expression in TSCs for each individual cell line. Error bars indicate mean \pm SD (n = 2). **c**, Western blot analysis showing the expression patterns of EVT TFs in the three CT TSC lines. HLA-G expression was measured to validate EVT differentiation. ACTB was used as a loading control. * ZNF439 (lower band).

1-5. The EVT migration assays used in Fig.3b is an important measure of functionality and should therefore include a quantitative readout.

In response to the reviewer's comment, the invasion assay results of the EVT factor KD and control cells in the original manuscript (**Fig. 3b** and **Response Fig. 4a**) were quantified by measuring the invasion area using the Image J program⁸ (**Response Fig. 4b**). We have added the corresponding graph and updated Method in the revised manuscript (**Fig. 3b**; page 29).

Response Fig. 4: Invasion ability of EVT following KD of each EVT regulator.

a, The invasion ability of EVTs was assessed after KD of individual EVT TFs, compared to control cells, using the matrigel chamber system. Scale bar: 100 μ m. **b**, The invasion area was calculated using Image J software⁸. Measurements were obtained from images taken in three independent biological replicates.

1-6a. In general, the KDs in Fig.3 should be characterized first in an unbiased way (i.e., not only in the context of the 4 clusters) and more comprehensively.

We regret not providing sufficient clarification. As described in our response to comment #1-3, in our original manuscript, we conducted unbiased analyses of the RNA-seq data from EVT factor KD cells using multiple methods, such as GSEA (**Fig. 3d** and **Extended Data Fig. 3g,h**), PCA (**Fig. 3f**), and GO analysis (**Fig. 3e,g**), without relying on the gene classes. We also utilized the four gene classes we defined to analyze the RNA-seq data (**Fig. 3c**). Noteworthy, the results obtained from other assays we performed in our original manuscript using EVT and CT marker gene sets defined from a scRNA-seq study of human

first-trimester placenta² and an EVT gene set from the Molecular Signatures Database^{6,7} were consistent with the results obtained from the analysis using the four gene classes (**Fig. 3d** and **Extended Data Fig. 3g**). This reinforces the validity of using the gene classes we defined to analyze the transcriptome of the EVT factor KD cells.

1-6b. How are other lineage markers affected by these KDs? Are the cells stuck in a progenitor state or do they deviate into other lineages (as reported for the ASCL2-KD, PMID:33649217)?

We greatly appreciate the reviewer for raising an excellent point. To investigate whether the KD cells are stuck in an early stage of differentiation, we performed an additional clustering analysis utilizing the DEGs during EVT differentiation. In this analysis, we used the log₂ fold change values of the RNA-seq data of EVT TF-KD cells and time-course EVT differentiating cells to account for potential batch effects and variability in the absolute expression value. As depicted in **Response Fig. 5a**, the late-stage TF-KD cells clustered together with TSC and cells in EVT on differentiation Day 1 (EVT D1), while the TFAP2C-KD cells showed a closer clustering with cells on differentiation Days 2 and 3. These findings suggest that although the absolute expression levels may vary among the KD cells for different EVT TFs and EVT differentiating cells, the expression patterns indicate that both late-stage TF-KD and TFAP2C-KD cells are arrested at the early stage of EVT differentiation.

In our original manuscript, we mentioned the expression patterns of TSC/CT marker genes in the RNA-seq data of the perturbed cells. The late-stage TF-KD cells and TFAP2C OE (Days 2 to 8 during EVT differentiation) cells showed higher expression levels of TSC-active and CT marker genes compared to the control cells, while TFAP2C KD cells showed lower expression levels of CT marker genes (**Fig. 3c, 4h,i** and **Extended Data Fig. 3g**). In response to the reviewer's suggestion, we reanalyzed the RNA-seq data and examined the expression levels of the ST gene set defined from a scRNA-seq study of the human placenta². Our findings were consistent with the previous report⁵, as we observed increased expression of ST lineage genes in ASCL2 KD cells compared to control cells (**Response Fig. 5b**). Likewise, TFAP2C KD/OE, DLX6 KD, and NRIP1 KD cells exhibited higher expression of ST genes compared to the individual controls (**Response Fig. 5b,c**). These findings are now included in the revised manuscript (**Extended Data Fig. 3h**; page 6).

Overall, the expression of CT marker genes was not downregulated adequately in all the late-stage TF-KD cells, while KD of DLX6, ASCL2, and NRIP1 resulted in higher expression of ST genes. The results suggest the potential regulatory effects of these factors on ST genes directly or indirectly. Notably, TFAP2C KD cells exhibited lower expression levels of CT genes (**Extended Data Fig. 3g**) and increased expression levels of ST marker genes compared to the control cells (**Response Fig. 5b**). These alterations are in conjunction with the inadequate induction of EVT marker genes, contributing to the KD cells displaying expression patterns similar to those observed in cells during the early stage of EVT differentiation.

Response Fig. 5: Expression patterns of EVT factor KD cells.

a, Heatmap showing the clusters of EVT factor KD cells and time-course EVT differentiating cells, based on the expression pattern of DEGs during EVT differentiation. **b** and **c**, GSEA results using ST marker genes defined from scRNA-seq study of first-trimester human placenta² to compare (**b**) EVT factor KD cells with control cells and (**c**) TFAP2C OE cells from day 2 to 8 during EVT differentiation (TFAP2C OE D2-8) with non-treated control cells.

1-7. As in Fig.3, the characterization of the TFAP2C-KD time course in Fig.4 does not clearly show what happens to other key lineage genes except HLA-G, MMP2 and TP63 during the differentiation. A heatmap containing a set of key trophoblast TFs and comparing their expression changes during the time course would be useful. What genes are impacted most by the KDs/OE and how does this relate back to trophoblast biology.

In our original manuscript, we conducted time-course TFAP2C KD experiments (**Fig. 4a, also shown in Response Fig. 6a**) and demonstrated that TFAP2C is required during the early stage of differentiation but is dispensable during the late stage for the induction of EVT-active genes (**Fig. 4b,d**). In response to the reviewer's suggestion, we performed additional RT-qPCR analysis on these time-course TFAP2C KD cells using additional representative marker genes for EVT, CT, and ST^{1-3, 9}. Consistent with the results

presented in our original manuscript (**Fig. 3d, 4b,d** and **Extended Data Fig. 3g**), EVT cells with TFAP2C KD from TSCs (KD-Early) exhibited lower expression of EVT and CT marker genes, while cells with TFAP2C KD starting from differentiation Day 3 (KD-Mid) or Day 5 (KD-Late) showed similar expression levels of those marker genes compared to the control cells (**Response Fig. 6b**). Furthermore, similar to our data shown in **Response Fig. 5b**, the expression levels of ST marker genes were higher in KD-Early cells than in control cells. Interestingly, KD-Mid cells also showed increased expression of ST genes, suggesting potential repressive effects of TFAP2C on ST genes during the mid-stage of differentiation (**Response Fig. 6b**).

We also confirmed that the downregulation of TFAP2C during the late stage of differentiation is necessary for proper EVT differentiation. In the original manuscript, we presented the data demonstrating impaired morphological changes and improper induction of EVT-active genes in TFAP2C OE cells starting from Day 2 of differentiation (**Fig 4e-j** and **Extended Data Fig. 4h,i**). To further monitor the changes in the expression of trophoblast marker genes over time as suggested by the Reviewer, we performed RT-qPCR experiments using TFAP2C OE cells collected at different time points after the TFAP2C induction starting on Day 2 of differentiation, following the same experimental conditions as described in our original manuscript (**Fig. 4a, also shown in Response Fig. 6a**). Consistent with our previous findings, TFAP2C OE cells monitored at EVT differentiation day 8 (D2-8 cells) exhibited impaired induction of EVT marker genes and higher expression levels of CT marker genes (**Fig. 4f-i, Extended Data Fig. 4h,i, and Response Fig. 6c**). This pattern was also observed from the D2-5 cells. Notably, the cells with brief TFAP2C OE monitored on Day 3 (D2-3 cells) displayed an overall opposite expression pattern of the tested genes compared to D2-8 or D2-5 cells, suggesting dynamic regulation of these genes during differentiation. Maintained levels of TFAP2C from Days 2 to 8 of differentiation increased expression levels of ST marker genes as confirmed by RNA-seq results (**Response Fig. 5c**). Consistent with this finding, RT-qPCR data demonstrated an overall increased expression of ST genes in Day 8 cells, while Days 3 and 5 cells showed mixed results.

This additional monitoring of CT, EVT, and ST markers in time-course TFAP2C KD and OE cells provides further confirmation of the finding presented in our original manuscript, indicating the requirement of TFAP2C during the early stage of EVT differentiation and the necessity for its downregulation during the late-stage differentiation. Interestingly, both KD and OE of TFAP2C resulted in increased expression of ST marker genes, suggesting potential regulatory roles of TFAP2C in ST genes. Given that opposite perturbations resulted in the induction of ST genes, we speculate that the regulatory mechanisms involved could be complex or indirect. Thus, exploring the roles of TFAP2C in the ST lineage and investigating the underlying mechanisms will be of interest to future research. The revised manuscript has incorporated these new results (**Extended Data Fig. 4b,j**; page 6-7).

Response Fig. 6: Expression levels of EVT, CT, and ST marker genes in TFAP2C KD and OE cells. **a**, Schematic representation of experimental design for time-course KD (top, light grey background) and OE (bottom, dark grey) of TFAP2C during EVT differentiation. **b**, Heatmap presenting the relative expression of EVT, CT, and ST marker genes in EVT day 8 cells infected with lentivirus particles expressing pLKO shRNA targeting TFAP2C at different time-points during EVT differentiation: day -1 (KD-Early), day 3 (KD-Mid), and day 5 (KD-Late). The fold change was calculated relative to EVT day 8 cells infected with lentivirus expressing a non-targeting sequence. **c**, Heatmap displaying the relative expression of EVT, CT, and ST marker genes in TFAP2C OE (Dox-treated) cells compared to individual non-treated control (-Dox control) cells. Dox (1 $\mu\text{g}/\text{mL}$) was treated on day 2 of EVT differentiation, and cells were collected on day 3 (D2-3), day 5 (D2-5), and day 8 (D2-8).

1-8. The integration of the ChIP-seq data from Fig.5 with the KD expression data from Fig.3/4 is surprisingly missing. This is essential to show which genes are directly bound and regulated/co-regulated by a given TF. What is the overlap of bound vs. misregulated genes? When are these bound genes expressed? To what extent TFAP2C and DLX6 share target genes overall and not just in specific clusters?

We appreciate the valuable comment provided by the reviewer. In our original manuscript, we presented data from an integrative analysis of ChIP-seq data for the late-stage TFs and RNA-seq results of cells upon KD of individual late-stage TFs (**Fig. 7c** and **Extended Data Fig. 7h**). In response to the reviewer's suggestion, we conducted additional analysis of the ChIP-seq data for TFAP2C and DLX6 and RNA-seq data. First, we investigated the target genes of TFAP2C defined in TSCs and DLX6 in mature EVT, respectively. As we suggested previously, there is a significant target overlap between TFAP2C and DLX6, although their targets were mapped at different stages of differentiation, indicating potential interconnection (**Response Fig. 7a**).

To gain a deeper understanding of the common target gene regulation, we performed an integrative analysis of the TFAP2C/DLX6 targets and the RNA-seq data from the KD cells of other EVT TFs. As presented, each KD cell exhibited similar impairments in the induction of EVT-active genes (**Fig. 3c,d**). We specifically investigated whether the DEGs resulting from the KD of individual EVT TFs are also direct targets of TFAP2C and DLX6. As shown in **Response Fig. 7b**, approximately 75%-84% of the DEGs upon each KD were directly bound by TFAP2C, DLX6, or both. Interestingly, around 49% and 60% of the upregulated and downregulated genes in each KD were common targets of TFAP2C and DLX6 (G2), suggesting functional similarities among the late-stage TFs and potential collaborative works between other the late-stage TFs and TFAP2C.

As requested, to gain further insight into the DEGs from KD cells that are common targets of TFAP2C and DLX6 (G2), we also examined their expression patterns during EVT differentiation. As presented in **Response Fig. 7c**, these targets include EVT-active genes (orange bars) and TSC-active genes (green bar). We have included parts of these results in **Extended Data Fig. 5g,i** and made corresponding revisions to the main text (page 8).

Response Fig. 7: Overlapping and distinct targets of TFAP2C and DLX6, and expression patterns of common targets of TFAP2C and DLX6 during EVT differentiation.

a, Venn diagram illustrating the distinct and shared target genes of TFAP2C in TSCs and DLX6 in EVTs. **b**, Bar graphs displaying the number of DEGs in EVTs following KD of individual EVT TFs that are directly bound by TFAP2C in TSCs (group 1, G1 targets), common to both TFAP2C and DLX6 (group 2, G2 targets), or specifically bound by DLX6 in EVT day 8 cells (group 3, G3 targets), allowing identification of DEGs directly regulated by TFAP2C and DLX6. **c**, Heatmap presenting the expression patterns of DEGs that are bound by both TFAP2C in TSCs and DLX6 in EVT D8 during EVT differentiation. Genes with decreased expression during differentiation are represented by the green bar, while genes active in EVTs are indicated by orange bars.

1-9. Along the same lines, while comprehensive ChIP-seq/RNA-seq analysis have been performed, they've not been comprehensively analyzed in an integrated manner. For instance, the authors make important conclusions (e.g., lines 263-268) and reference only genome browser snapshots without the global perspective (Fig.5d, Fig.6a and Extended Fig.6a,b).

We acknowledge the reviewer's comment and apologize for the inadequate explanation provided in our original manuscript. In lines 263–276, we intended to highlight the findings regarding the occupancy of TFAP2C in cis-regulatory elements of EVT-active genes in TSCs and during the early stages of EVT differentiation. We aimed to illustrate the concept of pioneer factor activity by presenting several examples of these binding patterns. However, by only referring to track images, it may have given the impression that our analysis relied solely on these images, and we previously accidentally omitted the reference figure information. We have now revised our manuscript to include appropriate reference figures, and it reads on page 8: "The pre-occupancy of TFAP2C for the future activation of EVT-active genes is reminiscent of pioneer factor activity (**Fig. 5d,g,h**). Indeed, the binding patterns of TFAP2C and DLX6 to target loci near late-stage TFs and EVT markers (HLA-G and MMP2) are consistent with this role (**Fig. 6a and Extended Data Fig. 6a,b**)."

1-10. Line235-237: Conclusion that "TFAP2C bound to its targets in self-renewing TSCs and cells in early-stage day 2, followed by decreased occupancy as differentiation progressed. On the other hand, we detected DLX6 target occupancy starting at day 5 of differentiation (Fig.5b)" is based on the number of peaks. Instead, it should be based on a quantitative analysis of TFAP2C and DLX6 binding (i.e., differential binding analysis).

We apologize for any confusion caused by the lack of clarity in our previous description. In the original manuscript, the number of peaks presented was determined through independent analysis of the TFAP2C and DLX6 ChIP-seq data using the MACS3 tool for peak calling, and the significant peaks were used to generate **Fig. 5b**. To provide further clarity and support our conclusions, we have generated new heatmaps that illustrate the binding patterns of TFAP2C and DLX6 in TSCs and differentiating cells into EVT at different time points (**Response Fig. 8**). The newly generated heatmaps demonstrate the decrease in TFAP2C binding during EVT differentiation and the binding of DLX6 in cells during the late stage of EVT differentiation. We have included these data in our revised manuscript as **Extended Data Fig. 5c,d**.

Response Fig. 8: Genomic binding loci of TFAP2C and DLX6 in TSC and differentiating cells towards EVT.

a and **b**, Heatmaps illustrate the occupancy signals of **(a)** TFAP2C and **(b)** DLX6 in TSCs and cells differentiating towards EVT at day 1 (EVT D1), day 2 (EVT D2), day 5 (EVT D5), and day 8 (EVT D8).

1-11. The display G1-G3 loci in Fig.3e,f is misleading. The authors should present data obtained for each time-point.

We suspect that the reviewer commented on **Fig. 5e,f**, and we apologize for any confusion caused by the lack of detailed explanation in the original manuscript. To clarify this, we provide a more thorough description of how we defined G1–G3 loci and plotted the ChIP-seq data of TFAP2C and DLX6 in TSCs and differentiating cells toward EVTs.

In our original manuscript, we initially observed that TFAP2C binds to cis-regulatory elements of genes that are not actively transcribed in TSCs or during the early stage of EVT differentiation but become activated in EVTs. To investigate whether TFAP2C binds to the regulatory elements of EVT-active genes when they are not even active, we conducted a differential binding analysis by comparing the target loci of TFAP2C in TSCs and DLX6 in EVTs. This analysis allowed us to identify three distinct groups of loci: unique target loci of TFAP2C in TSCs (Group 1, G1), TFAP2C-DLX6 common target loci (Group 2, G2), and unique target loci of DLX6 in EVTs (Group 3, G3; **Extended Data Fig. 5h**). To visualize the binding patterns of TFAP2C and DLX6 in TSCs and EVT differentiating cells, we created plots where all the ChIP-seq peaks of TFAP2C and DLX6 in TSCs, EVT D2, EVT D5, and EVT D8 are aligned to the same order of genomic loci (**Fig. 5e,f**). Each row in the plot represents the same genomic locus, allowing for observing the changes in TFAP2C and DLX6 binding patterns at different time points. It is important to note that we included all the ChIP-seq peaks obtained at each time point to ensure a comprehensive representation of the binding events. We apologize for any confusion caused by the lack of clarity in the original manuscript, and we appreciate the opportunity to provide a more detailed explanation of the analysis and data representation. We have edited the main text to clearly describe how we defined G1–G3 loci and plotted the ChIP-seq data of TFAP2C and DLX6 at different time points (page 8). Now it reads, “To further explore this relationship, we conducted a differential binding analysis comparing target loci of TFAP2C in TSCs and DLX6 in EVTs. This analysis allowed us to identify three distinct groups of loci: unique target loci of TFAP2C in TSCs (Group 1, G1), TFAP2C-DLX6 common target loci (Group 2, G2), and unique target loci of DLX6 in EVTs (Group 3, G3; **Extended Data Fig. 5h**). We then plotted the ChIP-seq peaks of TFAP2C and DLX6 according to the differentiation stage and target groups (**Fig. 5e,f**). Each row in the plot

represents the same genomic locus, allowing for observing the changes in TFAP2C and DLX6 binding patterns at different time points.”

1-12. A part of the paper focuses on the putative TFAP2C pioneer activity without discussing/excluding other possibilities. What criteria does a pioneer factor need to fulfill? Is it possible that TFAP2C acts as a repressor of late EVT genes, instead as a pioneer-activator? Pioneer factors do not just prime genes for later expression but bring in immediate change to the local chromatin environment by recruiting other chromatin modifying/remodeling activities, which has not been demonstrated.

We appreciate the reviewer’s critical comment regarding the putative pioneer factor activity of TFAP2C and the need to consider alternative possibilities. We acknowledge that the criteria for a pioneer factor go beyond priming genes for later expression and involve bringing about immediate changes to the local chromatin environment through the recruitment of other chromatin modifying/remodeling activities^{10, 11}, which we did not investigate in our original manuscript.

Regarding the potential role of TFAP2C as a repressor of EVT genes, we have demonstrated in our recent paper¹² that TFAP2C has dual roles and functions as both an activator and a repressor on its targets in self-renewing TSCs. However, in the context of EVT-active genes, we rule out the possibility of TFAP2C as a repressor as KD of TFAP2C results in insufficient upregulation of EVT-active genes during differentiation (**Fig. 3c,d** and **Extended Data Fig. 3a,c**). The discrepancy may arise from context-specific or target-specific roles of TFAP2C. Interestingly, **Response Data Fig. 6b** indicated that TFAP2C KD leads to increased expression of ST marker genes. While it is unclear if this regulation is direct or indirect, the data suggest potential repressor functions of TFAP2C on other classes of targets during EVT differentiation. Collectively, our results indicate that TFAP2C does not act as a repressor, at least on EVT-active genes, during EVT differentiation.

The original manuscript showed that TFAP2C binds to the genomic loci with low ATAC-seq signals in TSCs (**Fig. 6b** and **Response Fig. 9**). The ATAC-seq signals on these loci increase in EVT cells (**Fig. 6e** and **Response Fig. 9b**). The genes associated with these loci are associated with EVT-related terms (**Fig. 6c**). To investigate the potential pioneer factor activity of TFAP2C further, we monitored other enhancer signals on these TFAP2C ChIP-seq unique loci in TSCs and EVT cells. As shown in **Response Fig. 9c**, we observed increased signals of enhancer markers, including H3K27ac, H3K4me1, and P300, at the TFAP2C unique loci defined in TSCs in EVT cells. The results provide further support for the potential pioneer factor activity of TFAP2C. We have included this result in our revised manuscript and edited the main text accordingly (**Fig. 6f**, page 9).

We acknowledge that further investigation regarding recruiting other chromatin modifiers or remodelers by TFAP2C is necessary to establish its pioneer activity fully. Considering this, we have taken a more cautious approach in the revised manuscript by toning down the assertion of TFAP2C’s pioneer factor activity. We have addressed this in the Discussion section of our revised manuscript (page 12).

Response Fig. 9: Changes in enhancer signals at TFAP2C ChIP-seq unique loci between TSCs and EVT.

a, Heatmaps showing TFAP2C ChIP-seq unique loci, TFAP2C ChIP-seq/ATAC-seq common loci, and ATAC-seq unique loci in TSCs. TFAP2C ChIP-seq unique loci are indicated by a black bar. PC represents peak center. **b** and **c**, (**b**) ATAC-seq signals and (**c**) enhancer signals, including H3K27ac, H3K4me1, and P300 at TFAP2C ChIP-seq unique loci in TSCs and EVT. Statistical significance was determined by two-tailed Wilcoxon rank-sum test (**p < 0.01, ***p < 0.001).

1-13. A lot (in my view too much) prominence is given to the general GO-term analysis without mentioning specific gene examples. Instead, the authors should focus on what genes are affected and how this impacts EVT biology. Are there new interesting target genes that are bound by TFAP2C/DLX6? What critical stage in EVT development/function may they regulate?

We appreciate the reviewer for bringing this to our attention. Following the reviewer's suggestion, we thoroughly examined the DEGs upon KD of EVT TFs and have now included specific examples of these genes in our revised manuscript (pages 5-6 and 8):

1) In the "Both early and late-stage TFs are required for EVT differentiation" in the Results section (page 5)

"Consistently, metalloproteinases, such as MMP2^{13, 14} and ADAM19^{15, 16}, as well as previously known regulators of invasion, such as CXCR4^{17, 18} and EGFR-AS1¹⁹, were downregulated in the EVT factor KD cells. Furthermore, NOTUM, an extracellular Wnt deacylase^{20, 21}, was commonly downregulated in the EVT TF KD cells, suggesting impaired EVT differentiation, as Wnt inhibition is necessary for EVT lineage formation^{1, 22}."

2) In the "TFAP2C KD leads to a unique expression pattern consistent with a role in TSC self-renewal" in the Results section on page 6

"Similarly, several genes associated with cell cycle regulation, such as CDK1 and CCND1, along with the factors involved in cell proliferation, including MYBL2²³ and TOP2A (DNA topoisomerase 2-alpha, which plays a role in DNA replication and cell division), were significantly upregulated in the late-stage TF KD

cells. Moreover, the late-stage TF KD cells exhibited sustained expression of the genes associated with epithelial cells, including EPCAM²⁴, CDH1²⁵, and TJP1 (tight junction protein 1). Additionally, the key genes involved in Wnt signaling, such as AXIN2, FZD5, LRP5, and TCF7L1²⁶⁻²⁸, displayed increased expression in the late-stage TF KD cells, implying that the KD cells still retained a TSC-like gene expression program.”

Furthermore, we have highlighted in our revised manuscript that both TFAP2C and DLX6 occupy the genes associated with critical EVT functions at different time points. These additional examples further support our findings and have been included in our revised manuscript (“TFAP2C primes late-stage TFs and EVT-active genes during the early EVT differentiation”) in the Results section (page 8):

“We observed that matrix metalloproteinases, specifically MMP2 and MMP9, which are essential for invasion into the extracellular matrix, are associated with G2 loci. Additionally, we identified master regulators of epithelial-to-mesenchymal transition (EMT), namely SNAI1 and TWIST1²⁹, as targets of both TFAP2C and DLX6. This suggests that TFAP2C and DLX6 play crucial roles in controlling the genes responsible for the invasive function of EVTs. Notably, we also noticed that the angiogenic factor VEGF and its receptors KDR and FLT1³⁰ are occupied by both TFAP2C and DLX6, suggesting that both early- and late-stage TFs are involved in the activation of genes associated with spiral artery remodeling, a critical function of EVT.”

1-14. The ability of DLX5/6 to activate EMT genes should also be demonstrated in EVTs and not just cancer cells. The statement that SNAI1 might be an EVT-specific EMT TF is misleading as SNAI1 is a well-known general EMT regulator in different contexts. The general connection between cancer EMT and EVT differentiation could be explored a bit more by data mining. For instance, are there other relevant expression signatures shared by cancers and the EVT lineage? What about TFAP2C in this context? Is it also upstream of DLX genes in cancers? The general idea of faulty re-activation of extra-embryonic genes could be discussed as a wider context.

We greatly appreciate the reviewer's insightful suggestion and acknowledge the potential benefits of including publicly available human datasets to enrich our findings in the manuscript. However, after carefully considering the extensive experimentation required to validate our results and considering the suggestion from Reviewer #2 to remove this section, we have decided to exclude it from the current manuscript. Instead, we intend to pursue this aspect as a separate project for future research.

1-15. The Discussion needs a substantial revision in terms of structure. More detailed discussion of pioneer vs. possible repressor functions might be beneficial. It is requested that the authors discuss the results in the context of EVT biology beyond just general aspects of EMT.

We appreciate the suggestions provided by the reviewer regarding the Discussion section. We have made several additions and edits to enhance the clarity and depth of the discussion. First, we added new paragraphs addressing the following topics: 1) the *in vitro* model we employed and its advantages in mimicking *in vivo* EVT differentiation and potential EVT lineages, 2) the discrepancies between *in vitro* models and human placenta, and 3) the potential pioneer factor activities of TFAP2C and its possible repressor activities. Additionally, we revised the paragraph that discusses the roles of TFAP2C in the self-

renewal of TSCs versus differentiation, ensuring a more accurate and comprehensive description of TFAP2C's functions in these processes.

Minor comments:

1-M1. Fig.2c is unreadable.

We have revised **Fig. 2c** in our manuscript, and the updated figure can be found in **Extended Response Data Fig. 1** within this rebuttal.

1-M2. The original scans for the Western blots are missing.

We have included the original scans of the Western blot results in our revised manuscript, and the images can be also found in **Extended Response Data Fig. 2** within this rebuttal.

1-M3. Fig.4j and Fig.1f: x-axis labeling would be beneficial.

We have revised the figures according to the Reviewer's suggestion in our revised manuscript, and the updated version is also available in **Extended Response Data Fig. 3**.

1-M4. The EVT KD labeling in Fig.7f is unreadable.

We have decided to exclude the EMT section, as indicated in our response to the Reviewer comment #1-14 above. Therefore, we have excluded **Fig. 7f** from our revised manuscript.

1-M5. All genome browser snap-shots lack scale.

We have revised the track images in **Fig. 2b, 5d, 6a, 7a** and **Extended Data Fig. 6a, 6b, 7d** in our manuscript. The updated versions now include scale and locus information. The updated figures are also available in **Extended Response Data Fig. 4** within this rebuttal.

1-6M. The authors should specify settings of their bioinformatic analyses, also when standard settings were used.

We have added specific settings of the bioinformatic analysis in the revised manuscript. The revised method is also shown in **Extended Response Data Text. 1**.

Reviewer #2

The Manuscript by Kim et al. used human trophoblast stem cells (hTSC) as a model system to understand the transcriptional control of human extravillous trophoblast (EVT) stem cells. Authors concluded a multi-stage transcriptional mechanism of EVT differentiation process, in which transcription factor TFAP2C instigates the EVT differentiation process and later other transcription factors, such as DLX5/6, mediates the late stages of the EVT differentiating process. The strength of the manuscript is complementary genomics approaches that provide an understanding of longitudinal transcriptional mechanisms during EVT differentiation. However, the manuscript has several flaws in designing experiments, includes contradictory data compared to published reports and is extremely premature.

We sincerely appreciate the reviewer's critical and valuable comments on our manuscript and the opportunity to address the issues raised. The reviewer's feedback has been instrumental in improving the clarity of our work. We have carefully considered each comment and revised our manuscript to address the concerns raised. We hope that the revised manuscript adequately addresses the reviewer's concerns and enhances the quality of our research.

My concerns are mentioned below:

2-1a. The overwhelming issue with the manuscript is the significance of the findings. In Figure 1, authors presented transcriptomics data in early vs. late stages of EVT differentiation. However, there is no effort to correlate early stage EVTs and late stage EVTs with respect to the EVT populations at human placentation sites. It is true that the present concept accepts multiple EVT populations *in vivo*, such as proximal and distal placental EVTs as well as Interstitial and endovascular EVTs in the maternal uterine compartment. However, the manuscript lacks any data (Such as gene expression verification of class 1-4 genes) with human tissues to correlate early vs. late EVTs with respect to the actual EVT populations in a placentation site.

We appreciate the thoughtful comments provided by the Reviewer. In our original manuscript, we employed a well-established protocol for differentiating TSCs into EVTs, which exhibited a transcriptome profile similar to that of EVTs derived from the human placenta¹. However, it remains unclear whether this *in vitro* differentiation protocol could precisely recapitulate the differentiation process of progenitor cells into EVTs within the human placenta. Recently, Arutyunyan et al. conducted spatial transcriptomics analysis and snRNA-seq of human placenta at 8–9 post-conceptual weeks, identifying marker genes for various trophoblast populations⁴.

Considering their findings of the trophoblast subtype markers *in vivo*, we performed additional analysis of our RNA-seq data from the time-course EVT differentiation. As shown in **Response Fig. 10a**, the expression of marker genes associated with villous cytotrophoblasts (VCTs), the progenitor populations, was observed to be active in undifferentiated TSCs and gradually decreased as *in vitro* EVT differentiation progressed. The marker genes representing early-stage EVT differentiation in the human placenta, such as VCT proliferative cells (VTC-p) and CT cell column VCT (VCT-CCC), exhibited activation during the early stage of *in vitro* EVT differentiation. Additionally, the markers for intermediate-stage EVT populations *in vivo*, such as EVT-1 and EVT-2, were upregulated during Days 2–4 and 5–8, mirroring the *in vivo*

differentiation process. Similar to previous findings indicating that *in vitro* differentiated EVT cells exhibit interstitial EVT (iEVT) signatures but lack endovascular EVT (eEVT) signatures⁴, our analysis revealed the activation of markers associated with iEVTs during *in vitro* differentiated EVT cells. In contrast, the markers specific to eEVTs were not activated (**Response Fig. 10b**).

Therefore, although the *in vitro* system may not perfectly replicate the level of maturity observed *in vivo*⁴, the observed expression patterns of trophoblast marker genes during EVT differentiation imply that *in vitro* EVT cells mimic the cells found in the human placenta, with a predominant bias towards iEVT population. We have incorporated this point into the Discussion section of the revised manuscript (page 11).

Response Fig. 10: Expression patterns of trophoblast marker genes during *in vitro* EVT differentiation.

a, Heatmap displaying the expression patterns of trophoblast marker genes identified by Arutyunyan et al.⁴ through snRNA-seq analysis of P13 trophoblast nuclei (approximately 8-9 post-conceptual weeks) in the maternal-fetal interface (n=37,675 nuclei). Marker genes for villous cytotrophoblast (VCT), VCT proliferative cells (VCT-p), CT cell column VCT (VCT-CCC), two distinct EVT differentiating cells (EVT-1 and EVT-2), interstitial EVT (iEVT), and endovascular EVT (eEVT) are color-coded. The values represent the log₂-transformed count compared to the average expression level. **b**, Expression level of iEVT and eEVT marker genes in the RNA-seq data from time-course samples during EVT differentiation. The count values were utilized to generate the graph, representing the absolute expression levels.

2-1b. The major issue is variable gene expression patterns in human TSCs reported here with existing data. The available gene expression data by Okae et al. in initial Cell Stem Cell manuscript as well as multiple available single-cell RNA seq data clearly shows that TFAP2C is abundantly expressed in primary cytotrophoblasts (CTBs) and EVT. More importantly, unlike the findings here, TFAP2C expression is maintained in fully differentiated EVTs. Similarly, DLX5/6 are as highly expressed in CTBs as they are in EVTs and ZNF439 expression is suppressed in primary EVTs. Thus, given contradictory observations regarding expressions of key factors in this study vs. primary trophoblasts, without any *in vivo* validation, the whole manuscript seems an intellectual exercise. In relation to this, it is important to include FPKM values and fold changes of the genes in supplementary table (which only contain gene names). Otherwise, the information seems incomplete.

We appreciate the Reviewer for raising this important point, which was also raised by Reviewer #1. Regarding the expression levels of EVT TFs in the human placenta, please refer to our response to the comment #1-4 for further details. Briefly, we utilized publicly available bulk RNA-seq, scRNA-seq, and snRNA-seq data from the human placenta to assess the expression levels of EVT TFs. Although DLX5/6 and ZNF439 showed comparable expression levels between CTs and EVTs, we confirmed that all the late-stage TFs were highly expressed in EVTs in the human placenta (**Response Fig. 2**). Additionally, we observed lower expression levels of TFAP2C in EVTs compared to CTs, although the degree of difference varied (**Response Fig. 2** and **Response Table 1**). To further validate the expression patterns of EVT TFs, we performed additional experiments using multiple established TSC lines and validated consistent expression patterns of EVT TFs (both transcript and protein), at least in these TSC lines (**Response Fig. 3**). Therefore, despite discrepancies between the *in vitro* model and the human placenta, the late-stage TFs are highly expressed in EVTs in human placenta, and EVT TFs exhibit similar expression patterns across multiple TSC lines. In addition, we confirmed the similar KD outcomes of the tested TFs in multiple TSC lines (**Response Fig. 18**).

Regarding the normalized value of the RNA-seq data, we previously provided count values along with gene names in the GEO database (GSE212266, token: qjarecwgpdaifjqr). In response to the reviewer's suggestion, we have now included the count information for all the RNA-seq data in our manuscript as **Supplementary Table 1**. Additionally, we have included log₂ fold changes of class 1–4 genes in **Supplementary Table 2** for further clarity.

2-2a. The manuscript includes only Histone H3K27 modification to define enhancer signature. This experiments need verification with other marks such as H3K4Me1 and H3K4Me3. In addition, at least one non-histone signature needs to be included to identify super-enhancers. Otherwise, the enhancer dynamics prediction is incomplete. The experimental procedure needs to include more detail about how enhancer mapping was done. The supplementary table needs to indicate genomic locations of enhancers and super enhancers with respect to nearby genes.

The original manuscript employed H3K27ac signals to define enhancers due to its known superiority in identifying enhancers with minimal false positives³¹. To validate the enhancers identified using H3K27ac signals from our original manuscript, we conducted additional ChIP-seq experiments for other well-established enhancer markers, including P300 and Med1 occupancy and H3K4me1 signatures³¹, in TSCs, EVT D3, and EVT D8. The results of these experiments showed overall similarities in the enhancer landscape. Specifically, we observed co-occupancy of P300 and Med1 at the enhancer loci, along with enriched signals of H3K27ac and H3K4me1 close to these binding sites (**Response Fig. 11a**). These findings further support the reliability and accuracy of the enhancer identification based on H3K27ac signals in our original manuscript.

To further validate the SEs identified in our original manuscript, we utilized P300 and Med1 ChIP-seq data as predictive tools. Since H3K4me1 is known to mark both active and primed enhancers^{32, 33}, we did not incorporate H3K4me1 data for SE mapping. As depicted in **Response Fig. 11b**, there are overlapping SE-associated genes identified by different enhancer signals. However, the overlap among these genes was not extensive, which is consistent with a previous report in mouse embryonic stem cells³¹ (**Response Fig. 11c**). Nevertheless, we found that TFAP2C, an early-stage TF, was associated with SEs defined by all the enhancer marks, including H3K27ac, P300, and Med1, in both TSCs and EVT D3 (**Response Fig. 11b** and **Supplementary Table 3**). As for the late-stage TFs, ASCL2, ZNF439, and NRIP1 were associated with SEs predicted by all the tested enhancer ChIP-seq data. DLX5 and DLX6 were identified as SE-associated genes predicted by H3K27ac and P300 signals, specifically in EVT D8 cells. The quality of the Med1 ChIP-seq data could have influenced the results, as it exhibited a lower number of peaks with weaker signals (**Response Fig. 11d**).

In summary, mapping enhancers and SEs using the H3K27ac signature in our original manuscript is a reliable and robust approach, and we have now provided additional details about enhancer/SE mapping in the revised Methods section. Furthermore, we have included a comprehensive list of SEs along with their location information and associated genes in **Supplementary Table 3**. We have also uploaded the newly generated ChIP-seq data of P300, Med1, and H3K4me1 in TSCs, EVT D3, and EVT D8 to GEO (GSE212265, token: atgdeukgbjydvcf). Additionally, we have incorporated enhancer information into our revised manuscript as **Extended Data Fig. 1f** and edited the main text accordingly.

Response Fig. 11: Validation of enhancers defined by H3K27ac occupancies.

a, Heatmap depicting enhancer signals revealed by ChIP-seq analysis of H3K27ac, P300, Med1, H3K4me1, and active promoter signal of H3K4me3 in TSCs and differentiating cells toward EVT at day 3 (EVT D3) and day 8 (EVT D8). The ChIP-seq signals of P300 were sorted first, and the peaks from other ChIP-seq experiments were plotted to the P300 binding loci. **b**, Venn diagrams illustrating the overlapping and unique sets of SE-associated genes predicted by H3K27ac, P300, and Med1 enhancer mapping in TSC, EVT D3, and EVT D8 cells. **c**, Data from Hnisz et al.³¹. Venn diagrams showing the overlap between (left) SEs and (right) SE-associated genes identified using different markers in mouse embryonic stem cells. **d**, Gene track view displaying the binding loci of H3K27ac, P300, Med1, and H3K4me3 near late-stage TFs (DLX5 and DLX6) and EVT-active genes (MMP2 and FN1).

2-2b. It is not clear what is the cut-off distance of super-enhancers (The 12.5kb spanning and a minimal distance of 2.5 kb from TSS is not complete information). It seems several of the genes may have multiple nearby SEs. The authors should provide more information for better understanding of the manuscript.

We appreciate the reviewer's comment. In our study, we identify SEs using the ROSE program^{34, 35}, which provides two options, -s (stitching distance) and -t (TSS exclusion zone size). For our analysis, we utilized the parameters of -s 12500 and -t 2500, which exclude peaks within 2,500 bp from the transcription start site (TSS) to avoid capturing promoter peaks. As for our response, we have now included these details in the Methods section of the revised manuscript to provide more detailed information regarding SE mapping (page 27). The modified method is also shown in **Extended Response Data Text 1**.

Furthermore, we apologize for the redundant gene names in **Supplementary Table 2**. As accurately pointed out by the reviewer, this occurred because a gene can be associated with multiple SEs. Considering this feedback and the reviewer's comment #2-2a, we have provided more detailed information about SEs in **Supplementary Table 3**. The updated table includes the loci and associated gene information of SEs defined by multiple enhancer markers, ensuring more precise details on SEs.

2-3. The loss-of-function analyses of TFAP2C and Other genes are also of major concern. The efficiency of the time-course approach of TFAP2C-depletion is not clear (Did not notice any time-course of expression data and the methods section lacks the detail description of the RNAi approach). The approach is flawed as it creates window of extreme variability in day 0 vs. day 3 vs. day 5 cells. A better experimental approach, in which TFAP2C is conditionally depleted (using inducible shRNA) over different time-points need to be used.

We appreciate the reviewer's critical comment. Following the reviewer's suggestion, we have attempted to conditionally deplete TFAP2C using the EZ-Tet-pLKO inducible KD system³⁶. However, despite multiple trials, the efficiency of the inducible KD was not as successful as the one we achieved in our original manuscript (**Response Fig. 12a** and **Fig. 4b**).

Alternatively, to precisely determine the timing of TFAP2C KD after infection with lentivirus particles, we performed a test by collecting cells at 6 h, 12 h, 24 h, and 48 h post-infection and monitored the expression levels of TFAP2C using RT-qPCR. As shown in **Response Fig. 12b**, the TFAP2C expression levels started to downregulate at 12h post-infection, and the best KD efficiency remained stable from 24 h post-infection, as the TFAP2C levels of cells collected at 24 h and 48 h were similar. Based on the results, we concluded that at least in the TFAP2C KD case, it takes approximately 24 h to achieve the best KD with lentivirus-based shRNA approaches. As we treated cells with virus-containing media on Day -1 (KD-Early), Day 3 (KD-Mid), and Day 5 (KD-Late) in our original manuscript, it corresponds to the KD of TFAP2C on Day 0, 4, and 6 of EVT differentiation, respectively.

The KD timing-related information is also considered in our responses to the reviewer's comments #2-4 and #2-5a. In addition, we have added details of experimental procedures in the revised Methods section. The revised text is also shown in **Extended Response Data Text 1**.

Response Fig. 12: KD efficiencies of the EZ-Tet-pLKO inducible system and KD rate of the pLKO system targeting TFAP2C.

a and **b**, Relative TFAP2C expression in **(a)** TFAP2C KD TSCs using EZ-Tet-pLKO inducible KD system (Dox 1 µg/ml for 2 days) and **(b)** TSC at 6 h, 12 h, 24 h, and 48 h after infection with lentivirus particles expressing pLKO shRNA targeting TFAP2C. Non-treated control cells (-Dox control) and TSCs infected with non-targeting lentivirus particles were used as control, respectively. Error bars indicate mean ± SD (n = 2 for the 48 h samples, n = 3 for the others).

2-4. As TFAP2C is important to maintain the self-renewal in human TSCs, the lack of EVT differentiation in day 0 depleted cells may have other conclusions, such as extreme cell stress, induction of apoptosis, promotion of syncytiotrophoblast differentiation etc. These aspects are overlooked.

We appreciate the reviewer's insightful comment. In our original manuscript, lentivirus particles targeting TFAP2C were administered to TSCs, and EVT differentiation was initiated after 18 h. However, as the reviewer pointed out in the previous comment (#2-3), this approach does not allow precise control of the timing of TFAP2C KD in TSCs, leaving room for the possibility that TFAP2C KD in TSCs may influence further downstream cellular physiology, ultimately impeding EVT formation. To address this concern and distinguish between the effects of TFAP2C KD in TSCs and during EVT differentiation, we sought to perform additional KD experiments at multiple time points during the early stage of EVT differentiation.

Lentivirus particles targeting TFAP2C were administered on Days -1, 0, 1, 2, and 3 of differentiation. Based on the KD timing upon virus treatment (**response to the Reviewer's comment #2-3** and **Response Fig. 12b**), these time points correspond to TFAP2C KD on Days 0, 1, 2, 3, and 4, respectively. As shown in **Response Figure 13a**, the cells subjected to TFAP2C KD at days 0, 1, and 2 exhibited similar impairments in the induction of EVT markers (HLA-G and MMP2). However, TFAP2C KD on Days 3 and 4 did not lead to defects in the induction of EVT marker genes, confirming our previous observation of the requirement of TFAP2C during the early stage of EVT differentiation (**Fig. 4b-d** and **Extended Data Fig. 4a,b**).

In addition, as pointed out by the reviewer, our recent study¹² detected TFAP2C KD-induced apoptosis in self-renewing TSCs after 4 days of lentivirus infection. However, for the current study, we differentiated the cells after 18 h of infection. To assess cellular apoptosis at this specific time point, we conducted flow cytometry to measure apoptosis markers. As shown in **Response Fig. 13b**, TSCs infected with lentivirus targeting TFAP2C exhibited similar levels of Annexin V signals and caspase-3 activities compared to the control cells, suggesting that apoptosis was not significantly induced in TFAP2C KD cells after 18 h of infection, which aligns with the timing of EVT differentiation initiation in our original manuscript.

Overall, the newly obtained data suggested that the alteration in EVT differentiation resulted from the TFAP2C depletion during EVT differentiation. The lowered level of TFAP2C in TSCs is not the primary cause of the defects in EVT differentiation. Instead, TFAP2C plays a crucial role during the early stage of EVT differentiation. These results have been added to **Extended Data Fig. 4c,d**, and we revised the main text accordingly (page 7).

Response Fig. 13: Time-course KD of TFAP2C and its effect on EVT differentiation and apoptosis levels in TSCs following 18 h of lentivirus infection.

a, Relative mRNA expression levels of TFAP2C and EVT markers (HLA-G and MMP2) in EVT D8 cells infected with lentivirus particles expressing pLKO shRNA targeting TFAP2C at various time-points (day 1, 0, 1, 2, and 3) during EVT differentiation. The fold change was calculated relative to EVT D8 cells infected with lentivirus expressing a non-targeting sequence. Error bars indicate mean \pm SD ($n = 3$). Statistical significance was determined using Student's t-test (* $p < 0.05$, *** $p < 0.001$). **b**, Flow cytometry density plots illustrating TSCs displaying fluorescent signals from annexin V (conjugated to CF594) and caspase-3 activity (NucView 488) following 18 h of infection with lentivirus targeting TFAP2C. Staurosporine, a potent inducer of apoptosis, was used as control.

2-5a. If DLX5/6 and NRIP1 (which is induced upon EVT differentiation) are late acting factors, why they are depleted at day 0?

We appreciate the reviewer for bringing up this point. Based on our time-course transcriptome analysis showing a gradual increase in the expression of late-stage EVT TFs, we attempted to deplete these TFs when EVT differentiation starts in our original manuscript. Following the reviewer's comment, we conducted additional KD experiments by targeting individual late-stage TFs during the late-stage (on Days 3 and 5) of EVT differentiation. This corresponds to KD on Day 4 and Day 6, respectively, considering the results indicating the timing of KD (**Response to the Reviewer's comment #2-3** and **Response Fig. 12b**). As shown in **Response Fig. 14a,b**, the depletion of late-stage TFs on Days 3 and 5 also hindered the induction of EVT marker genes, including HLA-G and MMP2. These results confirmed that the late-stage TFs are necessary for activating EVT-active genes during the late stages of differentiation. We have added these new results to **Extended Data Fig. 4e** in our revised manuscript and made the necessary revisions in the main text accordingly (page 7).

Response Fig. 14: Impaired EVT differentiation upon KD of late-stage TFs at mid- and late-stage of differentiation.

a and b, KD efficiency of late-stage TFs and the expression levels of EVT marker genes, including HLA-G and MMP2, in EVT day 8 cells infected with lentiviral particles expressing pLKO shRNA targeting individual late-stage TFs at (a) day 3 and (b) day 5. The fold change was calculated relative to EVT day 8 cells infected with lentivirus expressing a non-targeting sequence. Error bars indicate mean \pm SD (n = 3). Statistical significance was determined using Student's t-test (*p < 0.05, **p < 0.01, ***p < 0.001).

2-5b. Also, the data needs western-blot expression analyses of target genes at different days (Fig. 3A includes western blot but is not in a time-dependent manner).

Following the reviewer's request, we conducted additional Western blot analysis with EVT factor KD cells collected at different time points during EVT differentiation. The experimental settings were consistent with those used for **Fig 3a** of our original manuscript, and the cells were collected on Days 3, 5, and 8 for Western blot analysis.

Consistent with the data presented in our original manuscript in **Fig. 3a**, the KD of EVT TFs resulted in impaired induction of EVT marker genes, HLA-G and MMP2, in the KD cells compared to the control cells on differentiation Day 8 (**Response Fig. 15**). Importantly, we also observed impaired induction of EVT marker proteins in the late-stage TF KD cells on Days 3 and 5, compared to their respective controls. The results suggest that the late-stage TFs are required during EVT differentiation, aligning with their gradual increase in expression throughout EVT differentiation (**Fig 2c** and **Response Fig. 3b,c**). In the revised manuscript, we have replaced **Fig. 3a** from the original manuscript with **Response Fig. 15**.

Response Fig. 15: KD of EVT TFs impairs the protein expression of EVT marker genes.

Western blot results demonstrating the KD efficiency and expression levels of EVT marker genes, including HLA-G and MMP2, in differentiating cells toward EVT at day 3 (D3), day 5 (D5), and day 8 (D8) following KD of each EVT regulator. EVTs infected by lentivirus expressing non-targeting shRNA were used as control.

2-6. The ATAC-seq data showing TFAP2C occupied regions have low accessibility needs better validation. Again the expression variability is a concern. TFAP2C is abundantly expressed in undifferentiated TSCs and is suppressed in late EVT. Also, as mentioned above, DLX5/6 are highly expressed in undifferentiated cells. Thus, the data seems contradictory.

We appreciate the reviewer's insightful comment. To validate the ATAC-seq data demonstrating TFAP2C binding on genomic loci with low ATAC-seq signals near EVT-active genes in TSCs, we conducted additional ATAC-qPCR analysis. We examined the loci with varying degrees of openness in TSCs, including those with very weak ATAC-seq signals near olfactory receptor genes (inactive), weak ATAC-seq signals near EVT-active genes not active in TSCs (EVT-active), strong ATAC-seq signals near TSC marker genes (TSC-active), and ribosomal protein-coding genes (TSC/EVT-active) (**Response Fig. 16a**). The ATAC-qPCR results in **Response Fig. 16b** confirmed that the genomic loci near EVT-active genes bound by TFAP2C in TSCs (red bars) exhibit overall lower openness compared to the TSC-active loci (blue bars) and TSC/EVT-active loci (green bars), while still being more open than closed loci near olfactory receptor genes (inactive, black bars). The results provide further evidence supporting our original observation that TFAP2C binds to the regions of relatively weak openness near EVT-active genes in TSCs, even when these genes are not transcriptionally active. We have incorporated these results into our revised manuscript as **Fig. 6d** and described the experimental procedure in the revised Methods section (pages 9 and 26-27; Also shown in **Extended Response Data Text 1**).

As we discussed extensively in our responses to the reviewer's comments #1-4 and #2-1b, DLX5 and DLX6 appear to be expressed in placental CTs and EVTs, suggesting their potential roles in CTs in addition to EVTs (**Response Fig. 2** and **Response Table 1**). Based on our findings using the *in vitro* model TSC lines, we proposed that TFAP2C occupies loci near the EVT-active genes, thereby priming the binding of DLX5 and DLX6 to these sites in TSCs. As EVT differentiation proceeds, the accessibility of cis-regulatory elements associated with the EVT-active genes increases due to a decrease in TFAP2C expression. This reduction in TFAP2C levels could facilitate the binding of DLX5 and DLX6, leading to upregulating EVT-active genes. Although speculative, considering the existence of some TFs that are initially located in the cytoplasm and later translocate into the nucleus to interact with target genomic sequences, the localization of DLX5 and DLX6 in placental CTs and EVTs will be of great interest for future study. In turn, Zadora et al. showed differential localization of DLX5 in villous trophoblasts and extravillous trophoblasts in human-term placenta (**Response Fig. 17a**)³⁷, and it was also observed that another trophoblast regulator Arid3a shows a context-dependent localization pattern in mouse early embryos (**Response Fig. 17b**)³⁸. While we admit that the human TSC models may not perfectly recapitulate *in vivo* counterparts, they still offer invaluable opportunities for studying underlying molecular mechanisms involved in EVT differentiation.

Response Fig. 16: TFAP2C occupancy in regulatory elements of EVT-active genes with low accessibility.

a, Gene track view illustrating ATAC-seq signals (green) in TSCs and EVT day 8 cells, along with TFAP2C (blue) and DLX6 (red) ChIP-seq signals in TSCs and differentiating cells to EVT at days 2 (EVT D2), 5 (EVT D5), and 8 (EVT D8). ATAC-qPCR targets were labeled for inactive (black), EVT-active (red), TSC-active (blue), and TSC/EVT-active (green) loci. **b**, ATAC-qPCR results validating the binding of TFAP2C to loci near EVT-active genes with low ATAC-seq signal. To account for primer variability, the Ct value was first normalized by subtracting the Ct value of sheared genomic DNA, resulting in a normalized Ct value. The deltaCt value was then calculated by subtracting the normalized Ct value of a genomic locus near an olfactory receptor gene (OR1A1), which exhibits very low ATAC-seq signal and low transcription levels in TSCs. The y-axis represents negative deltaCt values, indicating that an increase in value corresponds to increased openness.

Data from Zadora et al. (2017)

Response Fig. 17: Localization of DLX5 in human term placenta.

Immunofluorescence staining of term human placenta conducted by Zadora et al.³⁷ using antibodies for DLX5 and cytokeratin 7 (CK7), a trophoblast-specific cytoplasmic marker.

2-7. The EMT analyses and relationship to cancer (Presented in Fig. 7) is very preliminary, incomplete and out of the focus of the study. Authors should remove that section.

While we initially considered exploring the roles of EVT TFs involved in EVT invasion, a controlled process, in cancer EMT, an uncontrolled process, as an intriguing direction, developing this idea further would require considerable time and effort, as the reviewer pointed out. Therefore, as suggested, we have decided to exclude this aspect from the current manuscript and pursue it as a separate project for future research.

Minor Comments:

2-M1. As mentioned above, experimental procedure section needs more details.

We have added details of experimental procedures in the Methods section, and the revised text is also shown in **Extended Response Data Text 1** within this rebuttal.

2-M2. Supplementary tables should have more information (Mentioned above)

We have added details to the **Supplementary Tables**. The revised **Extended Response Data Table 1** also shows edited title and updates.

2-M3. Line 114 mentions “genes belonging to class ---which are highly active in fully differentiated EVT’s”. What is meant by “fully active”?

We regret not providing sufficient clarification. We attempted to convey that in the TFAP2C OE cells, the activation of EVT-active genes was not achieved to the same extent as in the control cells, even when subjected to EVT differentiation conditions. We revised the text, and it now reads as follows: “Accordingly, transcriptomic analysis revealed that EVT-active genes (classes 2 and 4) are not **adequately upregulated**, while the expression of TSC-active genes (class 1) is not downregulated normally in the TFAP2C OE cells (**Fig. 4h**).”

Reviewer #3

This manuscript by Kim et al. has attempted to track the trajectory of human trophoblast stem cells as they differentiate into extra-villous trophoblast (EVT). They focus on changes in the transcriptome and what the authors refer to as the "enhancer landscape". This is a valuable exercise because, surprisingly, the steps to EVT formation have not been pursued in any detail despite the availability of the TSC lines of Okae et al. for more than four years and the importance of EVT in relation to placental disease. Of particular interest is the role of TFAP2C in directing both the early and later steps of differentiation. Although much of the work is descriptive (though still of value) intervention in the process by shRNA silencing confirms many of the authors' inferences about the roles of particular groups of transcription factors. Moreover the description of a defined medium for TSC differentiation will be helpful to others.

The manuscript is clearly written and easy to follow. The conclusions are reasonable and not overblown. It provides data that will be useful to the ever growing number of groups interested in TB emergence and differentiation. I do, however, have a few comments that the authors should address.

We appreciate the reviewer's overall positive assessment of our study and valuable feedback. As the reviewer suggested, we have validated the importance of the EVT TFs we defined using additional TSC lines (both sexes and different origins). In addition, we have utilized recently published transcriptome data of EVT subtypes in human placentas to understand better the *in vitro* EVT differentiation system we used in our study. We hope that our revised manuscript adequately addresses the reviewer's concerns.

3-1. TSC lines, and particularly those originating from induced pluripotent stem, do not all behave uniformly. I was unsure whether only a single or multiple lines had been tested. If just a single line, how do we know whether the conclusions have general validity? Also the origin of the line(s), i.e., from blastocyst versus villous, and their sex should be made clear.

We appreciate the critical comment provided by the reviewer. The original manuscript primarily utilized the CT27 TSC line derived from CT in the human placenta (female). However, as the reviewer mentioned, multiple TSC lines derived from different origins (CT vs. blastocyst) and genders (female vs. male) are available, as outlined in **Response Table 2**. To ensure the consistency of our findings across multiple established TSC lines, we conducted EVT differentiation with additional TSC lines and observed similar morphological changes and consistent expression patterns of EVT TFs (**Response Fig. 3**). We also performed the KD of EVT TFs using multiple TSC lines and differentiated the cells into EVT to investigate whether the EVT regulators are essential for proper EVT differentiation. As presented in **Response Fig. 18**, the KD of individual EVT TFs resulted in impaired induction of EVT markers in all tested TSC lines, suggesting that our conclusion in the original manuscript is valid, regardless of the origin and gender. Among the new results, the data from the CT-derived TSC lines are included in the revised manuscript as **Extended Data Fig. 3a,c**. We also changed the main text accordingly (page 5).

Response Table 2. Origin and gender of the established TSC lines

Name	Origin	Gender
CT27	Cytotrophoblasts	Female
CT29	Cytotrophoblasts	Male
CT30	Cytotrophoblasts	Female
bTS5	Blastocysts	Female
bTS11	Blastocysts	Male

Response Fig. 18: Impaired EVT differentiation upon KD of EVT TFs in 5 different human TSC lines. TSC lines were infected with lentivirus particles expression pLKO shRNA targeting individual EVT TFs and differentiated to EVT. Relative expression levels of late-stage TFs to show KD efficiencies and EVT marker genes, including HLA-G and MMP2, in EVT day 8 cells were measured using RT-qPCR. The fold change was calculated relative to EVT day 8 cells infected with lentivirus expressing a non-targeting sequence. Error bars indicate mean \pm SD (n = 3). Statistical significance was determined using Student's t-test (*p < 0.05, **p < 0.01, ***p < 0.001).

3-2. EVT is likely not a single functional lineage, despite having overall similarities. For example is it agreed that the EVT that invades blood vessels is of the same genetic identity as those that form an EVT shell around the implanting conceptus or even give rise to the so-called trophoblast giant cells? This topic deserves some discussion in relation to differentiation trajectories.

We appreciate the insightful comment from the reviewer. As a similar comment was given by Reviewer #2, we kindly request referring to our response to comment #2-1a for detailed information. Briefly, we employed marker genes of different primary EVT subtypes identified from a recent spatial transcriptomics analysis and snRNA-seq of human placenta at 8–9 post-conceptual weeks⁴. Based on the expression patterns of these marker genes during *in vitro* EVT differentiation system, we found that the EVT differentiation *in vitro* closely recapitulates EVT differentiation in the human placenta, with a predominant bias toward iEVT differentiation (**Response Fig. 10**) We have further discussed the EVT lineages in the Discussion section of our revised manuscript (page 11).

Extended Response Data Fig. 1

Extended Response Data Fig. 2

Fig. 3a – TFAP2C KD

Fig. 3a – DLX5 KD

Extended Response Data Fig. 2 – continued

Fig. 3a – DLX6 KD

Fig. 3a – ASCL2 KD

Extended Response Data Fig. 2 – continued

Fig. 3a – ZNF439 KD

Fig. 3a – NRIP1 KD

Extended Response Data Fig. 2 – continued

Extended Data Fig. 4c

Extended Data Fig. 6d

Extended Response Data Fig. 3

Extended Response Data Fig. 4

Fig. 2b

Fig. 5d

Fig. 6a

Fig. 7a

Extended Response Data Fig. 4 – continued

Extended Data Fig. 6a

Extended Data Fig. 6b

Extended Data Fig. 7d

Extended Response Data Text 1

ATAC-seq and ATAC-qPCR

ATAC-seq libraries were prepared as previously described³⁹ using the ATAC-seq kit (Diagenode, C01080002) and 24 UDI for Tagmented libraries set II (Diagenode, C01011036). Briefly, 50,000 nuclei were pelleted and resuspended with tagmentase for 30 min at 37°C. The tagmented DNA was then purified and used for library generation through PCR amplification. Size selection was conducted using AMPure XP beads (Beckman coulter, A63881), and the purified ATAC-seq libraries were sequenced on the Illumina NovaSeq 6000 platform.

For ATAC-qPCR experiments, the final elution product of the ATAC protocol was diluted 1:80. qPCR was conducted using PerfeCTa SYBR Green FastMix Reaction Mixes (QuantaBio, 95072-012) and a StepOnePlus™ Real-Time PCR System (Thermo Fischer Scientific, brand Applied Biosystems). Genomic DNA was utilized as a control for primer variability. All primer sequences used for qPCR are in **Supplementary Table 6**. The ATAC-qPCR results were first normalized by subtracting the Ct value of genomic DNA, yielding a normalized Ct value. The deltaCt value was then calculated by subtracting the normalized Ct value of a genomic locus near olfactory receptor gene (OR1A1), which exhibits very low ATAC-seq signal and low transcription levels in TSCs. The data was presented as negative deltaCt values, where an increase in value corresponds to increased openness.

ChIP-seq, bioChIP-seq, and ATAC-seq data processing and analysis

In addition to the sequencing data we generated in this study, the published TFAP2C ChIP-seq in TSC data was used (GSE208539). The raw data were processed according to the following pipeline. The sequencing reads were aligned to the human reference genome hg38 using Bowtie2 (v2.4.5)⁴⁰ with the default setting. The aligned reads were filtered based on mapping quality (quality value > 10) using samtools⁴¹. The data was visualized on Integrative genomics viewer (IGV)⁴² with normalized wig files generated by deepTools (`bamCoverage -bs 10 --normalizeUsing CPM --extendReads 67`)⁴³. Individual peaks were called by MACS3⁴⁴ with default settings. Further score filtering was applied to remove weak peaks, and peaks found in simple redundant genome regions were also filtered out. Enriched motifs of binding loci of each TFs were identified using findMotifsGenome.pl module with the default setting under HOMER (v4.11)⁴⁵. For motif analysis of histone ChIP-seq data (H3K27ac ChIP-seq peaks in EVT D3), a region size of 500 bp was used as recommended. Quantitative comparison of ChIP-seq and ATAC-seq peaks was performed with MAnorm (v1.3.0)⁴⁶ with the default setting. The results were visualized using Java TreeView⁴⁷, and GO enrichment analysis was performed using the GREAT software⁴⁸.

Identification of super-enhancers

H3K27ac peaks were used to identify super-enhancers using the ROSE program^{34, 35}. Briefly, H3K27ac peaks called by MACS3 were transferred to gff files and used as input for the ROSE program. The program was run with the default setting along with the `-s 12500` and `-t 2500` options, which exclude peaks within 2,500 pb from the TSS to avoid capturing promoter peaks.

Mapping peaks to gene

The associated genes of ChIP-seq or bioChIP-seq peaks were mapped by navigating surrounding regions (± 20 Kb). All RefSeq genes in the refFlat file (hg38) downloaded from the UCSC genome browser were used for gene mapping. The genomic distribution of the ChIP-seq or bioChIP-seq peaks was monitored

based on the following criteria: intron, exon, promoter region (within \pm 2 Kb from the TSS), and intergenic region (binding sites except for intron, exon, and promoter).

Virus preparation and infections

Bacterial stocks containing lentiviral shRNA for a target gene were purchased from MilliporeSigma (MISSION shRNA library, **Supplementary Table 7**). Lentiviral vector carrying shRNA for KD or non-target sequence (MilliporeSigma, SHC202) was transfected to HEK293T cells at 70% confluency using GenJet™ In Vitro DNA Transfection Reagent (SignaGen Laboratories, SL100488), according to the manufacturer's instructions. After 18 h, HEK293T medium was replaced with TSC medium. Viral supernatants were harvested 24 h and 48 h following medium replacement. To KD a target gene, TSC were infected in viral supernatant **containing polybrene** (Santa Cruz Biotechnology, sc-134220) **at a final concentration of 10 μ g/mL**. **After 18 h, the infected cells were induced to differentiate into EVT using EVT differentiation media supplemented with puromycin at a final concentration of 1 μ g/mL**. **For the time-course KD of TFAP2C, the viral media was applied to TSC for early knockdown (Early-KD), and to differentiating cells at day 3 (Mid-KD) and day 5 (Late-KD)**. **The differentiation media supplemented with puromycin was used according to the specific differentiation time, replacing the media after 18 hours of infection.**

Stable cell line generation for inducible overexpression of tagged proteins

To establish stable cell lines with inducible OE of a target gene, we initially modified the pSBtet-GP vector (Addgene, plasmid #60495), a sleeping beauty vector with inducible promoter. The luciferase sequence in the vector was replaced with NotI and NheI restriction enzyme sites, and a flag tag and a biotinylation sequence were inserted at the N-terminal of the enzyme sites. This modified vector was referred to as the SBFB vector. To establish stable cell lines with inducible overexpression of a target gene, we initially modified the pSBtet-GP vector (Addgene, plasmid #60495), a sleeping beauty vector with an inducible promoter. The luciferase sequence in the vector was replaced with NotI and NheI restriction enzyme sites, and a flag tag along with a biotinylation sequence was inserted at the N-terminal of the enzyme sites. This modified vector was referred to as the SBFB vector. Next, the coding sequence of individual target genes was inserted into the SBFB vector. This construct, along with a vector containing the sequence for the BirA enzyme and a transposase-containing pCMV(CAT)T7-SB100 vector (Addgene, plasmid #34879), was transfected into TSC. After 24 h, the cells were subjected to selection using puromycin at a final concentration of 1 μ g/mL and G418 at a final concentration of 250 μ g/mL. To induce protein expression, doxycycline (Fisher Scientific) was administered at a final concentration of 1 μ g/mL.

Extended Response Data Table 1

	Revised Title	Revision
Supplementary Table 1	Count information of RNA-seq data	Newly added
Supplementary Table 2	Class 1-4 genes and their changes in expression during EVT differentiation	
Supplementary Table 3	Super-enhancer loci and associated genes	The information on the locus and associated genes of SEs defined by H3K27ac and other markers, which was newly conducted during the revision, has been updated.
Supplementary Table 4	DEGs upon KD of EVT TFs	
Supplementary Table 5	Genes associated with G1-G3 loci	
Supplementary Table 6	Primers used for RT-qPCR	Primer sequences that were newly used in the revision were updated.
Supplementary Table 7	Oligo sequences of shRNAs	

Response references

1. Okae, H. *et al.* Derivation of Human Trophoblast Stem Cells. *Cell Stem Cell* **22**, 50-63.e56 (2018).
2. Vento-Tormo, R. *et al.* Single-cell reconstruction of the early maternal-fetal interface in humans. *Nature* **563**, 347-353 (2018).
3. Suryawanshi, H. *et al.* A single-cell survey of the human first-trimester placenta and decidua. *Sci Adv* **4**, eaau4788 (2018).
4. Arutyunyan, A. *et al.* Spatial multiomics map of trophoblast development in early pregnancy. *Nature* **616**, 143-151 (2023).
5. Varberg, K.M. *et al.* ASCL2 reciprocally controls key trophoblast lineage decisions during hemochorial placenta development. *Proc Natl Acad Sci U S A* **118** (2021).
6. Subramanian, A. *et al.* Gene set enrichment analysis: a knowledge-based approach for interpreting genome-wide expression profiles. *Proc Natl Acad Sci U S A* **102**, 15545-15550 (2005).
7. Liberzon, A. *et al.* Molecular signatures database (MSigDB) 3.0. *Bioinformatics (Oxford, England)* **27**, 1739-1740 (2011).
8. Schneider, C.A., Rasband, W.S. & Eliceiri, K.W. NIH Image to ImageJ: 25 years of image analysis. *Nat Methods* **9**, 671-675 (2012).
9. Marsh, B., Zhou, Y., Kapidzic, M., Fisher, S. & Belloch, R. Regionally distinct trophoblast regulate barrier function and invasion in the human placenta. *Elife* **11** (2022).
10. Wolf, B.K. *et al.* Cooperation of chromatin remodeling SWI/SNF complex and pioneer factor AP-1 shapes 3D enhancer landscapes. *Nat Struct Mol Biol* **30**, 10-21 (2023).
11. Balsalobre, A. & Drouin, J. Pioneer factors as master regulators of the epigenome and cell fate. *Nat Rev Mol Cell Biol* **23**, 449-464 (2022).
12. Kim, M., Adu-Gyamfi, E.A., Kim, J. & Lee, B.K. Super-enhancer-associated transcription factors collaboratively regulate trophoblast-active gene expression programs in human trophoblast stem cells. *Nucleic Acids Res* **51**, 3806-3819 (2023).
13. Gao, Y. *et al.* CXCL5/CXCR2 axis promotes bladder cancer cell migration and invasion by activating PI3K/AKT-induced upregulation of MMP2/MMP9. *Int J Oncol* **47**, 690-700 (2015).
14. Li, H. *et al.* Knockdown of TRIM31 suppresses proliferation and invasion of gallbladder cancer cells by down-regulating MMP2/9 through the PI3K/Akt signaling pathway. *Biomed Pharmacother* **103**, 1272-1278 (2018).
15. Sun, L. *et al.* Epigenetic Regulation of a Disintegrin and Metalloproteinase (ADAM) Transcription in Colorectal Cancer Cells: Involvement of β -Catenin, BRG1, and KDM4. *Front Cell Dev Biol* **8**, 581692 (2020).
16. Peixoto, P. *et al.* EMT is associated with an epigenetic signature of ECM remodeling genes. *Cell Death Dis* **10**, 205 (2019).
17. Liu, S. *et al.* Estrogen receptor beta promotes lung cancer invasion via increasing CXCR4 expression. *Cell Death Dis* **13**, 70 (2022).
18. Bachelder, R.E., Wendt, M.A. & Mercurio, A.M. Vascular endothelial growth factor promotes breast carcinoma invasion in an autocrine manner by regulating the chemokine receptor CXCR4. *Cancer Res* **62**, 7203-7206 (2002).
19. Zhu, D. *et al.* A promising new cancer marker: Long noncoding RNA EGFR-AS1. *Front Oncol* **13**, 1130472 (2023).
20. Zhang, X. *et al.* Notum is required for neural and head induction via Wnt deacylation, oxidation, and inactivation. *Dev Cell* **32**, 719-730 (2015).
21. Kakugawa, S. *et al.* Notum deacylates Wnt proteins to suppress signalling activity. *Nature* **519**, 187-192 (2015).
22. Dietrich, B., Haider, S., Meinhardt, G., Pollheimer, J. & Knöfler, M. WNT and NOTCH signaling in human trophoblast development and differentiation. *Cell Mol Life Sci* **79**, 292 (2022).

23. Musa, J., Aynaud, M.M., Mirabeau, O., Delattre, O. & Grünewald, T.G. MYBL2 (B-Myb): a central regulator of cell proliferation, cell survival and differentiation involved in tumorigenesis. *Cell Death Dis* **8**, e2895 (2017).
24. Münz, M. *et al.* The carcinoma-associated antigen EpCAM upregulates c-myc and induces cell proliferation. *Oncogene* **23**, 5748-5758 (2004).
25. Sudo, T. *et al.* Activation of Cdh1-dependent APC is required for G1 cell cycle arrest and DNA damage-induced G2 checkpoint in vertebrate cells. *Embo j* **20**, 6499-6508 (2001).
26. Logan, C.Y. & Nusse, R. The Wnt signaling pathway in development and disease. *Annu Rev Cell Dev Biol* **20**, 781-810 (2004).
27. Ren, Q., Chen, J. & Liu, Y. LRP5 and LRP6 in Wnt Signaling: Similarity and Divergence. *Front Cell Dev Biol* **9**, 670960 (2021).
28. Shy, B.R. *et al.* Regulation of Tcf7l1 DNA binding and protein stability as principal mechanisms of Wnt/ β -catenin signaling. *Cell Rep* **4**, 1-9 (2013).
29. Lambert, A.W. & Weinberg, R.A. Linking EMT programmes to normal and neoplastic epithelial stem cells. *Nat Rev Cancer* **21**, 325-338 (2021).
30. Abhinand, C.S., Raju, R., Soumya, S.J., Arya, P.S. & Sudhakaran, P.R. VEGF-A/VEGFR2 signaling network in endothelial cells relevant to angiogenesis. *J Cell Commun Signal* **10**, 347-354 (2016).
31. Hnisz, D. *et al.* Super-enhancers in the control of cell identity and disease. *Cell* **155**, 934-947 (2013).
32. Rada-Iglesias, A. *et al.* A unique chromatin signature uncovers early developmental enhancers in humans. *Nature* **470**, 279-283 (2011).
33. Creyghton, M.P. *et al.* Histone H3K27ac separates active from poised enhancers and predicts developmental state. *Proc Natl Acad Sci U S A* **107**, 21931-21936 (2010).
34. Whyte, W.A. *et al.* Master transcription factors and mediator establish super-enhancers at key cell identity genes. *Cell* **153**, 307-319 (2013).
35. Lovén, J. *et al.* Selective inhibition of tumor oncogenes by disruption of super-enhancers. *Cell* **153**, 320-334 (2013).
36. Frank, S.B., Schulz, V.V. & Miranti, C.K. A streamlined method for the design and cloning of shRNAs into an optimized Dox-inducible lentiviral vector. *BMC Biotechnol* **17**, 24 (2017).
37. Zadora, J. *et al.* Disturbed Placental Imprinting in Preeclampsia Leads to Altered Expression of DLX5, a Human-Specific Early Trophoblast Marker. *Circulation* **136**, 1824-1839 (2017).
38. Rhee, C. *et al.* Arid3a is essential to execution of the first cell fate decision via direct embryonic and extraembryonic transcriptional regulation. *Genes Dev* **28**, 2219-2232 (2014).
39. Buenrostro, J.D., Wu, B., Chang, H.Y. & Greenleaf, W.J. ATAC-seq: A Method for Assaying Chromatin Accessibility Genome-Wide. *Curr Protoc Mol Biol* **109**, 21.29.21-21.29.29 (2015).
40. Langmead, B. & Salzberg, S.L. Fast gapped-read alignment with Bowtie 2. *Nat Methods* **9**, 357-359 (2012).
41. Li, H. *et al.* The Sequence Alignment/Map format and SAMtools. *Bioinformatics (Oxford, England)* **25**, 2078-2079 (2009).
42. Robinson, J.T. *et al.* Integrative genomics viewer. *Nat Biotechnol* **29**, 24-26 (2011).
43. Ramírez, F. *et al.* deepTools2: a next generation web server for deep-sequencing data analysis. *Nucleic Acids Res* **44**, W160-165 (2016).
44. Zhang, Y. *et al.* Model-based analysis of ChIP-Seq (MACS). *Genome Biol* **9**, R137 (2008).
45. Heinz, S. *et al.* Simple combinations of lineage-determining transcription factors prime cis-regulatory elements required for macrophage and B cell identities. *Mol Cell* **38**, 576-589 (2010).
46. Shao, Z., Zhang, Y., Yuan, G.C., Orkin, S.H. & Waxman, D.J. MAnorm: a robust model for quantitative comparison of ChIP-Seq data sets. *Genome Biol* **13**, R16 (2012).
47. Saldanha, A.J. Java Treeview--extensible visualization of microarray data. *Bioinformatics* **20**, 3246-3248 (2004).

48. McLean, C.Y. *et al.* GREAT improves functional interpretation of cis-regulatory regions. *Nat Biotechnol* **28**, 495-501 (2010).

REVIEWER COMMENTS

Reviewer #2 (Remarks to the Author):

The revised manuscript addresses many of my concerns. Especially, many new experiments address my concerns about discrepancy of expression patterns of transcription factors during EVT differentiation. Although some of my concerns are not fully addressed, it is understandable that the human TSC model has limitations and does not completely represent the in vivo EVT development, especially the endovascular EVT population. I do not have any other concern regarding this manuscript.

Reviewer #3 (Remarks to the Author):

The authors have done a considerable amount of new experimental work and provided an improved manuscript that is responsive to my original critique.

Editorial note

Reviewer#2 was additionally asked to comment in place of reviewer 1 who was unavailable to provide a comment during this round:

My Evaluation about the response of authors against each comments of Reviewer 1.

Major comments:

1) The second sentence in the abstract claims that regulatory factors of EVT differentiation are unknown, while the statements at the end of the first and second Introduction paragraphs are more tempered and appropriate as the role of some factors like ASCL2 in EVT differentiation has been recently described (PMID: 33649217). This should be mentioned in the Introduction and not just in the Results section.

My Comments: Revised manuscript has addressed this by adding new sentence and reference.

2) The efficiency of EVT differentiation has not been sufficiently characterized in Fig.1, as FACS quantification, immunostainings and comparison to real EVT transcriptome is missing. Since the paper is based on the bulk RNA-seq analysis coming up with potential EVT differentiation regulators, the efficiency and quality of the differentiation are essential to demonstrate. RNA expression dynamics of two markers is not sufficient to characterize any cell type.

My Comments: Revised manuscript has addressed this in Response Fig. 1 and in Extended Data Fig. 1a-C.

3) Throughout the manuscript, the authors focus on the 4 gene clusters identified in Fig.1 at the expense of EVT-related genes. As a result, the manuscript lacks insights into EVT biology and does not sufficiently discuss the results in this context. This point must be addressed in the revised version.

My Comments: Although Revised manuscript has addressed this and rebuttal letter referred to data presented in Fig. 1e, 3d, 4i, and Extended Data Fig. 3g, the manuscript should be further revised based on the recent findings by Arutyunyan et al. Nature paper.

4) In Fig.2, where two kinds of EVT regulators are being postulated, a basic immunostaining of these in the 1st. trimester placental tissue is necessary. In particular, in the context of DLX5 and DLX6 expression in CT (Fig.2e).

My Comments: This is still a major problem with the manuscript. Revised manuscript and the rebuttal letter used additional TSC lines mentioned multiple single-cell datasets to argue the repression of TFAP2C expression in EVT's and induction of DLX5/6 and ZNF 439 expressions in EVTS using TSC differentiation model. However, in a developing human placenta the CTBs express high levels of DLX5, DLX6 and ZNF439 and similar to TFAP2C they are partially suppressed in primary EVT's. In fact, ZNF439 suppression is stronger than TFAP2. This is in contrast to the observation in TSCS. Whereas it is understandable that TSC differentiation model might behave differently compared to the in vivo EVT's, in the context of this manuscript this is a significant Flaw. The manuscript stressed upon the importance of DLX5, DLX6 and ZNF439 as late EVT regulating factors. However, the in vivo expression data does not support that.

To support their conclusion they should at least perform the following experiments.

- (i) They should ectopically express these genes in undifferentiated TSCs similar to Primary CTB levels and show that the overexpression does not alter the gene expression patterns in stem.

- (ii) *In overexpression stem cell population The TFAP2C and DLX6 do not bind to same target genes.*
- (iii) *Overexpression of these genes does not suppress TFAP2C in stem-state TSCs and during early-stage EVT differentiation.*
- (iv) *They should analyze (differentiation and gene expression) of the overexpressed TSCs with or without TFAP2C knockdown for early EVT differentiation. If the authors are right, even with overexpression of these genes, EVT differentiation will be impaired in TFAP2C-KD cells.*

5) The EVT migration assays used in Fig.3b is an important measure of functionality and should therefore include a quantitative readout.

My Comments: The revised manuscript addressed this concern.

6) In general, the KDs in Fig.3 should be characterized first in an unbiased way (i.e., not only in the context of the 4 clusters) and more comprehensively. How are other lineage markers affected by these KDs? Are the cells stuck in a progenitor state or do they deviate into other lineages (as reported for the ASCL2-KD, PMID: 33649217)?

My Comments: The revised manuscript addressed this concern by mentioning data in Fig. 3D and Extended data Fig. 3g, h and in Response Fig. 5a-c.

7) As in Fig.3, the characterization of the TFAP2C-KD time course in Fig.4 does not clearly show what happens to other key lineage genes except HLA-G, MMP2 and TP63 during the differentiation. A heatmap containing a set of key trophoblast TFs and comparing their expression changes during the time course would be useful. What genes are impacted most by the KDs/OE and how does this relate back to trophoblast biology.

My Comments: The revised manuscript addressed this concerns in Fig. 4, b, f, j,i. Extended Data fig. 4h,I and in Response Fig. 6.

8) The integration of the ChIP-seq data from Fig.5 with the KD expression data from Fig.3/4 is surprisingly missing. This is essential to show which genes are directly bound and regulated/co-regulated by a given TF. What is the overlap of bound vs. misregulated genes? When are these bound genes expressed? To what extent TFAP2C and DLX6 share target genes overall and not just in specific clusters?

My Comments: The revised manuscript and response letter addressed this concern in Response Fig. 7a-c and in Extended Data Fig. 5g,i.

9) Along the same lines, while comprehensive ChIP-seq/RNA-seq analysis have been performed, they've not been comprehensively analyzed in an integrated manner. For instance, the authors make important conclusions (e.g., lines 263-276) and reference only genome browser snapshots without the global perspective (Fig.5d, Fig.6a and Extended Fig. 6a,b).

My Comments: The revised manuscript addressed this concern by mentioning data in Fig. 5d,g,h.

10) Line235-237: Conclusion that "TFAP2C bound to its targets in self-renewing TSCs and cells in early-stage day 2, followed by decreased occupancy as differentiation progressed. On the other hand, we detected DLX6 target occupancy starting at day 5 of differentiation (Fig.5b)" is based on the number of peaks. Instead, it should be based on a quantitative analysis of TFAP2C and DLX6 binding (i.e., differential binding analysis).

My Comments: *The revised manuscript addressed this concern by mentioning data in Extended Data Fig. 5c,d.*

11) The display G1-G3 loci in Fig.3e,f is misleading. The authors should present data obtained for each time-point.

My Comments: *The revised manuscript addressed this concern by mentioning data in Fig. 5e,f and Extended data Fig. 3g, h and in Response Fig. 5h.*

12) A part of the paper focuses on the putative TFAP2C pioneer activity without discussing/excluding other possibilities. What criteria does a pioneer factor need to fulfill? Is it possible that TFAP2C acts as a repressor of late EVT genes, instead as a pioneer-activator? Pioneer factors do not just prime genes for later expression but bring in immediate change to the local chromatin environment by recruiting other chromatin modifying/remodeling activities, which has not been demonstrated.

My Comments: *The response letter has addressed this concern by referring a new NAR paper by the authors and by mentioning data in different figures of the revised manuscript.*

13) A lot (in my view too much) prominence is given to the general GO-term analysis without mentioning specific gene examples. Instead, the authors should focus on what genes are affected and how this impacts EVT biology. Are there new interesting target genes that are bound by TFAP2C/DLX6? What critical stage in EVT development/function may they regulate?

My Comments: *The revised manuscript addressed this concern in texts of pages 5-6 and 8.*

14) The ability of DLX5/6 to activate EMT genes should also be demonstrated in EVT cells and not just cancer cells. The statement that SNAI1 might be an EVT-specific EMT TF is misleading as SNAI1 is a well-known general EMT regulator in different contexts. The general connection between cancer EMT and EVT differentiation could be explored a bit more by data mining. For instance, are there other relevant expression signatures shared by cancers and the EVT lineage? What about TFAP2C in this context? Is it also upstream of DLX genes in cancers? The general idea of faulty re-activation of extra-embryonic genes could be discussed as a wider context.

My Comments: *These aspect is removed from the revised manuscript based on my suggestion.*

15) The Discussion needs a substantial revision in terms of structure. More detailed discussion of pioneer vs. possible repressor functions might be beneficial. It is requested that the authors discuss the results in the context of EVT biology beyond just general aspects of EMT.

My Comments: *The revised manuscript addressed this concern.*

Minor comments:

- 1) Fig.2c is unreadable.
- 2) The original scans for the Western blots are missing.
- 3) Fig.4j and Fig.1f: x-axis labeling would be beneficial.
- 4) The EVT KD labeling in Fig. 7f is unreadable.
- 5) All genome browser snap-shots lack scale.
- 6) The authors should specify settings of their bioinformatic analyses, also when standard settings were used.

Significance

Taken together, this study represents a significant advance in exploring the role of TFs during EVT differentiation using a novel human TSC model. After revisions, the paper would be relevant not just to the trophoblast development and reproductive medicine community, but also in more general terms related to TF biology.

My Comments: *The revised manuscript addressed all these concerns.*

Overall Response

We appreciate the reviewer's suggestions, which contributed to clarifying our revised manuscript. While acknowledging the inherent limitations of the *in vitro* human trophoblast stem cell (TSC) model in replicating the *in vivo* environment, we have carefully considered the reviewer's recommendations for specific experiments to reinforce the robustness of our conclusions. We have conducted additional experiments to validate and substantiate our claims in response to these suggestions. Furthermore, we have incorporated relevant findings from the recent study by Arutyunyan et al., as suggested by the reviewer, into our revised manuscript. We hope our responses sufficiently address the reviewer's concerns.

REVIEWER COMMENTS AND POINT-BY-POINT RESPONSES

3) Throughout the manuscript, the authors focus on the 4 gene clusters identified in Fig.1 at the expense of EVT-related genes. As a result, the manuscript lacks insights into EVT biology and does not sufficiently discuss the results in this context. This point must be addressed in the revised version.

My Comments: Although Revised manuscript has addressed this and rebuttal letter referred to data presented in Fig. 1e, 3d, 4i, and Extended Data Fig. 3g, the manuscript should be further revised based on the recent findings by Arutyunyan et al. Nature paper,

We appreciate the reviewer's comment, and we have incorporated the use of marker genes identified by Arutyunyan et al. [1] to provide additional support for the inappropriate EVT differentiation observed in EVT factor KD cells (page 5). In fact, during our previous revision, we added a paragraph discussing the *in vitro* EVT differentiation system, utilizing recent data by Arutyunyan et al. In response to the reviewer's comment, we further revised our manuscript by expending our discussion on the immaturity and absence of certain cell types in the *in vitro* EVT differentiation of human TSC model, referencing the findings from Arutyunyan et al.'s study (page 11).

4) In Fig.2, where two kinds of EVT regulators are being postulated, a basic immunostaining of these in the 1st. trimester placental tissue is necessary. In particular, in the context of DLX5 and DLX6 expression in CT (Fig.2e).

My Comments: This is still a major problem with the manuscript. Revised manuscript and the rebuttal letter used additional TSC lines mentioned multiple single-cell datasets to argue the repression of TFAP2C expression in EVTs and induction of DLX5/6 and ZNF 439 expressions in EVTS using TSC differentiation model. However, in a developing human placenta the CTBs express high levels of DLX5, DLX6 and ZNF439 and similar to TFAP2C they are partially suppressed in primary EVTs. In fact, ZNF439 suppression is stronger than TFAP2. This is in contrast to the observation in TSCS. Whereas it is understandable that TSC differentiation model might behave differently compared to the *in vivo* EVTs, in the context of this manuscript this is a significant Flaw. The manuscript stressed upon the importance of DLX5, DLX6 and ZNF439 as late EVT regulating factors. However, the *in vivo* expression data does not support that.

To support their conclusion they should at least perform the following experiments.

(i) They should ectopically express these genes in undifferentiated TSCs similar to Primary CTB levels and show that the overexpression does not alter the gene expression patterns in stem.

(iii) Overexpression of these genes does not suppress TFAP2C in stem-state TSCs and during early-stage EVT differentiation.

We appreciate the reviewer's insightful and constructive comments. In response to the reviewer's suggestions (i) and (iii), we ectopically expressed DLX5 and DLX6 in TSCs and monitored their impact on the self-renewal of TSCs and the expression levels of TFAP2C. According to scRNA-seq data retrieved from the Human Protein Atlas [2, 3], DLX5 and DLX6 expression levels in CT are 69% and 47% of those in *in vivo* EVT, respectively. However, values vary in other datasets, such as bulk RNA-seq from Okae et al. [4] and another scRNA-seq dataset [5]. In response, we attempted to induce the expression of DLX5 and DLX6 in TSCs to levels similar to or slightly higher than those detected in *in vitro* EVT. To achieve this, we treated inducible DLX5 or DLX6 TSC lines used in our original manuscript with varying concentrations of doxycycline (Dox).

We then monitored the levels of DLX5 or DLX6 in the Dox-treated TSCs. As controls, non-treated TSCs and *in vitro* differentiated EVTs were tested by RT-qPCR. As shown in **Response Fig. 1a**, the cells treated with 62.5 ng/mL of Dox exhibited expression levels of DLX5 or DLX6 similar to control EVT. We further confirmed the similar or slightly higher induction levels of DLX5 or DLX6 in TSCs treated with Dox at 62.5 ng/mL compared to EVT through western blot and immunofluorescence (**Response Fig. 1b, c** and **Extended Response Data Fig. 1**). Therefore, we opted to utilize cells treated with Dox at a concentration of 62.5 ng/mL to mimic *in vivo* induction levels of DLX5 or DLX6 in CTs. Subsequently, we examined the impact of DLX5 or DLX6 induction on TSC context. As shown in **Response Fig. 2a**, the expression levels of TSC self-renewal markers (TP63, TEAD4, GATA2, ELF5, and MSX2) were not significantly altered with the overexpression of DLX5 or DLX6. Although the cells treated with 16 times higher concentrations of Dox (1000 ng/mL) displayed a slightly different morphology compared to the control cells, the cells with similar DLX5 or DLX6 levels to those in EVT (Dox, 62.5 ng/mL) exhibited a morphology comparable to the control cells (**Response Fig. 2b**).

Addressing the reviewer's suggestion, we also monitored the expression levels of TFAP2C in DLX5 or DLX6 overexpressed TSCs and early EVT differentiating cells. In line with the expression patterns of self-renewal markers, we observed that the levels of TFAP2C in TSCs, with the induction of DLX5 or DLX6 like the levels in EVT, were comparable to the TFAP2C's level in the control cells (**Response Fig. 2c**). Similarly, the levels of TFAP2C in EVT day 2 cells treated with Dox at 62.5 ng/mL, and even at 125 ng/mL, were comparable to TFAP2C's level in control cells (**Response Fig. 2d**). Of note, the expression levels of TFAP2C decreased in cells with super induction of DLX5 or DLX6 (Dox 1000 ng/mL) in both self-renewal and early differentiation settings.

In summary, within the contexts of self-renewing TSCs and early EVT differentiation, the ectopic expression of DLX5 or DLX6 to levels observed in EVT did not influence the self-renewal of TSCs or the expression of TFAP2C. This observation aligns with the data presented in our original manuscript (**Fig. 6g**) as we observed that only upon the super induction of these late-stage TFs did the resulting cells show decreased levels of TFAP2C. Therefore, in our revised manuscript, we have tempered and removed the portion describing direct suppression of TFAP2C by late-stage TFs to enhance the clarity of our findings. The Title and the Abstract have been adjusted accordingly.

Response Fig. 1: Ectopic induction levels of DLX5 and DLX6.

a, Expression levels of DLX5 and DLX6 in control TSCs (blue line), *in vitro* differentiated EVT (red line), and cells treated with doxycycline (Dox) for induction of DLX5 or DLX6 for 6h, 1 day, 2 days, and 6 days. Δ Ct values, calculated by subtracting the Ct value of GAPDH from the Ct value of DLX5 or DLX6, are presented. Error bars represent the mean \pm SD of two independent experiments ($n = 2$). **b**, Protein expression levels of DLX5 or DLX6 in inducible TSC lines treated with various concentrations of Dox for 2 days and *in vitro* differentiated EVTs. Ectopically expressed biotinylated DLX5 (Exo-DLX5) or DLX6 (Exo-DLX6) are indicated by blue arrows, and endogenous proteins are marked by red arrows. ACTB was used as the loading control. **c**, Immunofluorescence analysis of DLX5 and DLX6 in TSCs induced with DLX5 or DLX6 using different concentrations of Dox for 2 days and in EVT. Scale bar: 50 μ m.

Response Fig. 2: Effects of DLX5 or DLX6 induction on TSC self-renewal and TFAP2C expression. **a**, Relative mRNA expression (fold change) of self-renewal marker genes in TSCs induced with DLX5 or DLX6. The fold change was calculated relative to the expression in non-treated TSCs (-Dox control). Error bars indicate mean \pm SD ($n = 2$). **b**, Brightfield images of TSCs overexpressing DLX5 (top panels) or DLX6 (bottom panels). Non-treated TSCs (-Dox) were used as control. Scale bar: 100 μ m. **c** and **d**, Relative mRNA expression (fold change) of TFAP2C in DLX5- or DLX6-induced TSCs (**c**) at different time points (6h, day 1, day2, and day 6) and EVT differentiation day 2 (**d**). The fold change was calculated relative to the expression in non-treated TSCs (-Dox control). Error bars represent the mean \pm SD of two independent experiments ($n = 2$).

(ii) In overexpression stem cell population The TFAP2C and DLX6 do not bind to same target genes.

As recommended, we investigated the binding ability of induced DLX6 in TSCs to its target loci mapped in fully differentiated EVT, which are occupied by TFAP2C in TSCs, to test whether DLX6 binds to the same targets or not. We performed ChIP-qPCR using a native DLX6 antibody (NBP1-85929, Novus Biologicals) upon ectopic expression of DLX6 at levels similar to those in EVT, as shown in **Response Fig. 1**. We additionally tested cells with higher induction of DLX6 (cells treated with Dox at 125 ng/mL or 1000 ng/mL) and fully differentiated EVT as a positive control to demonstrate the binding of DLX6 on its target loci in EVT. For qPCR analysis, we tested the primers for the common loci where TFAP2C occupies in TSCs and during early EVT differentiation, and DLX6 occupies during late differentiation (**Response Fig. 3a** and **Extended Response Data Table 1**). As shown in **Response Fig. 3b**, even at high levels of DLX6 induction (cells treated with Dox at 1000 ng/mL), we did not observe significant DLX6 occupancy at the selected targets. However, we detected DLX6 occupancy on these target loci in EVT control cells as expected. These findings suggest that a simple induction of DLX6 in TSCs is insufficient for DLX6 to bind its genomic targets in EVT. Comprehensive changes during EVT differentiation, such as increased expression of co-factors, other TFs, chromatin status, and other reasons, might be necessary for DLX6's genomic binding and proper functioning for EVT functions. Alternatively, DLX6 expressed in CT *in vivo* may play cell-type specific roles, which need to be addressed in future studies.

Response Fig. 3: DLX6 binding in TSC on cis-regulatory elements of EVT-active genes occupied by TFAP2C.

a, Gene track view displaying TFAP2C (blue) and DLX6 (red) signals in TSCs, EVT D2, EVT D5, and EVT D8 at the locus near EVT-active genes. **b**, Binding of ectopically expressed DLX6 in TSCs on cis-regulatory elements associated with genes active in EVT, where TFAP2C binds. The binding was measured by ChIP-qPCR using a native DLX6 antibody. DLX6 binding affinity in EVT on the same loci served as the control. Values were calculated relative to the corresponding values in each input. Error bars represent the mean \pm SD of two independent experiments (n = 2).

(iv) They should analyze (differentiation and gene expression) of the overexpressed TSCs with or without TFAP2C knockdown for early EVT differentiation. If the authors are right, even with overexpression of these genes, EVT differentiation will be impaired in TFAP2C-KD cells.

In response to the reviewer’s suggestion aimed at validating the necessity of TFAP2C for early EVT differentiation, we initiated the overexpression of DLX5 or DLX6 from day 3 in EVT differentiating cells with TFAP2C knockdown (KD). On differentiation day 8, we collected the cells and investigated whether the induction of DLX5 or DLX6 could rescue the impaired EVT differentiation caused by TFAP2C KD. As depicted in **Response Fig. 4**, the overexpression of DLX5 or DLX6, even in cells with super induction (cells treated with Dox at 1000 ng/mL), still exhibited impaired induction of EVT marker genes (HLA-G and MMP2) similar to the levels shown in control TFAP2C KD cells (no Dox control). These results suggest that the induction of a single late-stage TF is insufficient to activate overall EVT differentiation. These findings further support our conclusion that TFAP2C is required for proper EVT differentiation.

Response Fig. 4: Inability of DLX5 or DLX6 overexpression to rescue EVT differentiation in TFAP2C knockdown TSC.

Relative mRNA expression (fold change) of TFAP2C, demonstrating KD efficiency, and EVT marker genes in TFAP2C-KD cells upon induction of DLX5 or DLX6 during EVT differentiation, achieved by treating with doxycycline (Dox 1000 ng/mL from EVT differentiation day 3 to day 8). Fold changes were calculated relative to cells infected with non-targeting shRNA. Error bars represent the mean \pm SD of two independent experiments (n = 2).

Extended Response Data Fig. 1

DLX5 induction levels

DLX6 induction levels

Extended Response Data Table 1

Locus	Forward (5'-3')	Reverse (5'-3')
HLA-G 0.5k up	CCTGACATTCTAGAAGCTTCACAAG	TGTTCTCTCTATAGTGGTCCTGCTA
HLA-G 11k up	CTACTTCTTGCTTTCCCTTTCATGC	TCTATGTTGCAAGCTAGCCTAAACT
ITGA1 8k up	ACATCTTCCAAGTCCCTTTTACCAT	CTTATGGGTTGTGGTAGGCAGAATA
ADAMTS20 40k up	CTTCTTTCTTCTTCGTCACCACAAC	GAATGGGCAGCTTTGAGATCTAGAT
ASCL2 70k down	GAGTGGGTGAGAGTCAGACAG	CAGGAGATGCTGTGGTTTTGTG
DLX6 3.5k up	TTTACCATGCACCACAACACTACCA	GGCTCTCAACCTTCCCCTAATAG
ZNF439 Intron 1	CCTGAGAAGTTTGAGCTTGTGAAA	ATACAACCTCCCTGGACCACTATCT
ZNF439 Intron 2	AATGTTAATGGGAGTTGAAGCTGTG	GTTGGGACCTAACATACTCAGACA

References

1. Arutyunyan, A., et al., *Spatial multiomics map of trophoblast development in early pregnancy*. Nature, 2023. **616**(7955): p. 143-151.
2. Karlsson, M., et al., *A single-cell type transcriptomics map of human tissues*. Sci Adv, 2021. **7**(31).
3. Vento-Tormo, R., et al., *Single-cell reconstruction of the early maternal-fetal interface in humans*. Nature, 2018. **563**(7731): p. 347-353.
4. Okae, H., et al., *Derivation of Human Trophoblast Stem Cells*. Cell Stem Cell, 2018. **22**(1): p. 50-63.e6.
5. Suryawanshi, H., et al., *A single-cell survey of the human first-trimester placenta and decidua*. Sci Adv, 2018. **4**(10): p. eaau4788.

REVIEWERS' COMMENTS

Reviewer #2 (Remarks to the Author):

The new experiments involving over-expression of DLX5 and DLX6 in TSCs and subsequent analyses presented in the response letter largely satisfy my concerns about the manuscript. Authors have correctly modified the text to removed the portion describing direct suppression of TFAP2C by late-stage TFs to justify their new findings. The modifications of the Title and the Abstract is also justified.

I have only one minor suggestion.

The claims about ZNF439 is still not fully supported by experiments. Authors should carefully validate expression levels of ZNF439 in vivo from multiple available dataset and include their findings in the discussion section highlighting any differences with in vitro human TSC model.

REVIEWER COMMENTS AND POINT-BY-POINT RESPONSES

Reviewer #2 (Remarks to the Author):

The new experiments involving over-expression of DLX5 and DLX6 in TSCs and subsequent analyses presented in the response letter largely satisfy my concerns about the manuscript. Authors have correctly modified the text to removed the portion describing direct suppression of TFAP2C by late-stage TFs to justify their new findings. The modifications of the Title and the Abstract is also justified.

I have only one minor suggestion.

The claims about ZNF439 is still not fully supported by experiments. Authors should carefully validate expression levels of ZNF439 in vivo from multiple available dataset and include their findings in the discussion section highlighting any differences with in vitro human TSC model.

We are delighted that our additional experiments have alleviated the concerns of the reviewer regarding the expression of DLX5 and DLX6 in cytotrophoblasts. Also, we appreciate the reviewer's suggestion regarding the expression levels of ZNF439. In response to this suggestion, we have described the comparable expression levels of ZNF439 in cytotrophoblasts compared to extravillous trophoblasts by referencing multiple studies on in vivo expression profiles of human placenta cells, including Okae et al., 2018 (PMID: 29249463), Vento-Tormo et al., 2018 (PMID: 30429548), Suryawanshi et al., 2018 (PMID: 30402542), and Arutyunyan et al., 2023 (PMID: 36991123). Furthermore, we have highlighted the observed differences in the expression levels of some late-stage TFs between in vivo and in vitro model systems, as also observed in Okae et al., 2018 (PMID: 29249463)'s transcriptome profiling data. We hope this change acknowledges readers and the reviewer of the differences between TSC models and their in vivo counterpart.